# Genomic investigations of unexplained acute hepatitis in children

Since its first identification in Scotland, over 1,000 cases of unexplained paediatric hepatitis in children have been reported worldwide, including 278 cases in the UK[1]. Here we report an investigation of 38 cases, 66 age-matched immunocompetent controls and 21 immunocompromised comparator participants, using a combination of genomic, transcriptomic, proteomic and immunohistochemical methods. We detected high levels of adeno-associated virus 2 (AAV2) DNA in the liver, blood, plasma or stool from 27 of 28 cases. We found low levels of adenovirus (HAdV) and human herpesvirus 6B (HHV-6B) in 23 of 31 and 16 of 23, respectively, of the cases tested. By contrast, AAV2 was infrequently detected and at low titre in the blood or the liver from control children with HAdV, even when profoundly immunosuppressed. AAV2, HAdV and HHV-6 phylogeny excluded the emergence of novel strains in cases. Histological analyses of explanted livers showed enrichment for T cells and B lineage cells. Proteomic comparison of liver tissue from cases and healthy controls identified increased expression of HLA class 2, immunoglobulin variable regions and complement proteins. HAdV and AAV2 proteins were not detected in the livers. Instead, we identified AAV2 DNA complexes reflecting both HAdV-mediated and HHV-6B-mediated replication. We hypothesize that high levels of abnormal AAV2 replication products aided by HAdV and, in severe cases, HHV-6B may have triggered immune-mediated hepatic disease in genetically and immunologically predisposed children.

In March 2022, the report of five cases of severe hepatitis of unknown aetiology led to the UK Health Security Agency (UKHSA) identifying 278 cases in total as of 30 September 2022[1]. Cases, defined as acute non-A–E hepatitis with serum transaminases of more than 500 IU in children under 10 years of age, were found to have been occurring since January 2022[2]. In the UK, 196 cases required hospitalization, 69 were admitted to intensive care and 13 required liver transplantation[1]. Case numbers have declined since April 2022[3].

UKHSA investigations identified HAdV to be commonly associated with the unexplained paediatric hepatitis, with 64.7% (156 of 241) testing positive in one or more samples from whole blood (the most sensitive sample type[4]) or mucosal swabs. HAdVs from the blood of 35 of 77 patients were typed as F41. Seven of eight patients in England who required liver transplantation tested HAdV positive in blood samples, with F41 found in five of five genotyped[2]. SARS-CoV-2 infection was detected in 8.9% (15 of 169) of UK and 12.8% (16 of 125) of English cases[2].

Given the uncertainty around the aetiology of this outbreak, and the potential that HAdV-F41, if implicated (Fig. 1a), could be a new or recombinant variant, we undertook untargeted metagenomic and metatranscriptomic sequencing of liver biopsies from five liver transplant cases and whole blood from five non-transplanted cases (Table 1 and Fig. 1b). The results were further verified by confirmatory PCRs of liver, blood, stool and nasopharyngeal samples from a total of 38 cases for which there was sufficient residual material. We compared our results with those from 13 healthy children and 52 previously healthy children presenting to hospital with other febrile illness, including HAdV, hepatitis unrelated to the current outbreak or a critical illness requiring admission to the intensive care unit. We also tested blood and liver biopsies from 17 profoundly immunosuppressed children with hepatitis who were not part of the current outbreak, in whom reactivation of latent infections might be expected.

## Cases

We received samples from 38 children meeting the case definition (Table 1). All cases were less than 10 years of age and 22 of 23 cases previously tested were positive by HAdV PCR (Table 2, Extended Data Table 1 and Supplementary Table 1). A summary of the samples received from these cases and the investigations carried out on them are shown in Fig. 1b,c.

## Clinical details

Pre-existing conditions, autoimmune, toxic and other infectious causes of hepatitis were excluded in 12 transplanted (cases 1–5, 28, 29, 31–34 and 36) and four non-transplanted (cases 30, 35, 37 and 38) children, investigated at two liver transplant units (Supplementary Table 1). The 12 transplanted cases reported gastrointestinal symptoms (nausea, vomiting and diarrhoea) preceding transplant by a median of 20 days (range 8–42 days). All 12 transplanted children survived, whereas the four children who did not receive liver transplants recovered without sequelae or evidence of chronic liver-related conditions. Five of the remaining 22 cases referred by Health Security Agencies, for whom

A list of authors and there affiliations appears at the end of the paper.

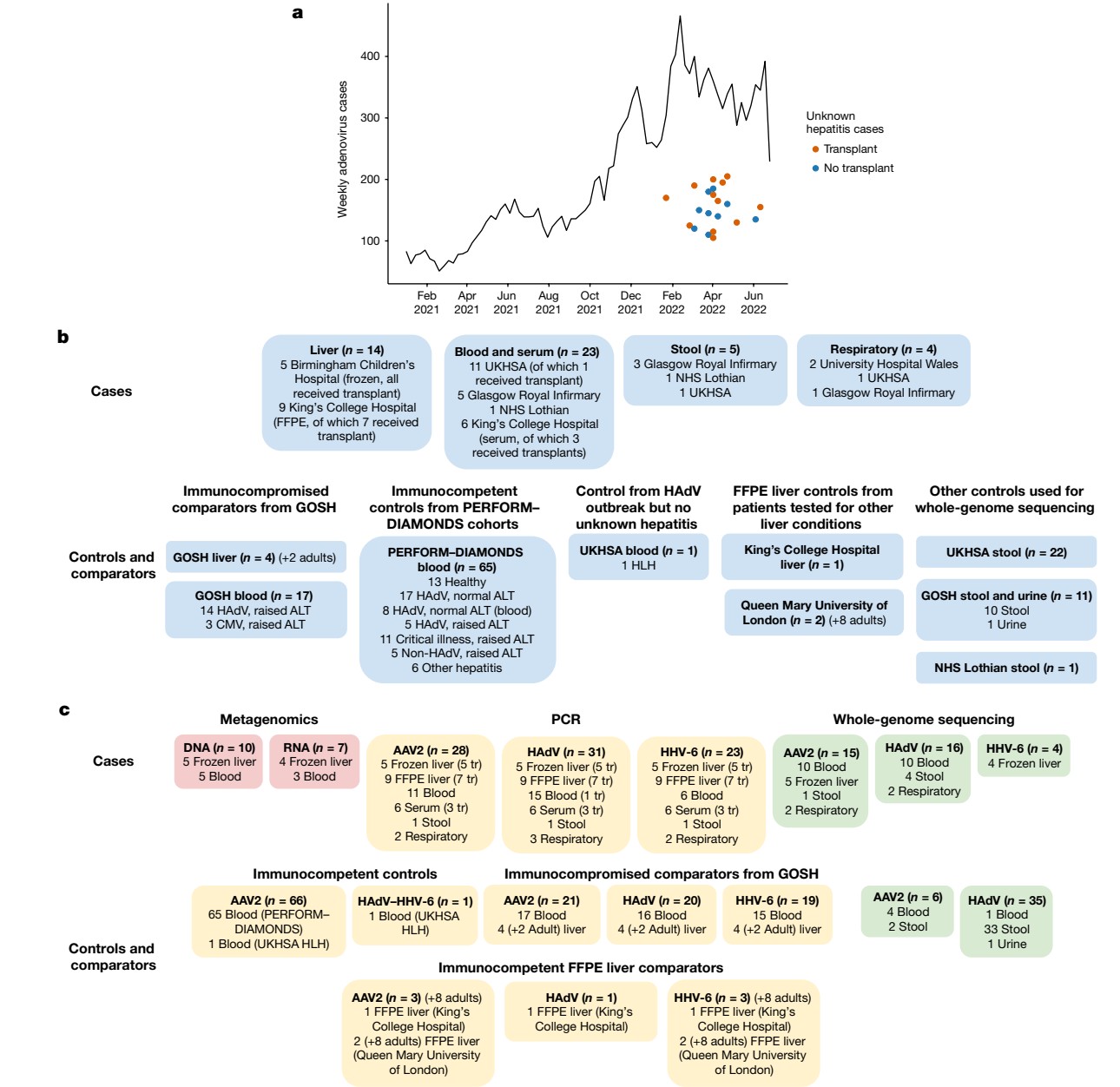

**Fig. 1 | HAdV epidemiology and experimental outline. a**, HAdV in all sample types (epidemiology since January 2022). Source: second-generation surveillance system data, that is, laboratory reports to UKHSA of a positive HAdV result conducted by a laboratory in England and includes any sample type. Dots represent the day of presentation for the 28 of 38 cases for which we had data. **b**, Case and control specimens by source. CMV, cytomegalovirus; HLH, haemophagocytic lymphohistiocytosis. **c**, Tests carried out by specimen type. More detail on samples tested and the results can be found in Tables 1 and 2. Not all tests were carried out on all samples due to lack of material. *n* refers to the total number of cases or controls. The numbers of each sample type may not sum to this total because samples of more than one type were sometimes taken from the same patient. For details, see Table 1. FFPE, formalin-fixed paraffin-embedded; tr, received liver transplant.

this information was available, recovered without sequelae (Table 1 and Supplementary Table 1).

## Metagenomic sequencing

We performed metagenomic and metatranscriptomic sequencing on samples of frozen explanted liver tissue from five cases who received liver transplants (median age of 3 years) and six blood samples from five non-transplanted hepatitis cases (median age of 5 years) (Table 1 and Fig. 1b). The liver samples had uniform and consistently high sequencing depth both for DNA sequencing (DNA-seq) and RNA-seq, whereas the blood samples had variable sequencing depth particularly for RNA-seq (Supplementary Table 2). We detected[5] abundant AAV2 reads in DNA-seq from five of five explanted livers and four of five blood samples from non-transplant cases (7–42 and 1.2–42 reads per million, respectively) (Table 2). Lower levels of HHV-6B were present in DNA-seq of all explanted liver samples (0.09–4 reads per million) but not in the six blood samples (Table 2). HAdV was detected (five reads) in one blood sample (Table 2).

## Evidence of AAV2 replication

Metatranscriptomics revealed AAV2, but not HHV-6B or HAdV, RNA reads, in liver and blood samples (0.7–10 and 0–7.8 reads per million,

## Table 1 | Characteristics of unexplained paediatric hepatitis cases and related specimens

| Case ID | Sex | Liver transplant | Sender | Specimen 1 | ID 1 | Specimen 2 | ID 2 | Specimen 3 | ID 3 |
|---|---|---|---|---|---|---|---|---|---|
| 1 | M | Yes | BCH | Liver | JBL1 | | | | |
| 2 | M | Yes | BCH, PHW | Liver | JBL4 | NPA | JBN1 | | |
| 3 | F | Yes | BCH | Liver | JBL3 | | | | |
| 4 | M | Yes | BCH, UKHSA | Liver | JBL2 | Blood | JBB25 | | |
| 5 | F | Yes | BCH | Liver | JBL5 | | | | |
| 6 | F | No | UKHSA | Blood | JBB9 | Blood | JBB14 | Blood | JBB16 |
| 7 | F | No | UKHSA | Blood | JBB11 | Blood | JBB10 | | |
| 8 | F | No | UKHSA | Serum | JBPL1 | Blood | JBB13 | | |
| 9 | M | No | UKHSA | Blood | JBB1 | | | | |
| 10 | M | No | UKHSA | Blood | JBB15 | | | | |
| 11 | NA | No | GRI | Blood | JBB2 | | | | |
| 12 | M | No | UKHSA | Blood | JBB12 | | | | |
| 13 | NA | No | GRI | Blood | JBB7 | | | | |
| 14 | NA | No | GRI | Blood | JBB8 | | | | |
| 15 | NA | No | GRI | Blood | JBB4 | Blood | JBB3 | | |
| 16 | NA | No | GRI | Blood | JBB5 | | | | |
| 17 | F | No | UKHSA | Throat swab | JBB18 | Stool | JBB17 | | |
| 18 | F | No | UKHSA | Blood | JBB19 | | | | |
| 19 | F | No | UKHSA | Blood | JBB20 | Blood | JBB23 | | |
| 20 | M | No | UKHSA | Blood | JBB21 | | | | |
| 21 | NA | No | PHW | NPA | JBB26 | | | | |
| 22 | NA | No | GRI | Stool | JBB27 | | | | |
| 23 | NA | No | GRI | Throat swab | JBB28 | Stool | JBB30 | | |
| 24 | NA | No | GRI | Stool | JBB29 | | | | |
| 25 | NA | No | NHSL | Blood | JBB31 | | | | |
| 26 | NA | No | NHSL | Stool | JBB32 | | | | |
| 27 | F | No | UKHSA | Blood | JBB24 | | | | |
| 28 | M | Yes | KCH | Liver | JBL6 | | | | |
| 29 | F | Yes | KCH | Liver | JBL7 | Liver | JBL8 | | |
| 30 | F | No | KCH | Liver | JBL9 | | | | |
| 31 | F | Yes | KCH | Liver | JBL10 | | | | |
| 32 | M | Yes | KCH | Liver | JBL11 | Serum | JBB34 | | |
| 33 | F | Yes | KCH | Liver | JBL12 | | | | |
| 34 | M | Yes | KCH | Liver | JBL13 | Serum | JBB36 | | |
| 35 | F | No | KCH | Liver | JBL14 | Serum | JBB35 | | |
| 36 | M | Yes | KCH | Liver | JBL15 | Serum | JBB37 | | |
| 37 | F | No | KCH | Serum | JBB38 | | | | |
| 38 | M | No | KCH | Serum | JBB39 | | | | |

The median age for the cases is 3 years of age (age range: 1–9 years of age). Case 10 was 9 years of age. All other cases were 7 years of age or younger.

Cases 1–5 underwent liver transplant and had metagenomic next-generation sequencing (mNGS), PCR and viral whole-genome sequencing (WGS) of their specimens. Cases 28, 29, 31–34 and 36 also underwent liver transplant and had PCR for all three viruses under investigation. BCH sent the liver explant for case 2, PHW the NPA. BCH sent the liver for case 4, UKHSA the blood.

Cases 6–27, 30, 35, 37 and 38 did not receive a liver transplant. Cases 30 and 35 had liver biopsies. Cases 6–10 had metagenomic next-generation sequencing, PCR and viral WGS on their samples. Cases 11–22 had PCR for one to two of the viruses under investigation and viral WGS of positive PCRs. Cases 23–27 only had HAdV WGS on their samples and there was no residual material for further testing. Cases 31, 36, 38 and 39 had PCR for all three viruses under investigation.

BCH, Birmingham Children's Hospital; F, female; GRI, Glasgow Royal Infirmary; KCH, King's College Hospital; M, male; NA, not applicable; NHSL, NHS Lothian; NPA, nasopharygeal aspirate; PHW, Public Health Wales.

respectively). Mapping liver RNA-seq data to the RefSeq AAV2 genome (NC_001401.2) identified high expression of the Cap open reading frame, particularly at the 3′ end of the capsid, suggesting viral replication[6] (Extended Data Fig. 1a), whereas reverse transcription (RT)–PCR of two livers confirmed the presence of AAV2 mRNA from the Cap open reading frame (Extended Data Fig. 1c). In the blood samples, which had not been treated to preserve RNA, we detected low levels of AAV2 RNA reads mapping throughout the genome (Extended Data Fig. 1b).

## Nanopore sequencing of explanted livers

Ligation-based untargeted nanopore sequencing was applied to DNA from four of five frozen liver samples. All four samples were initially sequenced at a lower depth (average N50 of 8.37 kb). Six to sixteen AAV2 reads were obtained from each sample (5.57–22.24 million total reads; Supplementary Table 3). Mapping revealed concatenation of the 4-kb genome, compatible with active AAV2 replication[7]. We observed alternating and head-to-tail concatemers, which could

**Table 2 | PCR, metagenomics and viral WGS results from cases in which metagenomic sequencing was performed**

| Case ID | Sample ID | PCR Ct values | | | Metagenomics reads | | | | | | Viral WGS coverage (10×) | | |
|---|---|---|---|---|---|---|---|---|---|---|---|---|---|
| | | | | | DNA | | | RNA | | | | | |
| | | AAV2 | HAdV | HHV-6B | AAV2 | HAdV | HHV-6B | AAV2 | HAdV | HHV-6B | AAV2 | HAdV | HHV-6B |
| *Liver* | | | | | | | | | | | | | |
| 1 | JBL1 | 17 | 37 | 29 | 1,343 | 0 | 8 | 574 | 0 | 0 | 97 | – | 3 |
| 2 | JBL4 | 21 | 42 | 32 | 360 | 0 | 8 | 49 | 0 | 0 | 93 | – | 2 |
| 3 | JBL3 | 20 | 37 | 30 | 1,189 | 0 | 4 | 95 | 0 | 0 | 98 | – | 2 |
| 4 | JBL2 | 20 | 37 | 27 | 1,564 | 0 | 203 | 42 | 0 | 0 | 98 | – | 94 |
| 5 | JBL5 | 21 | 37 | 28 | 266 | 0 | 12 | F | F | F | – | – | – |
| *Blood* | | | | | | | | | | | | | |
| 6[a] | JBB14, JBB16, JBB9 | 24 | 36 | 37 | 151 | 0 | 0 | 77 | 0 | 0 | 95 | 35.5 | – |
| 7 | JBB10, JBB11 | 21 | 36 | 37 | 103 | 0 | 0 | F | F | F | 49 | F | – |
| 8 | JBPL1, JBB13 | 25 | P/N | –/N | 277 | 0 | 0 | 165 | 0 | 0 | 94 | F | – |
| 9 | JBB1 | 19 | P/– | P/– | 1,936 | 5 | 0 | 0 | 0 | 0 | 94 | F | – |
| 10 | JBB15 | –/N | N/N | 37 | 0 | 0 | 0 | F | F | F | – | F | – |

Where two results are shown, the first refers to the referring laboratory and the second to GOSH. Where there was a discrepancy, the positive result is shown. Where there is more than one sample for a single patient, Ct values represent the mean across the samples that were tested. De novo assembly of unclassified metagenomics reads was unremarkable. –, not tested (at GOSH due to insufficient residual material); F, failed; N, negative PCR result; P, positive PCR result in referring laboratory.
[a]For metagenomics reads, the result of combining the datasets from two blood samples from the same case.

be consistent with both HAdV and human herpesvirus-mediated rolling hairpin and rolling circle replication, respectively[8]. Two of these samples were sequenced more deeply, resulting in 52 and 178 AAV2 reads in 82.9 and 122 million total (N50 of 4.40–8.52 kb) (Supplementary Table 3). Of the reads in the more deeply sequenced datasets, 42–48% comprised randomly linked, truncated and rearranged genomes, with few that were intact and of full length (Extended Data Fig. 2). The remaining reads were less than 3,000 bp long and may represent sections either of monomeric genomes or of more complex structures.

## Integration analysis

There was some evidence of AAV2 integration by deeper nanopore sequencing of explanted livers (Supplementary Table 3); however, none of the integration sites was confirmed by Illumina metagenomic or targeted AAV2 sequencing. The results are likely to represent artefacts of this library preparation method; chimeric reads have been described to occur in 1.7–3% of reads[9,10]. Given the number of human reads (72–120 million), we might expect to see this artefact occurring most commonly between AAV2 and human than between AAV2 reads.

## Confirmatory real-time PCR

Where sufficient residual material was available, PCR tests were performed for AAV2 (28 of 38 cases), HAdV (31 of 38 cases) and HHV-6B (23 of 38 cases). The results confirmed high levels (cycle threshold (Ct) values: 17–21) of AAV2 DNA in all five frozen explanted livers that had undergone metagenomics (Table 2 and Fig. 2d), and lower levels of HHV-6B and HAdV DNA (Ct values: 27–32 and 37–42, respectively). AAV2 DNA was also detected (Ct values: 19–25) in blood samples from four of five cases that had undergone metagenomics, whereas HAdV, at levels too low to genotype, and HHV-6B were detected in two of four and three of four cases, respectively (one case had insufficient material) (Table 2). One of the blood metagenomics cases (case 9, JBB1) with insufficient material to test for HAdV and HHV-6B, tested positive for both viruses in the referring laboratory. The AAV2-negative blood sample (case 10, JBB15) was also negative for HAdV but positive for HHV-6B

(Table 2). A further ten of ten blood samples tested from cases were positive for HAdV by PCR. Sufficient material was available for AAV2 PCR in six of these (all positive; Ct values: 20–23) and HHV-6B PCR in two (one positive Ct value: 37) (Extended Data Table 1).

AAV2 PCR was positive in nine formalin-fixed paraffin-embedded (FFPE) liver samples, including seven from transplanted cases (Ct values: 23–25) and two from non-transplanted cases (Ct values: 34–36; Extended Data Table 1). HHV-6B PCR was positive in six of seven FFPE samples (not case 32) from transplanted (Ct values: 30–37) and zero of two from non-transplanted (cases 30 and 35) cases, with positive HAdV (Ct values: 40–44) in four of nine cases. Three transplanted (cases 32, 34 and 36) and three non-transplanted (cases 35, 37 and 38) cases had serum available for testing. All were AAV2 positive (Ct values: 27–32) and HHV-6B negative, with one transplanted case and one non-transplanted case testing HAdV positive (Extended Data Table 1).

Together, 27 of 28 cases tested were AAV2 PCR positive, 23 of 31 were HAdV positive and 16 of 23 were HHV-6B positive. When results from referring laboratories were included, 33 of 38 cases were positive for HAdV and 19 of 26 cases were HHV-6B positive (Table 2 and Extended Data Table 1).

## Controls and comparators

To better contextualize the findings in cases with unexplained hepatitis, we selected control groups of children who were not part of the outbreak.

## Blood from immunocompetent children

Whole blood from 65 immunocompetent children matched by age to cases (median age of 3.8 years) (Fig. 1b, Extended Data Table 2a and Supplementary Table 4) who were healthy, or had HAdV infection, hepatitis or critical illness, including requiring critical care, were selected from the PERFORM (personalised risk assessment in febrile illness to optimise real-life management; www.perform2020.org) and DIAMONDS (diagnosis and management of febrile illness using RNA personalised molecular signature diagnosis study; www.diamonds2020.eu) studies. Both studies recruited children presenting to hospital with an acute-onset febrile illness between 2017 and 2020 (PERFORM) and July 2020 to October 2021, during the COVID-19 pandemic

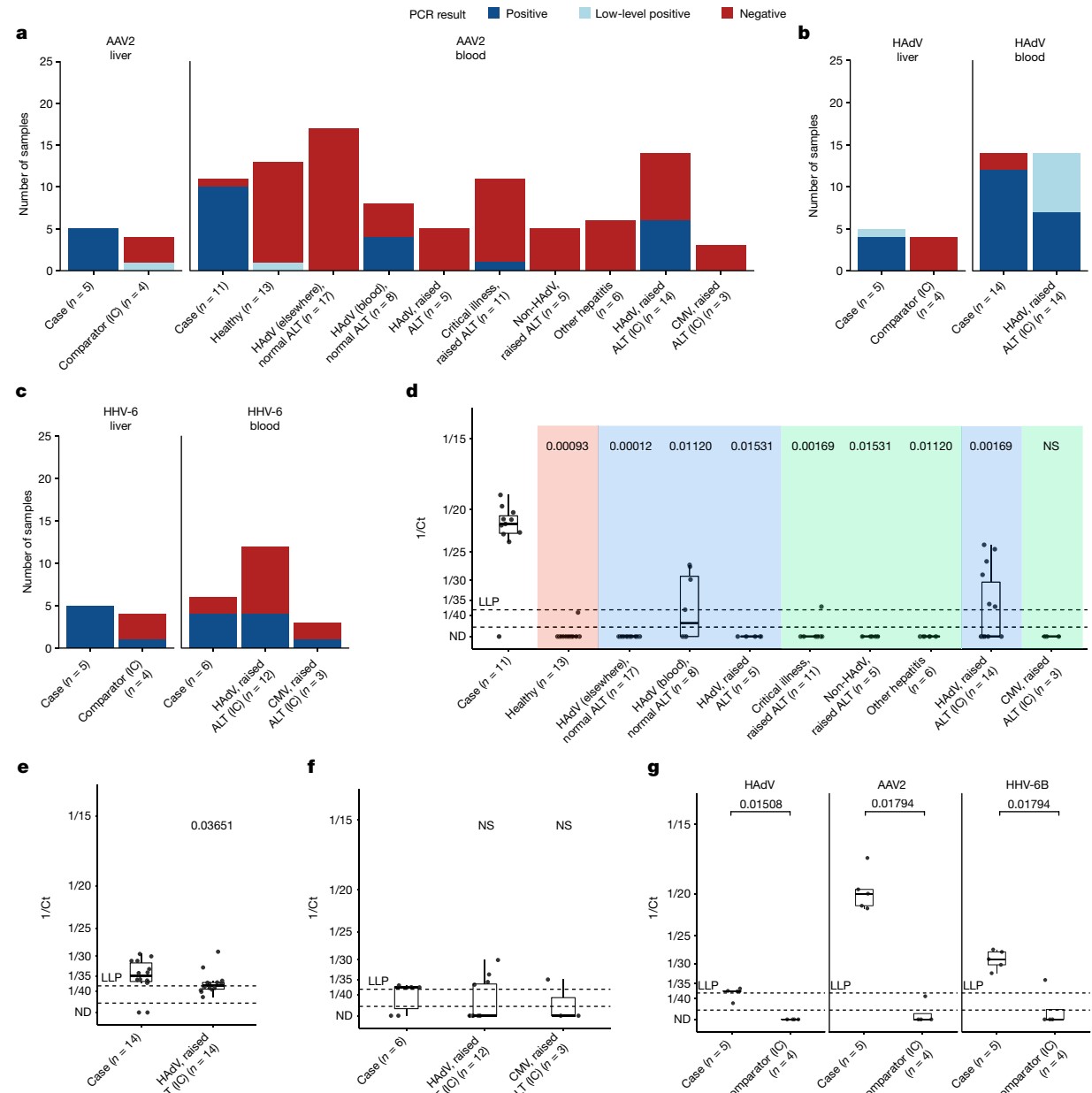

**Fig. 2 | Proportion of positive cases and viral loads (Ct values) for cases and controls. a**, Proportion of samples positive for AAV2. **b**, Proportion of samples positive for HAdV. **c**, Proportion of samples positive for HHV-6. Ct values less than 38 were defined as positive. Ct values more than 38 where the virus was detected within the maximum 45 cycles were defined as low-level positive (LLP). **d**, AAV2 in blood samples from cases, PERFORM–DIAMONDS immunocompetent controls and immunocompromised comparators (IC). HAdV infection is in blue, non-HAdV hepatitis is in green and healthy is in red. **e**, HAdV levels in whole blood from cases and immunocompromised comparators. **f**, HHV-6 in whole blood from cases and immunocompromised comparators. **g**, HAdV, AAV2 and HHV-6 levels in frozen liver tissue from cases and immunocompromised comparators. In the box plots, the bold middle line represents the median and the upper and lower horizontal lines represent the upper (75th percentile) and lower (25th percentile) quartiles, respectively. The whiskers show maximum and minimum values. Each point represents one case or control. *n* Refers to the number of cases or controls. Where more than one sample for a case was tested, the midpoint of the Ct was plotted. All repeat tests had values if less than 2 Ct values apart, that is, within the limits of methodological error. The upper dashed line marked LLP indicates the LLP threshold (Ct = 38). Points below the second dashed line represent samples below the limit of PCR detection (Ct = 45). Wilcoxon non-parametric rank sum tests were conducted for **e** and **g** and a Kruskal–Wallis test followed by pairwise Wilcoxon tests with a Benjamini–Hochberg correction for multiple comparisons were used for **d** and **f**. All tests were two-tailed. Numbers show the *P* value compared with cases. ND, not determined (negative PCR result); NS, not significant.

(DIAMONDS) (Supplementary Table 4). Of the PERFORM–DIAMONDS control whole-blood samples, 6 of 65 (9.2%) were AAV2 PCR positive (Supplementary Table 5), compared with 10 of 11 (91%) whole-blood samples from cases (Fig. 2a; $P = 8.466 \times 10^{-8}$, Fisher's exact test). AAV2 DNA levels were significantly higher in whole-blood samples from cases than from controls (Fig. 2e; $P = 2.747 \times 10^{-11}$, Mann–Whitney test).

One participant with an HAdV-F41-positive blood sample, originally thought to have unexplained paediatric hepatitis, was later found to have a previous condition that explained the hepatitis and was therefore reclassified as a control (referred to as 'reclassified control' or CONB40) (Supplementary Table 5). This blood sample was negative for AAV2 by PCR (Supplementary Table 5).

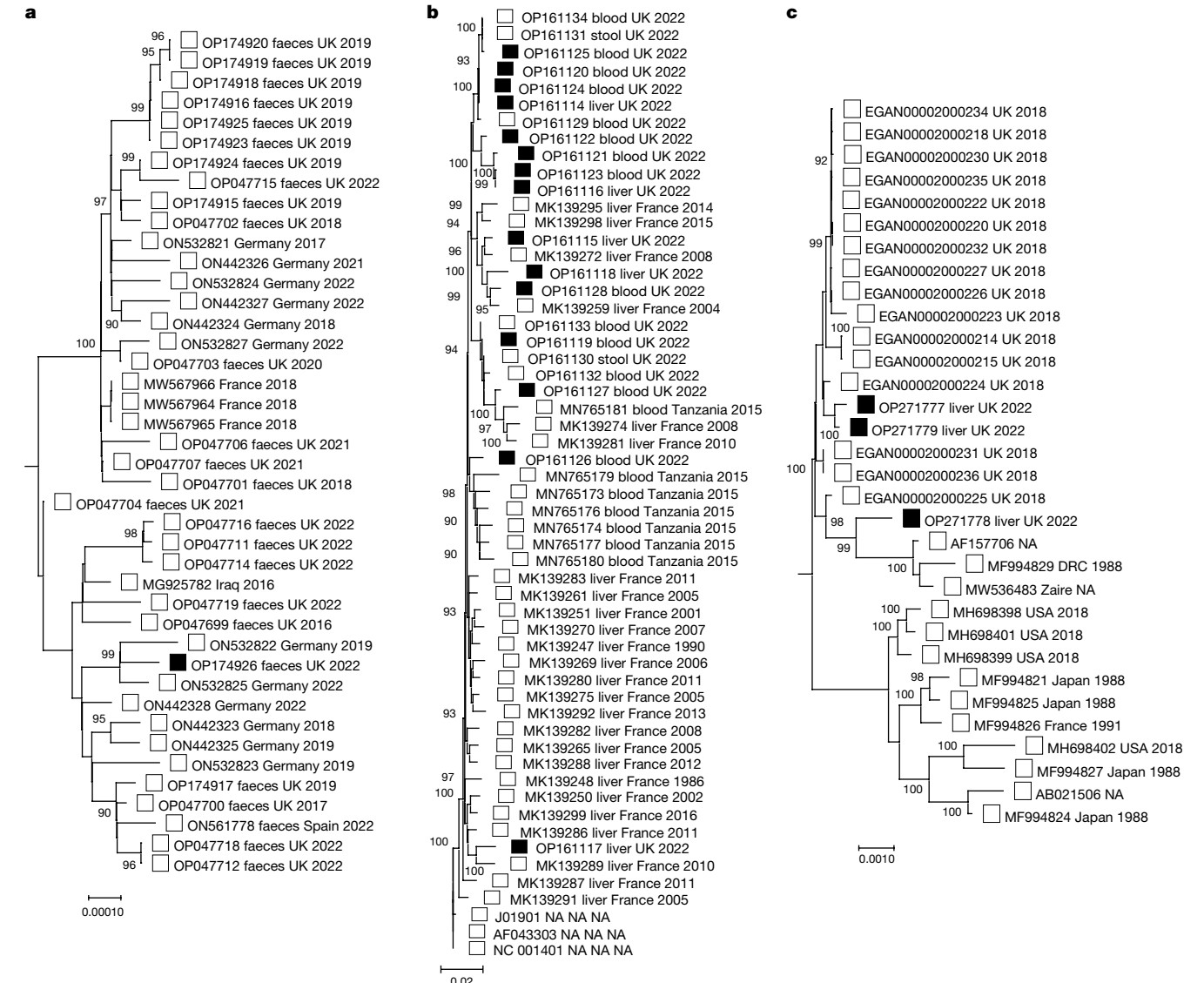

**Fig. 3 | Phylogenetic trees for HAdV, AAV2 and HHV-6B.** Maximum likelihood phylogenetic trees combining reference sequences from the RefSeq database, publicly available complete genomes from GenBank, UK non-outbreak controls (open squares) and unexplained hepatitis cases (black squares) for the different viruses involved. **a**, HAdV. **b**, AAV2. **c**, HHV-6B. HAdV and HHV-6B trees are midpoint rooted, whereas AAV2 is rooted from the RefSeq sequence NC_001401.2. Bootstrap values less than 90 are not shown. NA, value not known.

## Liver from immunocompromised children

Frozen liver biopsy material from four immunocompromised children (median age of 10 years) (CONL1–4) who had been investigated for other forms of hepatitis was also tested (Fig. 1b and Extended Data Table 2b). In three children, liver enzyme levels were raised (Supplementary Table 6); no results were available for CONL4. AAV2 was detected in CONL3 (Ct value: 39) and HHV-6B was detected in CONL2 (Ct value: 34), whereas HAdV was negative (Fig. 2d and Supplementary Table 5).

## Blood from immunocompromised comparators

We also tested immunocompromised children who are more likely to reactivate latent viruses. Whole-blood samples from 17 immuno-compromised children (median age of 1 year) with raised levels of liver transaminases (AST/ALT of more than 500 IU) and viraemia (HAdV or cytomegalovirus), all sampled in 2022 (Fig. 1b), were tested for AAV2, HHV-6B and HAdV (Extended Data Table 2b and Supplementary Table 5). The majority had received human stem cell or solid organ

transplants, and none was linked to the recent hepatitis outbreak (Extended Data Table 2b). Five of 15 (33%) whole-blood samples were positive for HHV-6B, whereas 6 of 17 (35%) were positive for AAV2, significantly fewer than in cases ($P = 0.005957$, Fisher's exact test) and at significantly lower Ct levels ($P = 6.517 \times 10^{-5}$, Mann–Whitney test) (Fig. 2 and Supplementary Table 5). One HAdV-positive and AAV2-positive immunocompromised comparator (CONB23) was also positive for HHV-6B (Supplementary Table 5).

Four of the six AAV2-positive children from the PERFORM–DIAMONDS cohort (Fig. 2a and Supplementary Table 5) and all six of the AAV2-positive immunocompromised children (Fig. 2a and Supplementary Table 5) were also HAdV positive.

## Viral whole-genome sequencing

One full HAdV-F41 genome sequence from the stool of one case (OP174926, case 22) (Supplementary Table 7) clustered phylogenetically with the HAdV-F41 sequence obtained from the reclassified control (CONB40) and with other HAdV-F41 sequences collected between

2015 and 2022, including 23 contemporaneous stool samples from children without the unexplained paediatric hepatitis (Figs. 1c and 3a). Sequencing and *k*-mer analysis[11] of HAdV from 13 cases with partial sequences identified the genotype HAdV-F41 in 12 cases (Supplementary Tables 7 and 8). The partial sequences showed most similarity to the control sequence OP047699 (Supplementary Table 8), mapping across the entire viral genome, thus further excluding a recombinant virus.

Single-nucleotide polymorphisms were largely shared between the single HAdV-positive stool from a case (OP174926) and control whole-genome sequences (Extended Data Fig. 3a). Given reported mutation rates for HAdV-F41 and other adenoviruses[12,13], any differences are likely to have arisen before the outbreak. No new or unique amino acid substitutions were noted in HAdV sequences from cases with only two substitutions overall (Extended Data Fig. 2d) and none in proteins critical for AAV2 replication.

AAV2 sequences from 15 cases, including five from the explanted livers and ten from whole blood from non-transplanted cases, clustered phylogenetically with control AAV2 sequences obtained from four immunocompromised HAdV-positive children with elevated levels of ALT in the comparator group (Extended Data Table 2b) and two healthy children with recent HAdV-F41 diarrhoea (Fig. 3b and Supplementary Table 9). The degree of diversity and lack of a unique common ancestor between case AAV2 genomes suggest that these are not specific to the hepatitis outbreak, but instead reflect the current viral diversity of the general population. Although comparison of the AAV2 sequences showed no difference between cases and controls, contemporary AAV2 sequences showed changes in the capsid compared with historic AAV2 (Extended Data Fig. 3c). None of these changes was shared with the hepatotropic AAV7 and AAV8 viruses (Extended Data Fig. 3b). The majority of the contemporary AAV2 genomes in cases and controls (20 of 21) contained a stop codon in the X gene, which is involved in viral replication[14], whereas historic AAV2 genomes contained this less frequently (11 of 35). The significance, if any, of this is currently unknown.

Although mean read depths for four HHV-6B genomes recovered from explanted livers were low (×5–10) (Supplementary Table 12), phylogeny (Fig. 3c) confirmed that all were different.

## Transduction of AAV2 capsid mutants

Using a recombinant AAV2 (rAAV2) vector with a VP1 sequence (Extended Data Fig. 4a) containing the consensus amino acid sequence from AAV2 cases (AAV2Hepcase) (Extended Data Fig. 3b), we generated functional rAAV particles that transduced Huh-7 cells with comparable efficacy to both canonical AAV2 and the synthetic liver-tropic LK03 AAV vector[15]. Unlike canonical AAV2, the AAV2Hepcase capsid, which contains mutations (R585S and R588T) that potentially affect the heparin sulfate proteoglycan (HSPG)-binding domain, was unaffected by heparin competition, a feature that is associated with increased hepatotropism[16,17] (Extended Data Fig. 4b,c).

## Histology and immunohistochemistry

Histological examination of the 12 liver explants and two liver biopsies showed nonspecific features of acute hepatitis with ballooning hepatocytes, disrupted liver architecture with varying degrees of perivenular, bridging or pan-acinar necrosis. There was no evidence of fibrosis suggestive of an underlying chronic liver disease. The appearances were similar to historic cases of seronegative hepatitis of unknown cause in children. There were no typical histological features of autoimmune hepatitis, notably no evidence of portal-based plasma cell-rich infiltrates. A cellular infiltrate was present in all cases, which on staining appeared to be predominantly of CD8[+] T cells but also included CD20[+] B cells. More widespread staining

with the CD79a pan-B cell lineage, which also identifies plasma cells, was also observed (Extended Data Fig. 5). Macrophage lineage cells showed some C4d complement staining, whereas staining for immunoglobulins was nonspecific with disruption of the normal canalicular staining seen in controls due to the architectural collapse. MHC class I and class II staining, although increased in cases, was nonspecific and associated with sinusoid-containing blood cells and necrotic tissue (Extended Data Fig. 6a). No viral inclusions were observed and there were no features suggestive of direct viral cytopathic effect.

Immunohistochemistry was negative for adenovirus. Staining of the five explanted livers with AAV2 antibodies demonstrated evidence of nonspecific ingested debris but not the nuclear staining seen in the positive AAV2-infected cell lines and infected mouse tissue (Extended Data Fig. 6b). All five liver explants showed positive staining of macrophage-derived cells with antibody to HHV-6B, with no staining of negative control serial sections (Extended Data Fig. 6b). No specific HHV-6B staining was observed in 13 control liver biopsies from patients (including three children less than 18 years of age) with other viral hepatitis, toxic liver necrosis, autoimmune and other hepatitis, and normal liver. The control set was also negative for HAdV and AAV2 by immunohistochemistry.

Liver sections were morphologically suboptimal for electron microscopy, but no viral particles were identified in hepatocytes, blood vessel endothelial cells and Kupffer cells.

## Transcriptomic analysis

We quantified functional cytokine activity by expression of independently derived cytokine-inducible transcriptional signatures of cell-mediated immunity (Supplementary Table 11) in bulk genome-wide transcriptional profiles from four of the frozen explanted livers. Results were compared with published data from normal adult livers ($n = 10$) and adult hepatitis B-associated acute liver failure ($n = 17$) (GSE96851)[18]. Data from the unexplained hepatitis cases revealed increased expression of diverse cytokines and pathways compared with normal liver. These pathways included prototypic cytokines associated with T cell responses, including IFNγ, IL-2, CD40LG, IL-4, IL-5, IL-7, IL-13 and IL-15 (Fig. 4a and Supplementary Table 12), as well as some evidence of innate immune type I interferon responses. Many of these responses showed substantially greater activity in unexplained hepatitis than in fulminant hepatitis B virus disease. The most striking enrichment was for TNF expression, and included other canonical pro-inflammatory cytokines including IL-1 and IL-6 (Extended Data Fig. 7). These data are consistent with an inflammatory process involving multiple pathways.

## Proteomics

Proteomic analysis of the five frozen explanted livers did not detect AAV2 or HAdV proteins. Expression of HHV-6B U4, a protein of unknown function, was found in four of five cases; U43, part of the helicase primase complex, was found in two of five cases; and U84, a homologue of cytomegalovirus UL117, implicated in HHV-6B nuclear replication, was found in two of five cases (Extended Data Fig. 8).

The human proteome from the five frozen liver explants was compared with publicly available data from seven control 'normal' livers, taken from two different studies[19,20]. Both protein and peptide analyses (Fig. 4b,c and Supplementary Tables 13 and 14) found increased expression in unexplained hepatitis cases of HLA class II proteins and peptides (for example, HLADRB1 and HLADRB4), multiple peptides from variable regions of the heavy and light chains of immunoglobulin, complement proteins (such as C1q) and intracellular and extracellular released proteins from neutrophils and macrophages (MMP8 and MPO).

There was no evidence of HAdV, AAV2 or HHV-6B in any of the control livers.

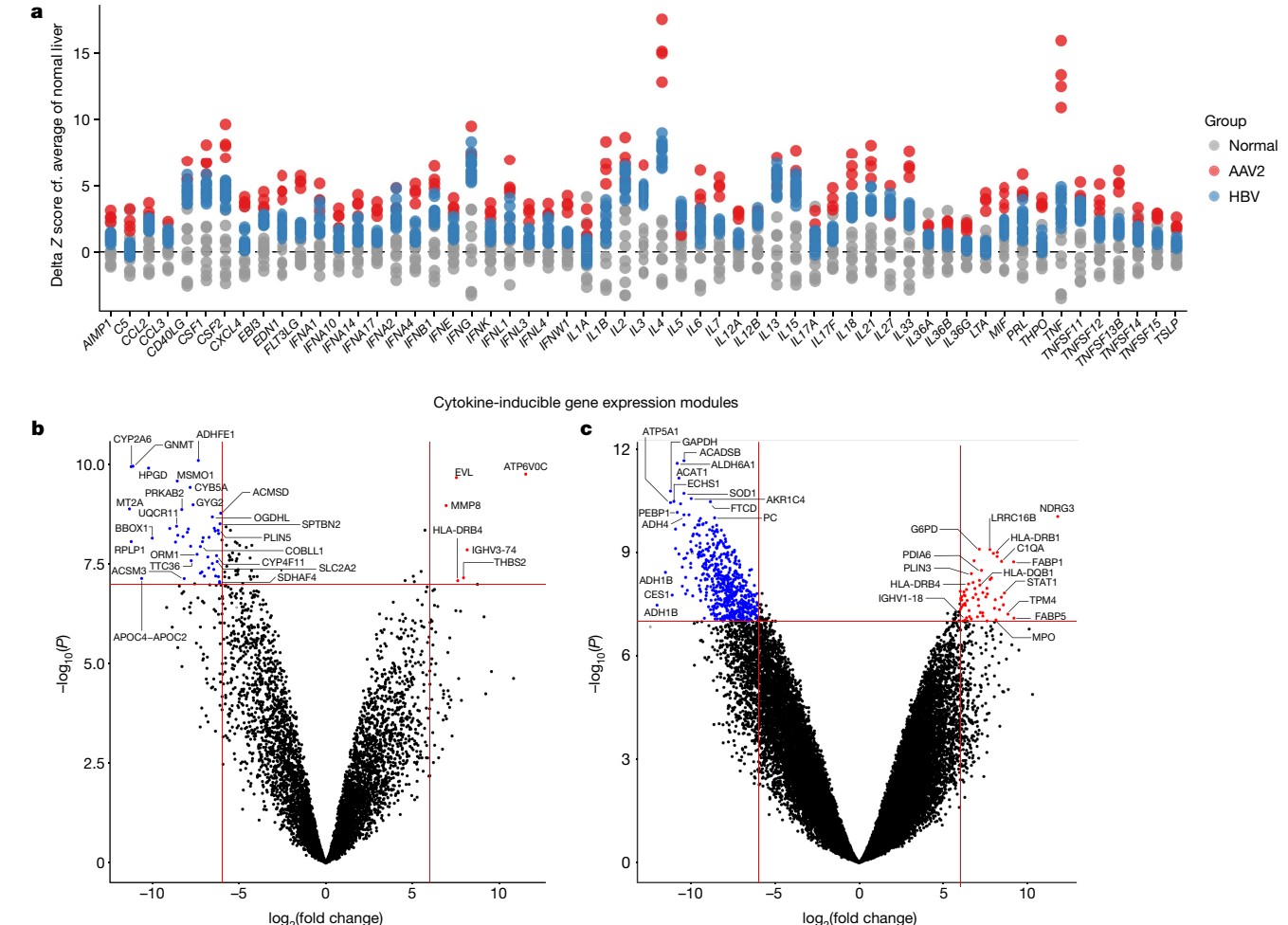

**Fig. 4 | Transcriptomic and proteomic analysis of case liver samples.**
Transcriptomic analysis was conducted for the five frozen case liver samples from transplanted patients. **a**, Expressions of cytokine-inducible transcriptional modules in normal liver, and AAV2-associated ($n = 4$) or HBV-associated ($n = 17$) hepatitis requiring transplantation are shown as delta $Z$ scores for the expression of each module, reflecting the difference from the average score from normal liver ($n = 10$) datasets, all from different patients. Each point represents the score form a single dataset or sample. **b,c**, Volcano plots of differentially expressed proteins (**b**) and peptides (**c**). The volcano plots illustrate fold changes and corresponding $P$ values for the comparison between five liver explants from five patients and seven control healthy livers from seven controls. Each dot represents a protein or peptide. The $P$ values were calculated by applying two-tailed empirical Bayes moderated $t$-statistics on protein-wise or peptide-wise linear models. Proteins (**b**) and peptides (**c**) differentially expressed (absolute $\log_2$(fold change) > 6 and $P < 1 \times 10^{-7}$) are coloured red (upregulated) and blue (downregulated). The $P$ values illustrated here are not adjusted for multiple comparisons. Full tables can be found in Supplementary Tables 12–14.

## Discussion

Despite reports implicating HAdV-F41 as causing the recent outbreak of unexplained paediatric hepatitis, we found very low levels of HAdV DNA, no proteins, inclusions or viral particles, including in explanted liver tissue from affected cases and no evidence of a change in the virus. By contrast, metagenomic and PCR analysis of liver tissue and blood identified high levels of DNA from AAV2, a member of the *Dependoparvovirus* genus, which has not been previously associated with clinical disease, in 27 of 28 cases. Replication of AAV2 requires co-infection with a helper virus, such as HAdV, herpesviruses or papillomavirus[21], and can also be triggered in the laboratory by cellular damage[22], raising the possibility that the AAV2 detected was a bystander of previous HAdV-F41 infection and/or liver damage. Against this, we found little or no AAV2 in blood from age-matched, immunocompetent children including those with HAdV infection, hepatitis or critical illness (Fig. 2d). AAV2 has been reported to establish latency in the liver[23]; however, even in critically ill immunosuppressed children with hepatitis in whom reactivation might occur, we detected AAV2 infrequently and at significantly lower levels in the blood or in liver biopsies (Fig. 2d,g).

RNA transcriptomic and real-time PCR data from explanted livers point to active AAV2 infection, although we did not detect AAV2 proteins by immunohistochemistry (Extended Data Fig. 6b) or proteomics (Extended Data Fig. 8) or any viral particles. The abundant AAV2 genomes in the explanted liver are concatenated with many complex and abnormal configurations. AAV genome concatenation may occur during AAV2 replication[8], whereas abnormal AAV2 DNA complexes and rearrangements have been observed in the liver following AAV gene therapy[7]. Hepatitis following AAV gene therapy has been well described[24–26], with deaths occurring, albeit rarely[27]. The pattern of complexes typify both HAdV and herpesvirus (including HHV-6B)-mediated AAV2 DNA replication[6]. The presence of HHV-6B DNA in 11 of 12 explanted livers, but not in livers (0 of 2) of non-transplanted children, or control livers as well as the expression, in 5 of 5 cases tested, of HHV-6B proteins, including U43, a homologue of the HSV1 helicase primase UL52, which is known to aid AAV2 replication, highlight a possible role for HHV-6B as well as HAdV in the pathogenesis of AAV2 hepatitis, particularly in severe cases. Although AAV2 is also capable of chromosomal integration[28–30], we found little evidence of this by long read sequencing, computational analysis of metagenomics data

or examination of unmapped reads, although further confirmatory studies may be required.

Although the pathogenesis of unexplained paediatric hepatitis and the role of AAV2 remain to be determined, our results point strongly to an immune-mediated process. Transcriptomic and proteomic data from the five explant livers identified significant immune dysregulation involving genes and proteins that are strongly associated with activation of B cells and T cells, neutrophils and macrophages as well as innate pathways. The findings are supported by immunohistochemical staining showing infiltration into liver tissue of CD8[+], B cell and B cell lineage cells. Upregulation of canonical pro-inflammatory cytokines including IL-15, which has also been seen in a mouse model of AAV hepatitis[31], IL-4 and TNF occurred at levels greater even than are seen in fulminant liver failure following infection with hepatitis B virus. Increased levels in the same immunoglobulin variable region peptides and corresponding proteins from both immunoglobulin heavy and light chains across all five livers point to specific antibody involvement[32]. HLA-DRB1*04:01 (12 of 13 cases tested) (Supplementary Table 1) among children in our study supports the same genetic predisposition as mooted in a parallel study conducted in Scotland[33].

An immune-mediated process is consistent with studies of hepatitis following AAV gene therapy, in which raised AAV2 IgG and capsid specific cytotoxic T lymphocytes are observed in the affected patients; however, whether these directly mediate hepatitis remains unclear[26,34]. Although we did not find that AAV2 sequences in cases differed from those in AAV2 occurring as co-infections in HAdV-F41-positive stool collected from control children during the contemporary HAdV-F41 gastroenteritis outbreak (Fig. 3b), rAAV capsid expressing a consensus capsid sequence from the unexplained hepatitis cases (AAV2Hepcase) showed reduced HSPG dependency, compared with canonical AAV2 (Extended Data Fig. 4), while retaining hepatocyte transduction ability. This points to likely greater in vivo hepatotropism of currently circulating AAV2 than has hitherto been assumed from data on canonical AAV2 (ref. 17). Another member of the parvovirus family, equine parvovirus-hepatitis, has also been associated with acute hepatitis in horses (Theiler's disease)[35].

There are several limitations to our study. Although other known infectious, autoimmune, toxic and metabolic aetiologies[3] have been excluded including by other studies[36,37], the number of cases investigated here is small, the study is retrospective, the immunocompromised controls were not perfectly age-matched, and only one immunocompetent and 17 immunocompromised controls were sampled during exactly the same period as the outbreak. Age-matched, immunocompetent controls contemporaneous with the outbreak from the DIAMONDS study, although few in number, were however found to be AAV2 negative in a separate study carried out in Scotland[33].

Finally, our data alone are not sufficient on their own to rule out a contribution from SARS-CoV-2 Omicron, the appearance of which preceded the outbreak of unexplained hepatitis (Supplementary Table 1). We did not detect SARS-CoV-2 metagenomically even in three participants who tested positive on admission. Moreover, although seropositivity was higher in our cases (15 of 20) than in controls (3 of 10), this was not the case for another UK cohort[36] (38%) or in preliminary data from a UKHSA case–control study[3], which showed similar SARS-CoV-2 antibody prevalence between unexplained hepatitis cases and population controls (less than 5 years of age: 60.5% versus 46.3%, respectively; 5–10 years of age: 66.7% versus 69.6%, respectively). In line with UK national recommendations at the time, none of the children had received a COVID vaccine.

Although we found little evidence for SARS-CoV-2 directly causing the hepatitis outbreak, we cannot exclude the effect of the COVID-19 pandemic on child mixing and infection patterns. The contemporaneous development of unexplained paediatric hepatitis with a national outbreak of HAdV-F41 (ref. 2) and the finding of HAdV-F41 in many cases suggest that the two are linked. Enteric HAdV infection is most common

in those younger than 5 years of age[2], and infection is influenced by mixing and hygiene[38]. Few cases of HAdV-F41 occurred between 2020 and 2022 and no major outbreaks were recorded[2]. The current HAdV outbreak followed relaxation of restrictions due to the pandemic and represented one of many infections, including other enteric pathogens that occurred in UK children following return to normal mixing[39]. Under normal circumstances, the levels of AAV2 antibodies are high at birth, subsequently declining to reach their lowest point at 7–11 months of age, increasing thereafter through childhood and adolescence[40]. AAV2 is known to spread with respiratory HAdVs, infections that declined during the COVID-19 pandemic, and has not been detected by us in over 30 SARS-CoV-2-positive nasopharyngeal aspirates (data not shown). We also found AAV2 DNA to be present in HAdV-F41-positive stool from both cases and controls (Supplementary Table 5). With loss of child mixing during the COVID-19 pandemic, reduced spread of common respiratory and enteric viral infections and no evidence of AAV2 in SARS-CoV-2-positive nasopharyngeal swabs, it is likely that immunity to both HAdV-F41 and AAV2 declined sharply in the age group affected by this unexplained hepatitis outbreak. Pre-existing antibody is known to reduce levels of AAV DNA in the liver of non-human primates following infusion of AAV gene therapy vectors[41]. The possibility that, in the absence of protective immunity, excessive replication of HAdV-F41 and AAV2 with accumulation of AAV2 DNA in the liver led to immune-mediated hepatic disease in genetically predisposed individuals needs further investigation. Evaluation of drugs that inhibit TNF and other cytokines massively elevated in this condition may identify important therapeutic options for future cases.

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

Sofia Morfopoulou[1,2,249], Sarah Buddle[1,249], Oscar Enrique Torres Montaguth[1], Laura Atkinson[3], José Afonso Guerra-Assunção[1], Mahdi Moradi Marjaneh[2,4], Riccardo Zennezini Chiozzi[5], Nathaniel Storey[3], Luis Campos[6], J. Ciaran Hutchinson[6], John R. Counsell[7], Gabriele Pollara[8], Sunando Roy[1], Cristina Venturini[1], Juan F. Antinao Diaz[7], Ala'a Siam[7,9], Luke J. Tappouni[7], Zeinab Asgarian[7], Joanne Ng[9], Killian S. Hanlon[7], Alexander Lennon[3], Andrew McArdle[2], Agata Czap[8], Joshua Rosenheim[8], Catarina Andrade[6], Glenn Anderson[6], Jack C. D. Lee[3], Rachel Williams[10], Charlotte A. Williams[10], Helena Tutill[10], Nadua Bayzid[10], Luz Marina Martin Bernal[10], Hannah Macpherson[11], Kylie-Ann Montgomery[10,11], Catherine Moore[12], Kate Templeton[13], Claire Neill[14], Matt Holden[15,16], Rory Gunson[17], Samantha J. Shepherd[17], Priyen Shah[2], Samantha Cooray[2], Marie Voice[18], Michael Steele[18], Colin Fink[18], Thomas E. Whittaker[19], Giorgia Santilli[19], Paul Gissen[10], Benedikt B. Kaufer[20], Jana Reich[20], Julien Andreani[21,22], Peter Simmonds[21], Dimah K. Alrabiah[10,23], Sergi Castellano[10,24], Primrose Chikowore[25], Miranda Odam[25], Tommy Rampling[8,26,27], Catherine Houlihan[8,26,28], Katja Hoschler[26], Tiina Talts[26], Cristina Celma[26], Suam Gonzalez[26], Eileen Gallagher[26], Ruth Simmons[26], Conall Watson[26], Sema Mandal[26], Maria Zambon[26], Meera Chand[26], James Hatcher[3], Surjo De[3], Kenneth Baillie[25], Malcolm Gracie Semple[29,30], DIAMONDS Consortium*, PERFORM Consortium*, ISARIC 4C Investigators*, Joanne Martin[31], Ines Ushiro-Lumb[32], Mahdad Noursadeghi[8], Maesha Deheragoda[33], Nedim Hadzic[33], Tassos Grammatikopoulos[33], Rachel Brown[34], Chayarani Kelgeri[35], Konstantinos Thalassinos[5,36,37], Simon N. Waddington[9,38], Thomas S. Jacques[6,39], Emma Thomson[40], Michael Levin[2], Julianne R. Brown[1] & Judith Breuer[1,3]✉

[1]Infection, Immunity and Inflammation Department, Great Ormond Street Institute of Child Health, University College London, London, UK. [2]Section for Paediatrics, Department of Infectious Diseases, Faculty of Medicine, Imperial College London, London, UK. [3]Department of Microbiology, Virology and Infection Control, Great Ormond Street Hospital for Children NHS Foundation Trust, London, UK. [4]Section of Virology, Department of Infectious Diseases, Faculty of Medicine, Imperial College London, London, UK. [5]University College London Mass Spectrometry Science Technology Platform, Division of Biosciences, University College London, London, UK. [6]Histopathology Department, Great Ormond Street Hospital for Children NHS Foundation Trust, London, UK. [7]Research Department of Targeted Intervention, Division of Surgery and Interventional Science, University College London, London, UK. [8]Division of Infection and Immunity, University College London, London, UK. [9]Gene Transfer Technology Group, EGA-Institute for Women's Health, University College London, London, UK. [10]Genetics and Genomic Medicine Department, Great Ormond Street Institute of Child Health, University College London, London, UK. [11]Department of Neurodegenerative Disease, Queen Square Institute of Neurology, University College London, London, UK. [12]Wales Specialist Virology Centre, Public Health Wales Microbiology Cardiff, University Hospital of Wales, Cardiff, UK. [13]Department of Medical Microbiology, Edinburgh Royal Infirmary, Edinburgh, UK. [14]Public Health Agency Northern Ireland, Belfast, UK. [15]School of Medicine, University of St. Andrews, St. Andrews, UK. [16]Public Health Scotland, Edinburgh, UK. [17]West of Scotland Specialist Virology Centre, Glasgow, UK. [18]Micropathology Ltd, University of Warwick Science Park, Coventry, UK. [19]Molecular and Cellular Immunology, Great Ormond Street Institute of Child Health, University College London, London, UK. [20]Institute of Virology, Freie Universität Berlin, Berlin, Germany. [21]Nuffield Department of Medicine, University of Oxford, Oxford, UK. [22]Centre Hospitalier Universitaire (CHU) Grenoble–Alpes, Grenoble, France. [23]National Centre for Biotechnology, King Abdulaziz City for Science and Technology, Riyadh, Saudi Arabia. [24]University College London Genomics, University College London, London, UK. [25]Roslin Institute, University of Edinburgh, Edinburgh, UK. [26]UK Health Security Agency, London, UK. [27]Hospital for Tropical Diseases, University College London Hospitals NHS Foundation Trust, London, UK. [28]Department of Clinical Virology, University College London Hospitals, London, UK. [29]Pandemic Institute, University of Liverpool, Liverpool, UK. [30]Respiratory Medicine, Alder Hey Children's Hospital NHS Foundation Trust, Liverpool, UK. [31]Centre for Genomics and Child Health, The Blizard Institute, Queen Mary University of London, London, UK. [32]NHS Blood and Transplant, Bristol, UK. [33]King's College Hospital, London, UK. [34]Department of Cellular Pathology, University Hospitals Birmingham NHS Foundation Trust, Birmingham, UK. [35]Liver Unit, Birmingham Women's and Children's NHS Foundation Trust, Birmingham, UK. [36]Institute of Structural and Molecular Biology, Division of Biosciences, University College London, London, UK. [37]Institute of Structural and Molecular Biology, Birkbeck College, University of London, London, UK. [38]Medical Research Council Antiviral Gene Therapy Research Unit, Faculty of Health Sciences, University of the Witswatersrand, Johannesburg, South Africa. [39]Developmental Biology and Cancer Department, Great Ormond Street Institute of Child Health, University College London, London, UK. [40]Medical Research Council–University of Glasgow Centre for Virus Research, Glasgow, UK. [249]These authors contributed equally: Sofia Morfopoulou, Sarah Buddle. *A list of authors and their affiliations appears online. ✉e-mail: j.breuer@ucl.ac.uk

DIAMONDS Consortium

Michael Levin[2], Evangelos Bellos[2], Claire Broderick[2], Samuel Channon-Wells[2], Samantha Cooray[2], Tisham De[2], Giselle D'Souza[2], Leire Estramiana Elorrieta[2], Diego Estrada-Rivadeneyra[2], Rachel Galassini[2], Dominic Habgood-Coote[2], Shea Hamilton[2], Heather Jackson[2], James Kavanagh[2], Mahdi Moradi Marjaneh[2], Stephanie Menikou[2], Samuel Nichols[2], Ruud Nijman[2], Harsita Patel[2], Ivana Pennisi[2], Oliver Powell[2], Ruth Reid[2], Priyen Shah[2], Ortensia Vito[2], Elizabeth Whittaker[2], Clare Wilson[2], Rebecca Womersley[2], Amina Abdulla[41], Sarah Darnell[41], Sobia Mustafa[41], Pantelis Georgiou[42], Jesus-Rodriguez Manzano[43], Nicolas Moser[42], Michael Carter[44,45], Shane Tibby[44,45], Jonathan Cohen[44], Francesca Davis[44], Julia Kenny[44], Paul Wellman[44], Marie White[44], Matthew Fish[46], Aislinn Jennings[47], Shankar-Hari[46,47], Katy Fidler[48], Dan Agranoff[49], Vivien Richmond[48,50], Matthew Seal[49], Saul Faust[51,52], Dan Owen[51,52], Ruth Ensom[51], Sarah McKay[51], Diana Mondo[53], Mariya Shaji[53], Rachel Schranz[53], Prita Rughnani[54,55,56], Amutha Anpananthar[54,55,56], Susan Liebeschuetz[55], Anna Riddell[54], Nosheen Khalid[54,56], Ivone Lancoma Malcolm[57], Teresa Simagan[56], Mark Peters[58], Alasdair Bamford[58,59], Nazima Pathan[60,61], Esther Daubney[60], Deborah White[60], Melissa Heightman[62], Sarah Eisen[62], Terry Segal[62], Lucy Wellings[62], Simon B. Drysdale[63], Nicole Branch[63], Lisa Hamzah[63], Heather Jarman[63], Maggie Nyirenda[64,65], Lisa Capozzi[64], Emma Gardiner[64], Robert Moots[66], Magda Nasher[67], Anita Hanson[67], Michelle Linforth[66], Sean O'Riordan[68], Donna Ellis[68], Akash Deep[33], Ivan Caro[33], Fiona Shackley[69], Arianna Bellini[69], Stuart Gormley[69], Samira Neshat[70], Barnaby J. Scholefield[71], Ceri Robbins[71], Helen Winmill[71], Stéphane C. Paulus[72,73,74,75], Andrew J. Pollard[72,73,74,75], Sarah Hopton[72], Danielle Miller[72], Zoe Oliver[72], Sally Beer[72], Bryony Ward[72], Shrijana Shrestha[76], Meeru Gurung[76], Puja Amatya[76], Bhishma Pokhrel[76], Sanjeev Man Bijukchhe[76], Tim Lubinda[74], Sarah Kelly[74], Peter O'Reilly[74], Federico Martinón-Torres[77,78], Antonio Salas[77,78,79,80], Fernando Álvez González[77,78,79,80], Xabier Bello[77,78,79,80], Mirian Ben García[77,78], Sandra Carnota[77,78], Miriam Cebey-López[77,78], María José Curras-Tuala[77,78,79,80], Carlos Durán Suárez[77,78], Luisa García Vicente[77,78], Alberto Gómez-Carballa[77,78,79,80], Jose Gómez Rial[77,78], Pilar Leboráns Iglesias[77,78], Nazareth Martinón-Torres[77,78], José María Martinón Sánchez[77,78], Belén Mosquera Pérez[77,78], Jacobo Pardo-Seco[77,78,79,80], Lidia Piñeiro Rodríguez[77,78], Sara Pischedda[77,78,79,80], Sara Rey Vázquez[77,78], Irene Rivero Calle[77,78], Carmen Rodríguez-Tenreiro[77,78], Lorenzo Redondo-Collazo[77,78], Miguel Sadiki Ora[77,78], Sonia Serén Fernández[77,78], Cristina Serén Trasorras[77,78], Marisol Vilas Iglesias[77,78], Enitan D. Carrol[81,82,83], Elizabeth Cocklin[81], Aakash Khanijau[81], Rebecca Lenihan[81], Nadia Lewis-Burke[81], Karen Newal[84], Sam Romaine[81], Maria Tsolia[85], Irini Eleftheriou[85], Nikos Spyridis[85], Maria Tambouratzi[85], Despoina Maritsi[85], Antonios Marmarinos[85], Marietta Xagorari[85], Lourida Panagiota[86], Pefanis Aggelos[86], Akinosoglou Karolina[87], Gogos Charalambos[87], Maragos Markos[87], Voulgarelis Michalis[88], Stergiou Ioanna[88], Marieke Emonts[89,90,91], Emma Lim[90,91,92], John Isaacs[90], Kathryn Bell[93], Stephen Crulley[93], Daniel Fabian[93], Evelyn Thomson[93], Caroline Miller[93], Ashley Bell[93], Fabian J. S. van der Velden[89,90], Geoff Shenton[94], Ashley Price[95,96], Owen Treloar[89,90], Daisy Thomas[89,90], Pablo Rojo[97,98], Cristina Epalza[97,99], Serena Villaverde[97], Sonia Márquez[99], Manuel Gijón[99], Fátima Machín[99], Laura Cabello[99], Irene Hernández[99], Lourdes Gutiérrez[99], Ángela Manzanares[97], Taco Kuijpers[100,101], Martijn van de Kuip[100], Marceline van Furth[100], Merlijn van den Berg[100], Giske Biesbroek[100], Floris Verkuil[100], Carlijn van der Zee[100], Dasja Pajkrt[100], Michael Boele van Hensbroek[100], Dieneke Schonenberg[100], Mariken Gruppen[100], Sietse Nagelkerke[100,101], Machiel H. Jansen[100], Ines Goetschalckx[101], Lorenza Romani[102], Maia De Luca[102], Sara Chiurchiù[102], Martina Di Giuseppe[102], Clementien L. Vermont[103], Henriëtte A. Moll[104], Dorine M. Borensztajn[104], Nienke N. Hagedoorn[104], Chantal Tan[104], Joany Zachariasse[104], W. Dik[105], Ching-Fen Shen[2,106], Dace Zavadska[107,108], Sniedze Laivacuma[107,108], Aleksandra Rudzate[107,108], Diana Stoldere[107,108], Arta Bardzina[107,108], Elza Bardzina[107,108], Sniedze Laivacuma[107,109], Monta Madelane[107,109], Dagne Gravele[108], Dace Svile[108], Romain Basmaci[110,111], Noémie Lachaume[110], Pauline Bories[110], Raja Ben Tkhayat[110], Laura Chériaux[110], Juraté Davoust[110], Kim-Thanh Ong[110], Marie Cotillon[110], Thibault de Groc[110], Sébastien Le[110], Nathalie Vergnault[110], Hélène Sée[110], Laure Cohen[110], Alice de Tugny[110], Nevena Danekova[110], Marine Mommert-Tripon[112], Karen Brengel-Pesce[112,113,114], Marko Pokorn[115,116,117], Mojca Kolnik[115,116], Tadej Avcin[116,117], Tanja Avramoska[115,116], Natalija Bahovec[115], Petra Bogovic[115], Lidija Kitanovski[116,117], Mirijam Nahtigal[115], Lea Papst[115], Tina Plankar Srovin[115,116], Franc Strle[115,116], Katarina Vincek[115], Michiel van der Flier[118,119], Wim J. E. Tissing[119], Roelie M. Wösten-van Asperen[120], Sebastiaan J. Vastert[121], Daniel C. Vijlbrief[122], Louis J. Bont[118,119], Tom F. W. Wolfs[118,119], Coco R. Beudeker[118,119], Philipp Agyeman[123], Luregn Schlapbach[124,125], Christoph Aebi[123], Mariama Usman[123], Stefanie Schlüchter[123], Verena Wyss[123], Nina Schöbi[123], Elisa Zimmermann[124], Kathrin Weber[124], Eric Giannoni[126,127], Martin Stocker[128], Klara M. Posfay-Barbe[129], Ulrich Heininger[130], Sara Bernhard-Stirnemann[131], Anita Niederer-Loher[132], Christian Kahlert[132], Giancarlo Natalucci[133], Christa Relly[134], Thomas Riedel[134], Christoph Berger[134], Christine Voice[18], Michael Steele[18], Colin Fink[18], Jennifer Holden[18], Leo Calvo-Bado[18], Benjamin Evans[18], Jake Stevens[18], Peter Matthews[18], Kyle Billing[18], Werner Zenz[136], Alexander Binder[136], Benno Kohlmaier[136], Daniela S. Kohlfürst[136], Nina A. Schweintzger[136], Christoph Zurl[136], Susanne Hösele[136], Manuel Leitner[136], Lena Pölz[136], Alexandra Rusu[136], Glorija Rajic[136], Bianca Stoiser[136], Martina Strempfl[136], Manfred G. Sagmeister[136], Sebastian Bauchinger[136], Martin Benesch[136,137], Astrid Ceolotto[136], Ernst Eber[138], Siegfried Gallistl[136], Harald Haidl[136], Almuthe Hauer[136], Christa Hude[136], Andreas Kapper[139], Markus Keldorfer[140], Sabine Löffler[140], Tobias Niedrist[141], Heidemarie Pilch[140], Andreas Pfleger[138], Klaus Pfurtscheller[137,142], Siegfried Rödl[137,142], Andrea Skrabl-Baumgartner[136], Volker Strenger[137], Elmar Wallner[139], Dennie Tempel[143], Danielle van Keulen[143], Annelieke M. Strijbosch[143], Maike K. Tauchert[144], Ulrich von Both[145,146], Laura Kolberg[145], Patricia Schmied[145], Irene Alba-Alejandre[147], Katharina Danhauser[148], Nikolaus Haas[149], Florian Hoffmann[150], Matthias Griese[151], Tobias Feuchtinger[152], Sabrina Juranek[153], Matthias Kappler[151], Eberhard Lurz[154], Esther Maier[153], Karl Reiter[150], Carola Schoen[150], Sebastian Schroepf[155], Shunmay Yeung[156,157,158], Manuel Dewez[156], David Bath[158], Elizabeth Fitchett[156] & Fiona Cresswell[156]

[41]Children's Clinical Research Unit, St. Mary's Hospital, London, UK. [42]Department of Electrical and Electronic Engineering, Imperial College London, London, UK. [43]Section of Adult Infectious Disease, Department of Infectious Disease, Imperial College London, London, UK. [44]Evelina London Children's Hospital, Guy's and St. Thomas' NHS Foundation Trust, London, UK. [45]Department of Women and Children's Health, School of Life Course Sciences, King's College London, London, UK. [46]Department of Infectious Diseases, School of Immunology and Microbial Sciences, King's College London, London, UK. [47]Department of Intensive Care Medicine, Guy's and St. Thomas' NHS Foundation Trust, London, UK. [48]Royal Alexandra Children's Hospital, University Hospitals Sussex, Brighton, UK. [49]Department of Infectious Diseases, University Hospitals Sussex, Brighton, UK. [50]Research Nurse Team, University Hospitals Sussex, Brighton, UK. [51]National Institute for Health Research Southampton Clinical Research Facility, University Hospital Southampton NHS Foundation Trust, Southampton, UK. [52]University of Southampton, Southampton, UK. [53]Department of Research and Development, University Hospital Southampton NHS Foundation Trust, Southampton, UK. [54]Royal London Hospital, London, UK. [55]Newham University Hospital, London, UK. [56]Whipps Cross University Hospital, London, UK. [57]Barts Health NHS Trust, London, UK. [58]Great Ormond Street Hospital NHS Foundation Trust, London, UK. [59]Great Ormond Street Institute of Child Health, University College London, London, UK. [60]Addenbrooke's Hospital, Cambridge, UK. [61]Department of Paediatrics, University of Cambridge, Cambridge, UK. [62]University College London Hospital, London, UK. [63]St George's Hospital, London, UK. [64]University Hospital Lewisham, London, UK. [65]Queen Elizabeth Hospital Greenwich, London, UK. [66]Aintree University Hospital, Liverpool, UK. [67]Royal Liverpool Hospital, Liverpool, UK. [68]Leeds Children's Hospital, Leeds, UK. [69]Sheffield Children's Hospital, Sheffield, UK. [70]Leicester General Hospital, Leicester, UK. [71]Birmingham Women's and Children's NHS Foundation Trust, Birmingham, UK. [72]John Radcliffe Hospital, Oxford University Hospitals NHS Foundation Trust, Oxford, UK. [73]Department of Paediatrics, University of Oxford, Oxford, UK. [74]Oxford Vaccine Group, Department of Paediatrics, University of Oxford, Oxford, UK. [75]National Institute for Health Research Oxford Biomedical Research Centre, Oxford, UK. [76]Paediatric Research Unit, Patan Academy of Health Sciences, Kathmandu, Nepal. [77]Translational Paediatrics and Infectious Diseases, Paediatrics Department, Hospital Clínico Universitario de Santiago, Santiago de Compostela, Spain. [78]GENVIP Research Group, Instituto de Investigación Sanitaria de Santiago, Universidad de Santiago de Compostela, Galicia, Spain. [79]Unidade de Xenética, Departamento de Anatomía Patolóxica e Ciencias Forenses, Instituto de Ciencias Forenses, Facultade de Medicina, Universidade de Santiago de Compostela, Galicia, Spain. [80]GenPop Research Group, Instituto de Investigacións Sanitarias (IDIS), Hospital Clínico Universitario de Santiago, Galicia, Spain. [81]Department of Clinical Infection, Microbiology and Immunology, University of Liverpool, Institute of Infection, Veterinary and Ecological Sciences, Liverpool, UK. [82]Department of Infectious Diseases, Alder Hey Children's Hospital, Liverpool, UK. [83]Liverpool Health Partners, Liverpool Science Park, Liverpool, UK. [84]Clinical Research Business Unit, Alder Hey Children's Hospital, Liverpool, UK. [85]Department of Paediatrics, National and Kapodistrian University of Athens (NKUA), P, and A. Kyriakou Children's Hospital, Athens, Greece. [86]Department of Infectious Diseases, Sotiria General Hospital, Athens, Greece. [87]Pathology Department, University of Patras, Panagia i Voithia General Hospital, Patras, Greece. [88]Pathophysiology Department, Medical Faculty, National and Kapodistrian University of Athens (NKUA), Laiko General Hospital, Athens, Greece. [89]Translational and Clinical Research Institute, Newcastle University, Newcastle upon Tyne, UK. [90]Paediatric Immunology, Infectious Diseases and Allergy, Great North Children's Hospital, Newcastle upon Tyne Hospitals NHS Foundation Trust, Newcastle upon Tyne, UK. [91]National Institute for Health Research Newcastle Biomedical Research Centre, Newcastle upon Tyne Hospitals NHS Foundation Trust and Newcastle University, Newcastle upon Tyne, UK. [92]Population Health Sciences Institute, Newcastle University, Newcastle upon Tyne, UK. [93]Research Unit, Great North Children's Hospital, Newcastle upon Tyne Hospitals NHS Foundation Trust, Newcastle upon Tyne, UK. [94]Paediatric Oncology, Great North Children's Hospital, Newcastle upon Tyne Hospitals NHS Foundation Trust, Newcastle upon Tyne, UK. [95]Department of Infection and Tropical Medicine, Newcastle upon Tyne Hospitals NHS Foundation Trust, Newcastle upon Tyne, UK. [96]National Institute for Health Research Newcastle In Vitro Diagnostics Co-operative (Newcastle MIC), Newcastle upon Tyne, UK. [97]Servicio Madrileño de Salud (SERMAS), Paediatric Infectious Diseases Unit, Department of Paediatrics, Hospital Universitario 12 de Octubre, Madrid, Spain. [98]Faculty of Medicine, Department of Paediatrics, Universidad Complutense de Madrid, Madrid, Spain. [99]Fundación Biomédica del Hospital Universitario 12 de Octubre (FIB-H12O), Unidad Pediátrica de Investigación y Ensayos Clínicos (UPIC), Hospital Universitario 12 de Octubre, Instituto de Investigación Sanitaria Hospital 12 de Octubre (i+12), Madrid, Spain. [100]Department of Paediatric Immunology, Rheumatology and Infectious Disease, Amsterdam University Medical Centre, University of Amsterdam, Amsterdam, The Netherlands. [101]Sanquin Research Institute, Department of Molecular Hematology, University Medical Centre, Amsterdam, The Netherlands. [102]Infectious Disease Unit, Academic Department of Paediatrics, Bambino Gesù Children's Hospital IRCCS, Rome, Italy. [103]Department of Paediatric Infectious Diseases and Immunology, Erasmus Medical Centre–Sophia Children's Hospital, Rotterdam, The Netherlands. [104]Department of General Paediatrics, Erasmus Medical Centre–Sophia Children's Hospital, Rotterdam, The Netherlands. [105]Department of Immunology, Erasmus Medical Centre, Rotterdam, The Netherlands. [106]Division of Infectious Disease, Department of Paediatrics, National Cheng Kung University, Tainan, Taiwan. [107]Riga Stradins University, Riga, Latvia. [108]Children's Clinical University Hospital, Riga, Latvia. [109]Riga East Clinical University Hospital, Riga, Latvia. [110]Service de Pédiatrie-Urgences, AP-HP, Hôpital Louis-Mourier, Colombes, France. [111]Université Paris Cité, INSERM, Paris, France. [112]Open Innovation and Partnerships, bioMérieux, Lyon, France. [113]Joint Research Unit Hospice Civils de Lyon–bioMérieux, Centre Hospitalier Lyon Sud, Lyon, France. [114]Pathophysiology of Injury-induced Immunosuppression, University of Lyon, Lyon, France. [115]Department of Infectious Diseases, University Medical Centre Ljubljana, Ljubljana, Slovenia. [116]University Children's Hospital, University Medical Centre Ljubljana, Ljubljana, Slovenia. [117]Faculty of Medicine, University of Ljubljana, Ljubljana, Slovenia. [118]Paediatric Infectious Diseases and Immunology, Wilhelmina Children's Hospital, University Medical Centre Utrecht, Utrecht, The Netherlands. [119]Princess Maxima Centre for Paediatric Oncology, Utrecht, The Netherlands. [120]Paediatric Intensive Care Unit, Wilhelmina Children's Hospital, University Medical Centre Utrecht, Utrecht, The Netherlands. [121]Paediatric Rheumatology, Wilhelmina Children's Hospital, University Medical Centre Utrecht, Utrecht, The Netherlands. [122]Paediatric Neonatal Intensive Care, Wilhelmina Children's Hospital,

University Medical Centre Utrecht, Utrecht, The Netherlands. [123]Department of Paediatrics, Inselspital, Bern University Hospital, University of Bern, Bern, Switzerland. [124]Department of Intensive Care and Neonatology, Children's Research Centre, University Children's Hospital Zurich, Zurich, Switzerland. [125]Child Health Research Centre, University of Queensland, Brisbane, Queensland, Australia. [126]Clinic of Neonatology, Department Mother-Woman-Child, Lausanne University Hospital and University of Lausanne, Lausanne, Switzerland. [127]Infectious Diseases Service, Department of Medicine, Lausanne University Hospital and University of Lausanne, Lausanne, Switzerland. [128]Department of Paediatrics, Children's Hospital Lucerne, Lucerne, Switzerland. [129]Paediatric Infectious Diseases Unit, Children's Hospital of Geneva, University Hospitals of Geneva, Geneva, Switzerland. [130]Infectious Diseases and Vaccinology, University of Basel Children's Hospital, Basel, Switzerland. [131]Children's Hospital Aarau, Aarau, Switzerland. [132]Division of Infectious Diseases and Hospital Epidemiology, Children's Hospital of Eastern Switzerland St. Gallen, St. Gallen, Switzerland. [133]Department of Neonatology, University Hospital Zurich, Zurich, Switzerland. [134]Division of Infectious Diseases and Hospital Epidemiology, Children's Research Centre, University Children's Hospital Zurich, Zurich, Switzerland. [135]Children's Hospital Chur, Chur, Switzerland. [136]Department of Paediatrics and Adolescent Medicine, Division of General Paediatrics, Medical University of Graz, Graz, Austria. [137]Department of Paediatric Hematooncology, Medical University of Graz, Graz, Austria. [138]Department of Paediatric Pulmonology, Medical University of Graz, Graz, Austria. [139]Department of Internal Medicine, State Hospital Graz II, Graz, Austria. [140]University Clinic of Paediatrics and Adolescent Medicine Graz, Medical University of Graz, Graz, Austria. [141]Clinical Institute of Medical and Chemical Laboratory Diagnostics, Medical University of Graz, Graz, Austria. [142]Paediatric Intensive Care Unit, Medical University of Graz, Graz, Austria. [143]SkylineDx, Rotterdam, The Netherlands. [144]Biobanking and BioMolecular Resources Research Infrastructure–European Research Infrastructure Consortium (BBMRI-ERIC), Graz, Austria. [145]Division of Paediatric Infectious Diseases, Hauner Children's Hospital, University Hospital, Ludwig Maximilian University Munich, Munich, Germany. [146]German Centre for Infection Research (DZIF), Partner Site Munich, Munich, Germany. [147]Department of Gynecology and Obstetrics, University Hospital, Ludwig Maximilian University Munich, Munich, Germany. [148]Division of Paediatric Rheumatology, Hauner Children's Hospital, University Hospital, Ludwig Maximilian University Munich, Munich, Germany. [149]Department of Paediatric Cardiology and Paediatric Intensive Care, Hauner Children's Hospital, University Hospital, Ludwig Maximilian University Munich, Munich, Germany. [150]Paediatric Intensive Care Unit, Hauner Children's Hospital, University Hospital, Ludwig Maximilian University Munich, Munich, Germany. [151]Division of Paediatric Pulmonology, Hauner Children's Hospital, University Hospital, Ludwig Maximilian University Munich, Munich, Germany. [152]Division of Paediatric Haematology and Oncology, Hauner Children's Hospital, University Hospital, Ludwig Maximilian University Munich, Munich, Germany. [153]Division of General Paediatrics, Hauner Children's Hospital, University Hospital, Ludwig Maximilian University Munich, Munich, Germany. [154]Division of Paediatric Gastroenterology, Hauner Children's Hospital, University Hospital, Ludwig Maximilian University Munich, Munich, Germany. [155]Neonatal Intensive Care Unit, Hauner Children's Hospital, University Hospital, Ludwig Maximilian University Munich, Munich, Germany. [156]Faculty of Infectious and Tropical Disease, London School of Hygiene and Tropical Medicine, London, UK. [157]Department of Paediatrics, St. Mary's Hospital, London, UK. [158]Faculty of Public Health and Policy, London School of Hygiene and Tropical Medicine, London, UK.

## PERFORM Consortium

Michael Levin[2], Aubrey Cunnington[2], Tisham De[2], Jethro Herberg[2], Mysini Kaforou[2], Victoria Wright[2], Lucas Baumard[2], Evangelos Bellos[2], Giselle D'Souza[2], Rachel Galassini[2], Dominic Habgood-Coote[2], Shea Hamilton[2], Clive Hoggart[2], Sara Hourmat[2], Heather Jackson[2], Ian Maconochie[2], Stephanie Menikou[2], Naomi Lin[2], Samuel Nichols[2], Ruud Nijman[2], Ivonne Pena Paz[2], Oliver Powell[2], Priyen Shah[2], Clare Wilson[2], Amina Abdulla[41], Ladan Ali[41], Sarah Darnell[41], Rikke Jorgensen[41], Sobia Mustafa[41], Salina Persand[41], Molly Stevens[42], Eunjung Kim[42], Benjamin Pierce[42], Katy Fidler[48], Julia Dudley[48], Vivien Richmond[48,50], Emma Tavliavini[48,50], Ching-Fen Shen[2,106], Ching-Chuan Liu[159], Shih-Min Wang[159], Federico Martinón-Torres[77,78], Antonio Salas[77,78,79,80], Fernando Álvez González[77,78,79,80], Cristina Balo Farto[77,78], Ruth Barral-Arca[77,78,79,80], Maria Barreiro Castro[77,78], Xabier Bello[77,78,79,80], Mirian Ben García[77,78], Sandra Carnota[77,78], Miriam Cebey-López[77,78], María José Curras-Tuala[77,78,79,80], Carlos Durán Suárez[77,78], Luisa García Vicente[77,78], Alberto Gómez-Carballa[77,78,79,80], Jose Gómez Rial[77,78], Pilar Leboráns Iglesias[77,78], Federico Martinón-Torres[77,78], Nazareth Martinón-Torres[77,78], José María Martinón Sánchez[77,78], Belén Mosquera Pérez[77,78], Jacobo Pardo-Seco[77,78,79,80], Lidia Piñeiro Rodríguez[77,78,79,80], Sara Pischedda[77,78,79,80], Sara Rey Vázquez[77,78], Irene Rivero Calle[77,78], Carmen Rodríguez-Tenreiro[77,78], Lorenzo Redondo-Collazo[77,78], Miguel Sadiki Ora[77,78], Sonia Serén Fernández[77,78], Cristina Serén Trasorras[77,78], Marisol Vilas Iglesias[77,78], Dace Zavadska[107,108], Anda Balode[107,108], Arta Barzdina[107,108], Dārta Deksne[107,108], Dagne Gravele[108], Ilze Grope[107,108], Anija Meiere[107,108], Ieva Nokalna[107,108], Jana Pavare[107,108], Zanda Pucuka[107,108], Katrina Selecka[107,108], Aleksandra Sidorova[107,108], Dace Svile[108], Urzula Nora Urbane[107,108], Effua Usuf[160], Kalifa Bojang[160], Syed M. A. Zaman[160], Fatou Secka[160], Suzanne Anderson[160], Anna Rocalsatou Sarr[160], Momodou Saidykhan[160], Saffiatou Darboe[160], Samba Ceesay[160], Umberto D'alessandro[160], Henriëtte A. Moll[104], Dorine M. Borensztajn[104], Nienke N. Hagedoorn[104], Chantal Tan[104], Clementien L. Vermont[103], Joany Zachariasse[104], W. Dik[105], Philipp Agyeman[123], Luregn J. Schlapbach[125,161,162], Christoph Aebi[123], Verena Wyss[123], Mariama Usman[123], Eric Giannoni[126,127], Martin Stocker[128], Klara M. Posfay-Barbe[129], Ulrich Heininger[130], Sara Bernhard-Stirnemann[131], Anita Niederer-Loher[132], Christian Kahlert[132], Giancarlo Natalucci[133], Christa Relly[134], Thomas Riedel[135], Christoph Berger[134], Enitan D. Carrol[81,82,83], Stéphane Paulus[81], Elizabeth Cocklin[81], Rebecca Jennings[84], Joanne Johnston[84], Simon Leigh[81], Karen Newall[84], Sam Romaine[81], Maria Tsolia[85], Irini Eleftheriou[85], Maria Tambouratzi[85], Antonis Marmarinos[85], Marietta Xagorari[85], Kelly Syggelou[85], Colin Fink[18], Marie Voice[18], Leo Calvo-Bado[18], Werner Zenz[136], Benno Kohlmaier[136], Nina A. Schweintzger[136], Manfred G. Sagmeister[136], Daniela S. Kohlfürst[136], Christoph Zurl[136], Alexander Binder[136], Susanne Hösele[136], Manuel Leitner[136], Lena Pölz[136], Glorija Rajic[136], Sebastian Bauchinger[136], Hinrich Baumgart[142], Martin Benesch[137,136], Astrid Ceolotto[136], Ernst Eber[138],

Siegfried Gallistl[136], Gunther Gores[140], Harald Haidl[136], Almuthe Hauer[136], Christa Hude[136], Markus Keldorfer[140], Larissa Krenn[137], Heidemarie Pilch[140], Andreas Pfleger[138], Klaus Pfurtscheller[137,142], Gudrun Nordberg[140], Tobias Niedrist[141], Siegfried Rödl[137,142], Andrea Skrabl-Baumgartner[136], Matthias Sperl[163], Laura Stampfer[137], Volker Strenger[137], Holger Till[164], Andreas Trobisch[140], Sabine Löffler[140], Shunmay Yeung[156,157,158], Juan Emmanuel Dewez[156], Martin Hibberd[156], David Bath[158], Alec Miners[158], Ruud Nijman[157], Catherine Wedderburn[156], Anne Meierford[156], Baptiste Leurent[165], Ronald de Groot[166], Michiel van der Flier[166,167,168], Marien I. de Jonge[168], Koen van Aerde[166,167], Wynand Alkema[166], Bryan van den Broek[166], Jolein Gloerich[166], Alain J. van Gool[166], Stefanie Henriet[166,167], Martijn Huijnen[166], Ria Philipsen[166], Esther Willems[166], G. P. J. M. Gerrits[169], M. van Leur[169], J. Heidema[170], L. de Haan[166,167], C. J. Miedema[171], C. Neeleman[166], C. C. Obihara[172], G. A. Tramper-Stranders[172,173], Andrew J. Pollard[72,73,74,75], Rama Kandasamy[74,75], Stéphane Paulus[74,75], Michael J. Carter[74,75], Daniel O'Connor[74,75], Sagida Bibi[74,75], Dominic F. Kelly[74,75], Meeru Gurung[76], Stephen Thorson[76], Imran Ansari[76], David R. Murdoch[174], Shrijana Shrestha[76], Marieke Emonts[89,90,91], Emma Lim[90,91,92], Lucille Valentine[175], Karen Allen[93], Kathryn Bell[93], Adora Chan[93], Stephen Crulley[93], Kirsty Devine[93], Daniel Fabian[93], Sharon King[93], Paul McAlinden[93], Sam McDonald[93], Anne McDonnell[90,93], Ailsa Pickering[90,93], Evelyn Thomson[93], Amanda Wood[93], Diane Wallia[93], Phil Woodsford[93], Frances Baxter[93], Ashley Bell[93], Mathew Rhodes[93], Rachel Agbeko[176], Christine Mackerness[176], Bryan Baas[90], Lieke Kloosterhuis[90], Wilma Oosthoek[90], Tasnim Arif[94], Joshua Bennet[90], Kalvin Collings[90], Ilona van der Giessen[90], Alex Martin[90], Aqeela Rashid[94], Emily Rowlands[90], Gabriella de Vries[90], Fabian van der Velden[90], Mike Martin[177], Ravi Mistry[90], Ulrich von Both[145,146], Laura Kolberg[145], Manuela Zwerenz[145], Judith Buschbeck[145], Christoph Bidlingmaier[153], Vera Binder[152], Katharina Danhauser[148], Nikolaus Haas[149], Matthias Griese[151], Tobias Feuchtinger[152], Julia Keil[150], Matthias Kappler[151], Eberhard Lurz[154], Georg Muench[155], Karl Reiter[150], Carola Schoen[150], François Mallet[112,113,114], Karen Brengel-Pesce[112,113,114], Alexandre Pachot[112], Marine Mommert[112,113], Marko Pokorn[115,116,178], Mojca Kolnik[115,116], Katarina Vincek[115], Tina Plankar Srovin[115,116], Natalija Bahovec[115], Petra Prunk[115], Veronika Osterman[115], Tanja Avramoska[115,116], Taco Kuijpers[100,179], Ilse Jongerius[179], J. M. van den Berg[100], D. Schonenberg[100], A. M. Barendregt[100], D. Pajkrt[100], M. van der Kuip[100,180], A. M. van Furth[100,180], Evelien Sprenkeler[179], Judith Zandstra[179], G. van Mierlo[179] & J. Geissler[179]

[159]Centre of Clinical Medicine Research, National Cheng Kung University, Tainan, Taiwan. [160]Medical Research Council Unit The Gambia at the London School for Hygiene and Tropical Medicine, Fajara, The Gambia. [161]Neonatal and Paediatric Intensive Care Unit, Children's Research Centre, University Children's Hospital Zurich, University of Zurich, Zurich, Switzerland. [162]Queensland Children's Hospital, Brisbane, Queensland, Australia. [163]Department of Paediatric Orthopedics, Medical University of Graz, Graz, Austria. [164]Department of Paediatric and Adolescence Surgery, Medical University of Graz, Graz, Austria. [165]Faculty of Epidemiology and Population Health, London School of Hygiene and Tropical Medicine, London, UK. [166]Radboud University Medical Centre, Nijmegen, The Netherlands. [167]Amalia Children's Hospital, Nijmegen, The Netherlands. [168]Wilhelmina Children's Hospital, University Medical Centre Utrecht, Utrecht, The Netherlands. [169]Canisius Wilhelmina Hospital, Nijmegen, The Netherlands. [170]St. Antonius Hospital, Nieuwegein, The Netherlands. [171]Catharina Hospital, Eindhoven, The Netherlands. [172]ETZ Elisabeth, Tilburg, The Netherlands. [173]Franciscus Gasthuis, Rotterdam, The Netherlands. [174]Department of Pathology, University of Otago, Christchurch, New Zealand. [175]Newcastle University Business School, Centre for Knowledge, Innovation, Technology and Enterprise (KITE), Newcastle upon Tyne, UK. [176]Paediatric Intensive Care Unit, Great North Children's Hospital, Newcastle upon Tyne Hospitals NHS Foundation Trust, Newcastle upon Tyne, UK. [177]Northumbria University, Newcastle upon Tyne, UK. [178]Department of Infectious Diseases and Epidemiology, Faculty of Medicine, University of Ljubljana, Ljubljana, Slovenia. [179]Sanquin Research Institute, Landsteiner Laboratory at the AMC, University of Amsterdam, Amsterdam, The Netherlands. [180]Department of Paediatric Infectious Diseases and Immunology, Amsterdam University Medical Centre, Free University (VU) Amsterdam, Amsterdam, The Netherlands.

## ISARIC 4C Investigators

Kenneth Baillie[25], Malcolm Gracie Semple[29,30], Gail Carson[181], Peter J. M. Openshaw[182,183], Jake Dunning[182,184], Laura Merson[181], Clark D. Russell[185], David Dorward[186], Maria Zambon[26], Meera Chand[26], Richard S. Tedder[187,188,189], Say Khoo[90], Lance C. W. Turtle[191,192], Tom Solomon[191,193], Samreen Ijaz[194], Tom Fletcher[195], Massimo Palmarini[40], Antonia Y. W. Ho[40], Emma Thomson[40], Nicholas Price[196,197], Judith Breuer[1,3], Thushan de Silva[198], Chloe Donohue[199], Hayley Hardwick[191], Wilma Oosthuyzen[26], Miranda Odam[25], Primrose Chikowore[25], Lauren Obosi[26], Sara Clohisey[26], Andrew Law[26], Lucy Norris[200], Sarah Tait[16], Murray Wham[201], Richard Clark[202], Audrey Coutts[202], Lorna Donelly[202], Angie Fawkes[202], Tammy Gilchrist[202], Katarzyna Hafezi[202], Louise MacGillivray[202], Alan Maclean[202], Sarah McCafferty[202], Kirstie Morrice[202], Lee Murphy[202], Nicola Wrobel[202], Sarah E. McDonald[39,203], Victoria Shaw[204], Jane A. Armstrong[205], Lauren Lett[206], Paul Henderson[207], Louisa Pollock[208], Shyla Kishore[209], Helen Brotherton[210,211], Lawrence Armstrong[212,213], Andrew Mita[214], Anna Dall[215], Kristyna Bohmova[216], Sheena Logan[216], Louise Gannon[217], Ken Agwuh[218], Srikanth Chukkambotla[219], Ingrid DuRand[220], Duncan Fullerton[221], Sanjeev Garg[222], Clive Graham[223], Tassos Grammatikopoulos[33], Stuart Hartshorn[71], Luke Hodgson[224], Paul Jennings[225], George Koshy[226], Tamas Leiner[226], James Limb[227], Jeff Little[228], Elijah Matovu[221], Fiona McGill[229], Craig Morris[230], John Morrice[210,211], David Price[231], Henrik Reschreiter[232], Tim Reynolds[230], Paul Whittaker[233], Rachel Tayler[234], Clare Irving[235], Maxine Ramsay[207], Margaret Millar[207], Barry Milligan[236], Naomy Hickey[236], Maggie Connon[209], Catriona Ward[209], Laura Beveridge[210], Susan MacFarlane[237], Karen Leitch[238], Claire Bell[212], Lauren Finlayson[215], Joy Dawson[215], Janie Candlish[214], Laura McGenily[216], Tara Roome[71], Cynthia Diaba[239], Jasmine Player[240], Natassia Powell[33], Ruth Howman[71], Sara Burling[71], Sharon Floyd[224], Sarah Farmer[218], Susie Ferguson[241], Susan Hope[242], Lucy Rubick[232], Rachel Swingler[243], Emma Collins[244], Collette Spencer[229], Amaryl Jones[221], Barbara Wilson[245], Diane Armstrong[246], Mark Birt[247], Holly Dickinson[230], Rosemary Harper[246], Darran Martin[248], Amy Roff[232] & Sarah Mills[232]

[181]ISARIC Global Support Centre, Centre for Tropical Medicine and Global Health, Nuffield Department of Medicine, University of Oxford, Oxford, UK. [182]National Heart and Lung Institute, Imperial College London, London, UK. [183]Imperial College Healthcare NHS Foundation Trust, London, UK. [184]National Infection Service, Public Health England, London, UK. [185]Centre for Inflammation Research, The Queen's Medical Research Institute, University of Edinburgh, Edinburgh, UK. [186]Edinburgh Pathology, University of Edinburgh, Edinburgh, UK. [187]Blood Borne Virus Unit, Virus Reference Department, National Infection Service, Public Health England, London, UK. [188]Transfusion Microbiology, National Health Service Blood and Transplant, London, UK. [189]Department of Medicine, Imperial College London, London, UK. [190]Department of Pharmacology, University of Liverpool, Liverpool, UK. [191]National Institute for Health Research Health Protection Research Unit, Institute of Infection, Veterinary and Ecological Sciences, Faculty of Health and Life Sciences, University of Liverpool, Liverpool, UK. [192]Tropical and Infectious Disease Unit, Royal Liverpool University Hospital, Liverpool, UK. [193]Walton Centre NHS Foundation Trust, Liverpool, UK. [194]Virology Reference Department, National Infection Service, Public Health England, London, UK. [195]Liverpool School of Tropical Medicine, Liverpool, UK. [196]Centre for Clinical Infection and Diagnostics Research, Department of Infectious Diseases, School of Immunology and Microbial Sciences, King's College London, London, UK. [197]Department of Infectious Diseases, Guy's and St. Thomas' NHS Foundation Trust, London, UK. [198]The Florey Institute for Host–Pathogen Interactions, Department of Infection, Immunity and Cardiovascular Disease, University of Sheffield, Sheffield, UK. [199]Liverpool Clinical Trials Centre, University of Liverpool, Liverpool, UK. [200]Edinburgh Parallel Computing Centre (EPCC), University of Edinburgh, Edinburgh, UK. [201]Medical Research Council Human Genetics Unit, Medical Research Council Institute of Genetics and Molecular Medicine, University of Edinburgh, Edinburgh, UK. [202]Edinburgh Clinical Research Facility, University of Edinburgh, Edinburgh, UK. [203]Department of Histopathology, Great Ormond Street Hospital for Children NHS Foundation Trust, London, UK. [204]Institute of Translational Medicine, University of Liverpool, Liverpool, UK. [205]Sheffield Teaching Hospitals NHS Foundation Trust, Sheffield, UK. [206]University of Liverpool, Liverpool, UK. [207]Royal Hospital For Children and Young People, Edinburgh, UK. [208]Department of Paediatric Infectious Diseases and Immunology, Royal Hospital for Children Glasgow, Glasgow, UK. [209]Royal Aberdeen Children's Hospital, Aberdeen, UK. [210]Queen Margaret Hospital, Dumfermline, UK. [211]Victoria Hospital, Kirkcaldy, UK. [212]University Hospital Crosshouse, Crosshouse, UK. [213]University Hospital Ayr, Ayr, UK. [214]Dumfries and Galloway Royal Infirmary, Dumfries, UK. [215]Borders General Hospital, Melrose, UK. [216]Forth Valley Hospital, Larbert, UK. [217]Tayside Children's Hospital and Ninewells Hospital, NHS Tayside, Dundee, UK. [218]Doncaster and Bassetlaw NHS Foundation Trust, Doncaster, UK. [219]Burnley General Hospital, Burnley, UK. [220]Hereford County Hospital, Hereford, UK. [221]Leighton Hospital, Leighton, UK. [222]Walsall Healthcare NHS Foundation Trust, Walsall, UK. [223]Cumberland Infirmary, Cumberland, UK. [224]St. Richard's Hospital, Chichester, UK. [225]Airedale Hospital, Keighley, UK. [226]Hinchingbrooke Hospital, Huntingdon, UK. [227]Darlington Memorial Hospital, Darlington, UK. [228]Warrington Hospital, Warrington, UK. [229]Leeds Teaching Hospitals NHS Trust, Leeds, UK. [230]Queen's Hospital Burton, Burton, UK. [231]Royal Victoria Infirmary, Newcastle upon Tyne, UK. [232]University Hospitals Dorset NHS Foundation Trust, Dorset, UK. [233]Bradford Royal Infirmary, Bradford, UK. [234]Department of Paediatric Gastroenterology, Heptalogy and Nutrition, Royal Hospital for Children Glasgow, Glasgow, UK. [235]Avon and Wiltshire Mental Health Partnership NHS Foundation Trust, Bath, UK. [236]Queen Elizabeth University Hospital, Glasgow, UK. [237]Tayside Children's Hospital, Dundee, UK. [238]University Hospital Wishaw, Wishaw, UK. [239]Royal Free Hospital, London, UK. [240]Diana Princess of Wales Hospital, Grimsby, UK. [241]Weston General Hospital, Weston-super-Mare, UK. [242]Barnsley Hospital, Barnsley, UK. [243]Bradford Teaching Hospitals NHS Foundation Trust, Bradford, UK. [244]Wye Valley NHS Foundation Trust, Hereford, UK. [245]Newcastle upon Tyne Hospitals, Newcastle upon Tyne, UK. [246]West Cumberland Hospital, Whitehaven, UK. [247]University of North Durham, Durham, UK. [248]Worthing Hospital, Worthing, UK.

## Methods

### Ethics

Metagenomic analysis and HAdV sequencing were carried out by the routine diagnostic service at Great Ormond Street Hospital (GOSH). Additional PCRs, immunohistochemistry and proteomics on samples received for metagenomics are part of the GOSH protocol for confirmation of new and unexpected pathogens. The use for research of anonymized laboratory request data, diagnostic results and residual material from any specimen received in the GOSH diagnostic laboratory, including all cases received from Birmingham's Children Hospital UKHSA, Public Health Wales, Public Health Scotland as well as non-case samples from UKHSA, Public Health Scotland and GOSH research was approved by UCL Partners Pathogen Biobank under ethical approval granted by the NRES Committee London-Fulham (REC reference: 17/LO/1530).

Children undergoing liver transplant were consented for additional research under the International Severe Acute Respiratory and Emerging Infection Consortium (ISARIC) WHO Clinical Characterisation Protocol UK (CCP-UK) (ISRCTN 66726260) (RQ3001-0591, RQ301-0594, RQ301-0596, RQ301-0597 and RQ301-0598). Ethical approval for the ISARIC CCP-UK study was given by the South Central–Oxford Research Ethics Committee in England (13/SC/0149), the Scotland A Research Ethics Committee (20/SS/0028) and the WHO Ethics Review Committee (RPC571 and RPC572).

The UKHSA has legal permission, provided by regulation 3 of The Health Service (Control of Patient Information) Regulations 2002, to process patient confidential information for national surveillance of communicable diseases and, as such, individual patient consent is not required.

Control participants from the EU Horizon 2020 research and innovation program DIAMONDS–PERFORM (grant agreement nos. 668303 and 848196) were recruited according to the approved enrolment procedures of each study, and with the informed consent of parents or guardians: DIAMONDS (London-Dulwich Research Ethics Committee: 20/HRA/1714) and PERFORM (London-Central Research Ethics Committee: 16/LO/1684).

The sample IDs for the cases and controls are anonymized IDs that cannot reveal the identity of the study participants and are not known to anyone outside the research group, such as the patients or the hospital staff.

### Samples

Initial diagnostic testing by metagenomics and PCR was performed at GOSH Microbiology and Virology clinical laboratories. Further WGS and characterization were performed at UCL.

### Cases

Birmingham Children's Hospital provided us with explanted liver tissue from five biopsy sites from five cases, five whole blood 500 μl from four cases and serum plasma from one case (Table 1 and Fig. 1b). These were used in metagenomics testing (Table 2), followed by HAdV, HHV-6 and AAV2 testing by PCR and, depending on the Ct value, WGS (Supplementary Tables 7, 9 and 10). We subsequently received 25 additional specimens from UKHSA, Public Health Wales and Public Health Scotland/Edinburgh Royal Infirmary, including 16 additional blood samples, four respiratory specimens and five stool samples, for HAdV WGS and, depending on residual material, for AAV2 PCR testing followed by sequencing (Tables 1 and 2, Fig. 1b and Supplementary Tables 7, 9 and 10). We also received ten FFPE liver biopsy samples and six serum samples from 11 cases from King's College Hospital (Table 1). Of these cases, seven had received liver transplants.

### Controls from DIAMONDS and PERFORM

PERFORM recruited children from ten EU countries (2016–2020). PERFORM was funded by the European Union's Horizon 2020 programme under GA no. 668303.

DIAMONDS is funded by the European Union Horizon 2020 programme grant number 848196. Recruitment commenced in 2020 and is ongoing. Both studies recruited children presenting with suspected infection or inflammation and assigned them to diagnostic groups according to a standardized algorithm.

### Controls from GOSH for PCR

Blood samples from 17 patients not linked to the non-A–E hepatitis outbreak were tested by real-time PCR targeting AAV2 (Extended Data Table 2b). These comparators were patients with ALT/AST of more than 500 and HAdV or cytomegalovirus viraemia. These were purified DNA from residual diagnostic specimens received in the GOSH microbiology and virology laboratory in the previous year. All residual specimens were stored at −80 °C before testing and pseudo-anonymized at the point of processing and analysis. Viraemia was initially detected using targeted real-time PCR during routine diagnostic testing with UKAS-accredited laboratory-developed assays that conform to ISO:15189 standards.

In addition to the blood samples, four residual liver biopsies from four control patients referred for investigation of infection were tested by AAV2 and HHV-6B PCR. The liver biopsies were submitted to the GOSH microbiology laboratory for routine diagnosis by bacterial broad-range 16S rRNA gene PCR or metagenomics testing in 2021 and 2022. Three of four control patients were known to have elevated levels of liver enzymes. Two adult frozen liver samples previously tested by metagenomics were negative for AAV2 and positive for HHV-6B (Supplementary Table 5).

### Controls from UKHSA

We received a blood sample from one patient with elevated levels of liver enzymes and HAdV infection. We also received one control stool sample from Public Health Scotland/Edinburgh Royal Infirmary and 22 control stool samples for sequencing.

### Controls from King's College Hospital

A single FFPE liver biopsy control of normal marginal tissue from a hepatoblastoma from a child was negative for AAV2 and HAdV, but positive for HHV-6B (Ct = 37).

### Controls from Queen Mary University of London

We received FFPE liver control samples from ten adults and three children (under 18 years of age) with other viral hepatitis, toxic liver necrosis, autoimmune and other hepatitis, and normal liver, from Queen Mary University of London. PCR gave valid results for samples from two children and eight adults, all of which were negative by PCR for AAV2 and HHV-6, apart from one adult sample, which was positive for HHV-6 at a high Ct value (Supplementary Table 5).

### Metagenomic sequencing

**Nucleic acid purification.** Frozen liver biopsies were infused overnight at −20 °C with RNAlater-ICE. Up to 20 mg biopsy was lysed with 1.4-mm ceramic, 0.1-mm silica and 4-mm glass beads, before DNA and RNA purification using the Qiagen AllPrep DNA/RNA Mini kit as per the manufacturer's instructions, with a 30 μl elution volume for RNA and 50 μl for DNA.

Up to 400 μl whole blood was lysed with 0.5-mm and 0.1-mm glass beads before DNA and RNA purification on a Qiagen EZ1 instrument with an EZ1 virus mini kit as per the manufacturer's instructions, with a 60 μl elution volume.

For quality assurance, every batch of samples was accompanied by a control sample containing feline calicivirus RNA and cowpox DNA, which was processed alongside clinical specimens, from nucleic acid purification through to sequencing. All specimens and controls were spiked with MS2 phage RNA internal control before nucleic acid purification.

**Library preparation and sequencing.** RNA from whole-blood samples with an RNA yield of more than 2.5 ng μl$^{-1}$ and from biopsies underwent ribosomal RNA depletion and library preparation with KAPA RNA HyperPrep kit with RiboErase, according to the manufacturer's instructions. RNA from whole blood with an RNA yield of less than 2.5 ng μl$^{-1}$ did not undergo rRNA depletion before library preparation.

DNA from whole-blood samples with a DNA yield of more than 1 ng μl$^{-1}$ and from biopsies underwent depletion of CpG-methylated DNA using the NEBNext Microbiome DNA Enrichment Kit, followed by library preparation with the NEBNext Ultra II FS DNA Library Prep Kit for Illumina, according to manufacturer's instructions. DNA from whole blood with a DNA yield of less than 1 ng μl$^{-1}$ did not undergo depletion of CpG-methylated DNA before library preparation.

Sequencing was performed with a NextSeq High output 150 cycle kit with a maximum of 12 libraries pooled per run, including controls.

## Metagenomics data analysis

**Pre-processing pipeline.** An initial quality control step was performed by trimming adapters and low-quality ends from the reads (Trim Galore! [42] 0.3.7). Human sequences were then removed using the human reference GRCH38 p.9 (Bowtie2 (ref. 43), version 2.4.1) followed by removal of low-quality and low-complexity sequences (PrinSeq [44], version 0.20.3). An additional step of human sequences removal followed (megaBLAST [45], version 2.9.0). For RNA-seq, rRNA sequences were also removed using a similar two-step approach (Bowtie2 and megaBLAST). Finally, nucleotide similarity and protein similarity searches were performed (megaBLAST and DIAMOND [46] (version 0.9.30), respectively) against custom reference databases that consisted of nucleotide and protein sequences of the RefSeq collections (downloaded March 2020) for viruses, bacteria, fungi, parasites and human.

**Taxonomic classification.** DNA and RNA sequence data were analysed with metaMix [5] (version 0.4) nucleotide and protein analysis pipelines.

metaMix resolves metagenomics mixtures using Bayesian mixture models and a parallel Markov chain Monte Carlo search of the potential species space to infer the most likely species profile.

metaMix considers all reads simultaneously to infer relative abundances and probabilistically assign the reads to the species most likely to be present. It uses an 'unknown' category to capture the fact that some reads cannot be assigned to any species. The resulting metagenomic profile includes posterior probabilities of species presence as well as Bayes factor for presence versus absence of specific species. There are two modes: metaMix-protein, which is optimal for RNA virus detection, and metaMix-nucl, which is best for speciation of DNA microorganisms. Both modes were used for RNA-seq, whereas metaMix-nucl was used for DNA-seq.

For sequence results to be valid, MS2 phage RNA had to be detected in every sample and feline calicivirus RNA and cowpox DNA, with no additional unexpected organisms, detected in the controls.

**Confirmatory mapping of AAV2.** The RNA-seq reads were mapped to the AAV2 reference genome (NCBI reference sequence NC_001401) using Bowtie2, with the −very-sensitive option. Samtools [47] (version 1.9) and Picard (version 2.26.9; http://broadinstitute.github.io/picard/) were used to sort, deduplicate and index the alignments, and to create a depth file, which was plotted using a custom script in R.

**De novo assembly of unclassified reads.** We performed a de novo assembly step with metaSPADES [48] (v3.15.5), using all the reads with no matches to the nucleotide database that we used for our similarity search. A search using megaBLAST with the standard nucleotide collection was carried out on all resulting contigs over 1,000 bp in length. All of the contigs longer than 1,000 bp matched to human, except two that mapped to Torque Teno virus.

**Nanopore sequencing.** DNA from up to 20 mg of liver was purified using the Qiagen DNeasy Blood & Tissue kit as per the manufacturer's instructions. Samples with limited amount of DNA were fragmented to an average size of 10 kb using a Megaruptor 3 (Diagenode) to reach an optimal molar concentration for library preparation. Quality control was perform using a Femto Pulse System (Agilent Technologies) and a Qubit fluorometer (Invitrogen). Samples were prepared for Nanopore sequencing using the ligation sequencing kit SQK-LSK110. DNA was sequenced on a PromethION using R9.4.1 flowcells (Oxford Nanopore Technologies). Samples were run for 72 h including a washing and re-load step after 24 h and 48 h.

All library preparation and sequencing were performed by the UCL Long Read Sequencing facility.

Passed reads from Minknow were mapped to the reference AAV2 genome (NC_001401) using minimap2 (ref. 49) using the default parameters. Reads were trimmed of adapters using Porechop v0.2.4 (https://github.com/rrwick/Porechop/), with the sequences of the adapters used added to adapters.py, and using an adapter threshold of 85. Reads that also mapped by minimap to the human genome (Ensemble GRCh38_v107), which could be ligation artefacts, were excluded from further analysis. The passed reads were also classified using Kraken2 (ref. 50) with the PlusPF database (17 May 2021). The data relating to AAV2 reads in Supplementary Table 3 refer to reads that were classified as AAV2 by both minimap2 and Kraken2 (version 2.0.8-beta), as the results from both methods were similar. Four reads across all four lower-depth samples were classified as HHV-6B by the EPI2ME WIMP [51] pipeline. No reads were classified as HAdV or HHV-6B by Kraken2 in the two higher-depth samples. Alignment dot plots were created for the AAV2 reads using redotable (version 1.1) [52], with a window size of 20. These were manually classified into possible complex and monomeric structures.

**Integration analysis of Illumina data.** We investigated potential integrations of AAV2 and HHV-6 viruses into the genome using the Illumina metagenomics data for five liver transplant cases. We first processed the pair-end reads (average sequence coverage per genome = 5×), quality checking using FastQC [53], with barcode and adaptor sequence trimmed by TrimGalore (phred-score = 20). Potential viral integrations were investigated with Vseq-Toolkit [54] (mode 3 with default settings except for high stringency levels). Predicted genomic integrations were visualized with IGV [55], requiring at least three reads supporting an integration site, spanning both human and viral sequences. Predicted integrations were supported by only one read, thus not fulfilling the algorithm criteria. Sequencing was performed at a lower depth than optimal for integration analysis, but no evidence was found for AAV2 or HHV-6B integration into the genomes of cases.

**PCR.** Real-time PCR targeting a 62-nt region of the AAV2 inverted terminal repeat sequence was performed using primers and probes previously described [56]. This assay has been predicted to amplify AAV2 and AAV6. The Qiagen QuantiNova probe PCR kit (PERFORM and DIAMONDS controls) or the Qiagen Quantifast probe PCR kit (all other samples) were used. Each 25-μl reaction consisted of 0.1 μM forward primer, 0.34 μM reverse primer and 0.1 μM probe with 5 μl template DNA.

Real-time PCR targeting a 74-bp region of the HHV-6 DNA polymerase gene was performed using primers and probes previously described [57] multiplexed with an internal positive control targeting mouse (*mus*) DNA spiked into each sample during DNA purification, as previously described [58]. In brief, each 25-μl reaction consisted of 0.5 μM of each primer, 0.3 μM HHV-6 probe, 0.12 μM of each *mus* primer, 0.08 μM *mus* probe and 12.5 μl Qiagen Quantifast Fast mastermix with 10 μl template DNA.

Real-time PCR targeting a 132-bp region of the HAdV hexon gene was performed using primers and probes previously described[59] multiplexed with an internal positive control targeting mouse (*mus*) DNA spiked into each sample during DNA purification, as previously described[58]. In brief, each 25-µl reaction consisted of 0.6 µM of each HHV-6 primer, 0.4 µM HHV-6 probe, 0.12 µM of each *mus* primer, 0.08 µM *mus* probe and 12.5 µl Qiagen Quantifast Fast mastermix with 10 µl template DNA.

PCR cycling for all targets, apart from the controls from the PER-FORM and DIAMONDS studies, was performed on an ABI 7500 Fast thermocycler and consisted of 95 °C for 5 min followed by 45 cycles of 95 °C for 30 s and 60 °C for 30 s. For the PERFORM and DIAMONDS controls, PCR was performed on a StepOnePlus Real-Time PCR System and consisted of 95 °C for 2 min followed by 45 cycles of 95 °C for 5 s and 60 °C for 10 s. Each PCR run included a no template control and a DNA-positive control for each target.

Neat DNA extracts of the FFPE material were inhibitory to PCR, so PCR results shown were performed following a 1 in 10 dilution.

**AAV2 quantitative PCR with reverse transcription.** RNA samples were treated with the Turbo-DNA free kit (Thermo) to remove residual genomic DNA. Complementary DNA (cDNA) was synthesized using the QuantiTect Reverse Transcription kit. In brief, 12 µl of RNA was mixed with 2 µl of genomic DNA Wipeout buffer and incubated at 42 °C for 2 min and transferred to ice. For reverse transcription, 6 µl mastermix was used and incubated at 42 °C for 20 min followed by 3 min at 95 °C.

Real-time PCR targeting a 120-nt region of the AAV2 cap open reading frame sequence was performed using primers AAV2_cap_Fw-ATCCTTCG ACCACCTTCAGT, AAV2_cap_Rv-GATT CCAGCGTTTGCTGTT and the probe AAV2_cap_Pr FAM-ACACAGTAT/ZEN/TCC ACGG GACAGGT-IBFQ. This assay has been predicted to amplify AAV2 and AAV6. The Qiagen QuantiNova probe PCR kit was used. Each 25-µl reaction consisted of 0.1 µM forward primer, 0.1 µM reverse primer and 0.2 µM probe with 2.5 µl template cDNA.

PCR was performed on a StepOnePlus Real-Time PCR System and consisted of incubation at 95 °C for 2 min followed by 45 cycles of 95 °C for 5 s and 60 °C for 10 s. Each PCR run included a no template control, a DNA-positive control and a RNA control from each sample to verify efficient removal of genomic DNA.

**Immunohistochemistry.** All immunohistochemistry was done on FFPE tissue cut at a thickness of 3 µm.

**Adenovirus.** AdV immunohistochemistry was carried out using the Ventana Benchmark ULTRA, Optiview Detection Kit, PIER with protease 1 for 4 min and antibody incubation for 32 min (AdV clone 2/6 and 20/11, Roche, 760-4870, pre-diluted). The positive control was a known HAdV-positive gastrointestinal surgical case.

**Preparation of AAV2-positive controls.** The plasmid used for transfection was pAAV2/2 (addgene, plasmid #104963; https://www.addgene.org/104963/), which expresses the genes encoding Rep/Cap of AAV2. This was delivered by tail-vein hydrodynamic injection[60] into albino C57BL/6 mice (5 mg in 2 ml PBS). Negative controls received PBS alone. At 48 h, mice were terminally exsanguinated and perfused by PBS. Livers were collected into 10% neutral buffered formalin (CellPath UK). This was performed under Home Office License PAD4E6357.

AAV2 immunohistochemistry was carried out with four commercially available antibodies:

• Leica Bond-III, Bond Polymer Refine Detection Kit with DAB Enhancer, HIER with Bond Epitope Retrieval Solution 1 (citrate based pH 6) for 30 min and antibody incubation for 30 min (anti-AAV VP1/VP2/VP3 clone B1, PROGEN, 690058S, 1:100).
• Leica Bond-III, Bond Polymer Refine Detection Kit with DAB Enhancer, HIER with Bond Epitope Retrieval Solution 1 (citrate based pH 6) for

40 min and antibody incubation for 30 min (anti-AAV VP1/VP2/VP3 rabbit polyclonal, OriGene, BP5024, 1:100).
• Leica Bond-III, Bond Polymer Refine Detection Kit with DAB Enhancer, HIER with Bond Epitope Retrieval Solution 1 (citrate based pH 6) for 40 min and antibody incubation for 30 min (anti-AAV VP1 clone A1, OriGene, BM5013, 1:100).
• Leica Bond-III, Bond Polymer Refine Detection Kit with DAB Enhancer, HIER with Bond Epitope Retrieval Solution 1 (citrate based pH 6) for 40 min and antibody incubation for 30 min (anti-AAV VP1/VP2 clone A69, OriGene, BM5014, 1:100). HHV-6 immunohistochemistry strain-ing was carried out with:
• Leica Bond-III, Bond Polymer Refine Detection Kit with DAB Enhancer, PIER with Bond Enzyme 1 Kit for 10 min and antibody incubation for 30 min (mouse monoclonal antibody (C3108-103) to HHV-6, ABCAM, ab128404, 1:100).

Negative reagent control slides were stained using the same antigen retrieval conditions and staining protocol incubation times using only BondTM Primary Antibody Diluent #AR9352 for the antibody incubation.

**Electron microscopy.** Samples of liver were fixed in 2.5% glutaral-dehyde in 0.1 M cacodylate buffer followed by secondary fixation in 1.0% osmium tetroxide. Tissues were dehydrated in graded ethanol, transferred to an intermediate reagent, propylene oxide and then in-filtrated and embedded in Agar 100 epoxy resin. Polymerization was undertaken at 60 °C for 48 h. Ultrathin sections of 90 nm were cut using a Diatome diamond knife on a Leica UC7 ultramicrotome. Sections were transferred to copper grids and stained with alcoholic urynal acetate and Reynold's lead citrate. The samples were examined using a JEOL 1400 transmission electron microscope. Images were captured on an AMT XR80 digital camera.

## WGS

**Bait design.** To produce the capture probes for hybridization, bioti-nylated RNA oligonucleotides (baits) used in the SureSelectXT proto-cols for HAdV and HHV-6 WGS were designed in-house using Agilent community design baits with part numbers 5191-6711 and 5191-6713, respectively. They were synthesized by Agilent Technologies (2021) (available through Agilent's Community Designs programme: SSXT CD Pan Adenovirus and SSXT CD Pan HHV-6 and used previously[61,62]).

**Library preparation and sequencing.** For WGS of HAdV and HHV-6B, DNA (bulked with male human genomic DNA (Promega) if required) was sheared using a Covaris E220 focused ultrasonication system (PIP 75, duty factor of 10, 1,000 cycles per burst). End-repair, non-templated addition of 3′ poly A, adapter ligation, hybridization, PCR (pre-capture cycles dependent on DNA input and post-capture cycles dependent on viral load) and all post-reaction clean-up steps were performed according to either the SureSelectXT Low Input Target Enrichment for Illumina Paired-End Multiplexed Sequencing protocol (version A0), the SureSelectXT Target Enrichment for Illumina Paired-End Multi-plexed Sequencing protocol (version C3) or the SureSelectXTHS Target Enrichment using the Magnis NGS Prep System protocol (version A0) (Agilent Technologies). Quality control steps were performed on the 4200 TapeStation (Agilent Technologies). Samples were sequenced using the Illumina MiSeq platform. Base calling and sample demulti-plexing were performed as standard for the MiSeq platform, generating paired FASTQ files for each sample. A negative control was included on each processing run. A targeted enrichment approach was used due to the predicted high variability of the HHV-6 and HAdV genomes.

For AAV2 WGS, an AAV2 primer scheme was designed using primal-scheme[63] with 17 AAV2 sequences from NCBI and one AAV2 sequence provided by GOSH from metagenomic sequencing of a liver biopsy DNA extract as the reference material. These primers amplify 15 overlapping 400-bp amplicons. Primers were supplied by Merck. Two multiplex

PCRs were prepared using Q5 Hot Start High-Fidelity 2X Master Mix, with a 65 °C, 3 min annealing/extension temperature. Pools 1 and 2 multiplex PCRs were run for 35 cycles. Of each PCR, 10 μl was combined and 20 μl nuclease-free water was added. Libraries were prepared either manually or on the Agilent Bravo NGS workstation option B, following a reduced-scale version of the Illumina DNA protocol as used in the CoronaHiT protocol[64]. Equal volumes of the final libraries were pooled, bead purified and sequenced on the Illumina MiSeq. A negative control was included on each processing run.

All library preparation and sequencing were performed by UCL Genomics.

**AAV2 sequence analysis.** The raw fastq reads were adapted, trimmed and low-quality reads were removed. The reads were mapped to the NC_001401 reference sequence and then the amplicon primers regions were trimmed using the location provided in a bed file. Consensus sequences were then called at a minimum of 10× coverage. The entire processing of raw reads to consensus was carried out using the nf-core/viralrecon pipeline (https://nf-co.re/viralrecon/2.4.1; https://doi.org/10.5281/zenodo.3901628). Basic quality metrics for the samples sequenced are in Supplementary Table 9. All samples that gave 10× genome coverage over 90% were then used for further phylogenetic analysis. Samples were aligned along with known reference strains from GenBank using MAFFT[65] (version v7.271), and the trees were built with IQ-TREE[66] (multicore version 1.6.12) with 1,000 rapid bootstraps and approximate likelihood-ratio test support. The samples were then labelled based on type and provider on the trees (Fig. 3a).

For each AAV2 sample, we aligned the consensus nucleotide sequence to the AAV2 reference sequence. From these alignments, the exact coordinates of the sample capsid were determined. We then used the coordinates to extract the corresponding nucleotide sequence and translated it to find the amino acid sequence. Next, we compared each sample to the reference to identify amino acid changes. Amino acid sequences from AAV capsid sequences were retrieved from GenBank for AAV1 to AAV12. Amino acid sequences of capsid constructs designed to be more hepatotropic were retrieved from refs. 16,67. These sequence sets were then aligned to the AAV2 reference sequence using MAFFT[65]. We then compared each construct to the AAV2 reference to identify amino acid changes present, while retaining the AAV2 coordinate set.

**HAdV and HHV-6B sequence analysis.** Raw data quality control was performed using trim-galore (v.0.6.7) on the raw FASTQ files.

For HHV-6B, short reads were mapped with BWA mem[68] (0.7.17-r1188) using the RefSeq reference NC_000898.

For HAdV, genotyping is performed using AYUKA[11] (version 22-111). This novel tool is used to confidently assign one or more HAdV genotypes to a sample of interest, assessing inter-genotype recombination if more than one genotype is detected. The results from this screening step guide which downstream analyses are performed and which reference genome (or genomes) is used. If mixed infection is suspected, reads are separated using bbsplit (https://sourceforge.net/projects/bbmap/), and each genotype is analysed independently as normal. If recombination is suspected, a more detailed analysis is performed using Recombination Detection Program (RDP) and the sample is excluded from phylogenetic analysis. After genotyping, the cleaned read data are mapped using BWA to the relevant reference sequence (or sequences), and SNPs and small insertions and deletions are called using bcftool (version 1.15.1, https://github.com/samtools/bcftools) and a consensus sequence is generated also with bcftools, masking with Ns positions that do not have enough read support (15× by default). Consensus sequences generated with the pipeline are then concatenated to previously sequenced samples and a multiple sequence alignment is performed using the G-INS-I algorithm in the MAFFT software (MAFFT G-INS-I v7.481). The multiple sequence alignment is then used for phylogenetic analysis with IQ-TREE (IQ-TREE 2 2.2.0), using modelfinder and performing 1,000 rapid bootstraps.

**Proteomics data generation.** Liver explant tissue from cases was homogenized in lysis buffer, 100 mM Tris (pH 8.5), 5% sodium dodecyl sulfate, 5 mM tris(2-carboxyethyl)phosphine and 20 mM chloroacetamide then heated at 95 °C for 10 min and sonicated in an ultrasonic bath for another 10 min. The lysed proteins were quantified with NanoDrop 2000 (Thermo Fisher Scientific). One-hundred micrograms was precipitated with the methanol/chloroform protocol and then protein pellets were reconstituted in 100 mM Tris (pH 8.5) and 4% sodium deoxycholate (SDC). The proteins were subjected to proteolysis with 1:50 trypsin overnight at 37 °C with constant shaking. Digestion was stopped by adding 1% trifluoroacetic acid to a final concentration of 0.5%. Precipitated SDC was removed by centrifugation at 10,000g for 5 min, and the supernatant containing digested peptides was desalted on an SOLAμ HRP (Thermo Fisher Scientific). Of the desalted peptide, 50 μg was then fractionated on Vanquish HPLC (Thermo Fisher Scientific) using a Acquity BEH C18 column (2.1 × 50 mm with 1.7-μm particles from Waters): buffer A was 10 mM ammonium formiate at pH 10, whereas buffer B was 80% acetonitrile and the flow was set to 500 μl per minute. We used a gradient of 8 min to collect 24 fractions that were then concatenated to obtain 12 fractions. These 12 fractions were dried and dissolved in 2% formic acid before liquid chromatography–tandem mass spectrometry analysis. An estimated total of 2,000 ng from each fraction was analysed using an Ultimate3000 high-performance liquid chromatography system coupled online to an Eclipse mass spectrometer (Thermo Fisher Scientific). Buffer A consisted of water acidified with 0.1% formic acid, whereas buffer B was 80% acetonitrile and 20% water with 0.1% formic acid. The peptides were first trapped for 1 min at 30 μl per minute with 100% buffer A on a trap (0.3 mm × 5 mm with PepMap C18, 5 μm, 100 Å; Thermo Fisher Scientific); after trapping, the peptides were separated by a 50-cm analytical column (Acclaim PepMap, 3 μm; Thermo Fisher Scientific). The gradient was 9–35% buffer B for 103 min at 300 nl per minute. Buffer B was then raised to 55% in 2 min and increased to 99% for the cleaning step. Peptides were ionized using a spray voltage of 2.1 kV and a capillary heated at 280 °C. The mass spectrometer was set to acquire full-scan mass spectrometry spectra (350:1,400 mass:charge ratio) for a maximum injection time set to auto at a mass resolution of 120,000 and an automated gain control target value of 100%. For a second, the most intense precursor ions were selected for tandem mass spectrometry. Higher energy collisional dissocation (HCD) fragmentation was performed in the HCD cell, with the readout in the Orbitrap mass analyser at a resolution of 15,000 (isolation window of 3 Th) and an automated gain control target value of 200% with a maximum injection time set to auto and a normalized collision energy of 30%. All raw files were analysed by MaxQuant[69] v2.1 software using the integrated Andromeda search engine and searched against the Human UniProt Reference Proteome (February release with 79,057 protein sequences) together with UniProt-reported AAV proteins and specific fasta created using EMBOSS Sixpack translating patient's virus genome. MaxQuant was used with the standard parameters with only the addition of deamidation (N) as variable modification. Data analysis was then carried out with Perseus[70] v2.05: proteins reported in the file 'proteinGroups.txt' were filtered for reverse and potential contaminants. Figures were created using Origin pro version 2022b.

**Transduction of AAV2 capsid mutants.** A transgene sequence containing enhanced green fluorescent protein (eGFP) was packaged into rAAV2 particles to track their expression in transduced cells, compared with rAAV capsids derived from canonical AAV2, AAV9 and a synthetic liver-tropic AAV vector called LK03 (ref. 15).

rAAV vector particles were delivered to Huh-7 hepatocytes at a multiplicity of infection of 100,000 vector genomes per cell before analysing eGFP expression by flow cytometry 72 h later.

**Recombinant AAV capsid sequence.** The VP1 sequence was generated by generating a consensus sequence from a multiple sequence alignment of sequenced AAV2 genomes derived from patient samples, using the Biopython[71] package AlignIO. The designed VP1 sequence was then synthesized as a 'gBlock' (Integrated DNA Technologies) and incorporated into an AAV2 RepCap plasmid (AAV2/2 was a gift from M. Fan, Addgene plasmid #104963) between the SwaI and XmaI restriction sites, using InFusion cloning reagent (product 638948, Clontech).

**AAV vector production.** rAAV particles were generated by transient transfection of HEK 293T cells as previously described[72]. In brief, $1.8 \times 10^7$ cells were plated in 15-cm dishes before transfecting the pAAV-CAG-eGFP transgene plasmid (a gift from E. Boyden, Addgene plasmid #37825), the relevant RepCap plasmid and the pAdDeltaF6 helper plasmid (a gift from J. M. Wilson, Addgene plasmid #112867), at a ratio of 10.5 μg, 10.5 μg and 30.5 μg, respectively, using PEIPro transfection reagent (PolyPlus) at a ratio of 1 μl per 1 μg DNA. Seventy-two hours post-transfection, cell pellets and supernatant were harvested and rAAV particles were purified using an Akta HPLC platform. rAAV particle genome copy numbers were calculated by quantitative PCR targeting the vector transgene region. The rAAV2 vector used in this study was purchased as ready-to-use AAV2 particles from Addgene (Addgene viral prep #37825-AAV2).

**Analysis of rAAV transduction.** Huh-7 hepatocytes (a gift from J. Baruteau, UCL) were plated in DMEM medium supplemented with 10% FBS and 1% penicillin–streptomycin supplement. The cell line was validated by testing for glypican-3 and was not tested for mycoplasma contamination. Cells were plated at a density of $1.5 \times 10^3$ cells per square centimetre and transduced with $1 \times 10^5$ viral genomes per cell. Transductions were performed in the presence or absence of 400 μg ml$^{-1}$ heparin, which was supplemented directly to cell media. Seventy-two hours after transduction, cells were analysed by microscopy using an EVOS Cell Imaging System (Thermo Fisher Scientific) before quantifying eGFP expression by flow cytometry using a Cytoflex Flow Cytometer (Beckman). eGFP-positive cells were determined by gating the live-cell population and quantifying the level of eGFP signal versus untransduced controls.

### Human short-read data analysis

**Cytokine transcriptomics analysis.** Cytokine inducible gene expression modules were derived from previously published bulk tissue genome-wide transcriptomes of the tuberculin skin test that have been shown to reflect canonical human in vivo cell-mediated immune pathways[73] using a validated bioinformatic approach[74]. Cytokine regulators of genes enriched in the tuberculin skin[73] test (ArrayExpress accession number E-MTAB-6816) were identified using Ingenuity Pathway Analysis (Qiagen). Average correlation of $\log_2$-transformed transcripts per million data for every gene pair in each of the target gene modules were compared with 100 iterations of randomly selected gene modules of the same size, to select cytokine-inducible modules that showed significantly greater co-correlation (adjusted $P < 0.05$), representing co-regulated transcriptional networks for each 59 cytokines. We then used the average $\log_2$-transformed transcripts per million expression of all the genes in each of these co-regulated modules to quantify the biological activity of the associated upstream cytokine within bulk genome-wide transcriptional profiles from AAV2-associated hepatitis ($n = 4$) obtained in the present study, compared with published $\log_2$-transformed and normalized microarray data from normal adult liver ($n = 10$) and hepatitis B adult liver ($n = 17$) (Gene Expression Omnibus accession number GSE96851)[18]. To enable comparison across the datasets, we transformed average gene expression values for each cytokine-inducible module to standardized ($Z$ scores) using mean and standard deviation of randomly selected gene sets of the same size within each individual dataset. Statistically significant differences

in $Z$ scores between groups were identified by Student's $t$-tests with multiple testing correction (adjusted $P < 0.05$).

**Proteomics differential expression.** To compare the proteomics data from the explanted livers of cases with data from healthy livers, we downloaded the raw files from two studies[19,20] from PRIDE. The raw files were searched together with our files using the same settings and databases.

We performed differential expression analyses at the protein level and peptide level using a hybrid approach including statistical inference on the abundance (quantitative approach), as well as the presence or absence (binary approach) of proteins or peptides. DEP R package version 1.18.0 was used for quantitative analysis[75]. Proteins or peptides were filtered for those detected in all replicates of at least one group (case or control). The data were background corrected and variance was normalized using variance-stabilizing transformation. Missing intensity values were not distributed randomly and were biased to specific samples (either cases or controls). Therefore, for imputing the missing data, we applied random draws from a manually defined left-shifted Gaussian distribution using the DEP impute function with parameters fun:"man", shift:1.8 and scale:0.3. The test_diff function based on linear models and the empirical Bayes method was used for testing differential expressions between the case and control samples.

**HLA typing methods.** Typing was undertaken in the liver centre units. Next-generation sequencing (sequencing by synthesis (Illumina) using AllType kits (VHBio/OneLambda), a high-resolution HLA typing method, was used.

### Statistical analysis

Fisher's exact test and two-sided Wilcoxon (Mann–Whitney) non-parametric rank sum test were used for differences between case and control groups. Where multiple groups were compared, Kruskal–Wallis tests followed by Wilcoxon pairwise tests using a Benjamini–Hochberg correction were performed. All analysis were performed in R version 4.2.0.

### Reporting summary

Further information on research design is available in the Nature Portfolio Reporting Summary linked to this article.

## Data availability

The consensus genomes from viral WGS data are deposited in GenBank. IDs can be found in Supplementary Table 7 (HAdV), Supplementary Table 9 (AAV2) and Supplementary Table 10 (HHV6). The MS proteomics data have been deposited in the ProteomeXchange Consortium via the PRIDE partner repository with the dataset identifier PXD035925.

## Code availability

The code for metagenomics and PCR analysis can be found at https://github.com/sarah-buddle/unknown-hepatitis. The transcriptomics analysis code is available at https://github.com/innate2adaptive/Bulk-RNAseq-analysis/tree/main/Zscore_gene_expression_module_analysis. The proteomics differential expression analysis code can be found at https://github.com/MahdiMoradiMarjaneh/proteomics_and_transcriptomics_of_hepatitis.

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

**Acknowledgements** UKHSA funded the metagenomics and HAdV sequencing. We thank A. Nathwani for helpful discussions. We acknowledge the considerable contribution from the GOSH microbiology laboratory. We thank the medical students who contributed to the DIAMOND consortium. All research at GOSH and UCL GOSH Institute of Child Health is made possible by the NIHR GOSH Biomedical Research Centre. The views expressed are those of the authors and not necessarily those of the NHS, the National Institute for Health Research (NIHR), the UKRI or the Department of Health and Social Care. The work was part funded by the NIHR Blood and Transplant Research Unit in Genomics to Enhance Microbiology Screening (GEMS), the National Institute for Health and Care Research (CO-CIN-01) or jointly by NIHR and UK Research and Innovation (CV220-169, MC_PC_19059). S. Morfopoulou is funded by a W.T. Henry Wellcome fellowship (206478/Z/17/Z). S.B. and O.E.T.M. are funded by the NIHR Blood and Transplant Research Unit (GEMS). M.M.M. and M.L. are supported in part by the NIHR Biomedical Research Centre of Imperial College NHS Trust. J.B. receives NIHR Senior Investigator Funding. M.N. and J.B. are supported by the Wellcome Trust (207511/Z/17/Z and 203268/Z/16/Z). M.N., J.B. and G.P. are supported by the NIHR University College London Hospitals Biomedical Research Centre. P. Simmonds is supported by the NIHR (NIHR203338). T.S.J. is grateful for funding from the Brain Tumour Charity, Children with Cancer UK, GOSH Children's Charity, Olivia Hodson Cancer Fund, Cancer Research UK and the NIHR. DIAMONDS is funded by the European Union (Horizon 2020; grant 848196). PERFORM was funded by the European Union (Horizon 2020; grant 668303).

**Author contributions** J.B., S. Morfopoulou and S.B. conceived the study, analysed the data and wrote the manuscript. J.R.B., L.A., N.S., A.L., J.C.D.L., J.H. and S.D. coordinated samples and carried out the metagenomics and confirmatory PCRs. O.E.T.M., J.A.G.-A., S.R., C.V., L.M.M.B., R.W., C.A.W., H.T., N.B., H.M., K.-A.M., S.C.H. and D.K.A. carried out genome sequencing and analyses. M.M.M., M.N., G.P., A.C., A.M., C.V., J. Rosenheim and M.L. analysed transcriptomic data. K. Thalassinos, M.L., M.M.M. and R.Z.C. generated and analysed proteomic data. S.N.W., J.R.C., J.F.A.D., A.S., L.J.T., Z.A., J.N. and K.S.H. carried out AAV2 tropism experiments. G.S., P.G., T.E.W., S.N.W. and J.R.C. helped with AAV2 PCR development. L.C., R.B., M.D., J.M., J.C.H., C.A., G.A. and T.S.J. carried out histology, immunohistochemistry and electron microscopy. B.B.K. and J. Reich provided control HHV-6 material. P. Shah and J.A. provided control samples. M.L., P. Simmonds, S.C., M.V., C.F. and M.S. provided PERFORM and DIAMONDS control samples. K.B., M.G.S., P.C. and M.O. coordinated consent and data collection from ISARIC. T.G., N.H. and C.K. provided data and samples from KCH and Birmingham Liver Units. I.U.-L., M.C., M.Z., S. Mandal, C.W., R.S., E.G., S.G., C.C., T.T., K.H., C.H., T.R., C.M., K. Templeton, C.N., M.H., R.G. and S.J.S. provided data and samples from UKHSA and devolved nations. E.T. provided reagents and contributed helpful discussions.

**Competing interests** J.B. is a member of the MHRA COVID Vaccines and Therapeutics committees; holder of Wellcome Trust, UKRI and NIHR funding; and principal investigator on the GSK LUNAR study to provide MHRA with data on SARS-CoV-2 sequences in patients treated with sotrovimab.

**Additional information**
**Correspondence and requests for materials** should be addressed to Judith Breuer.

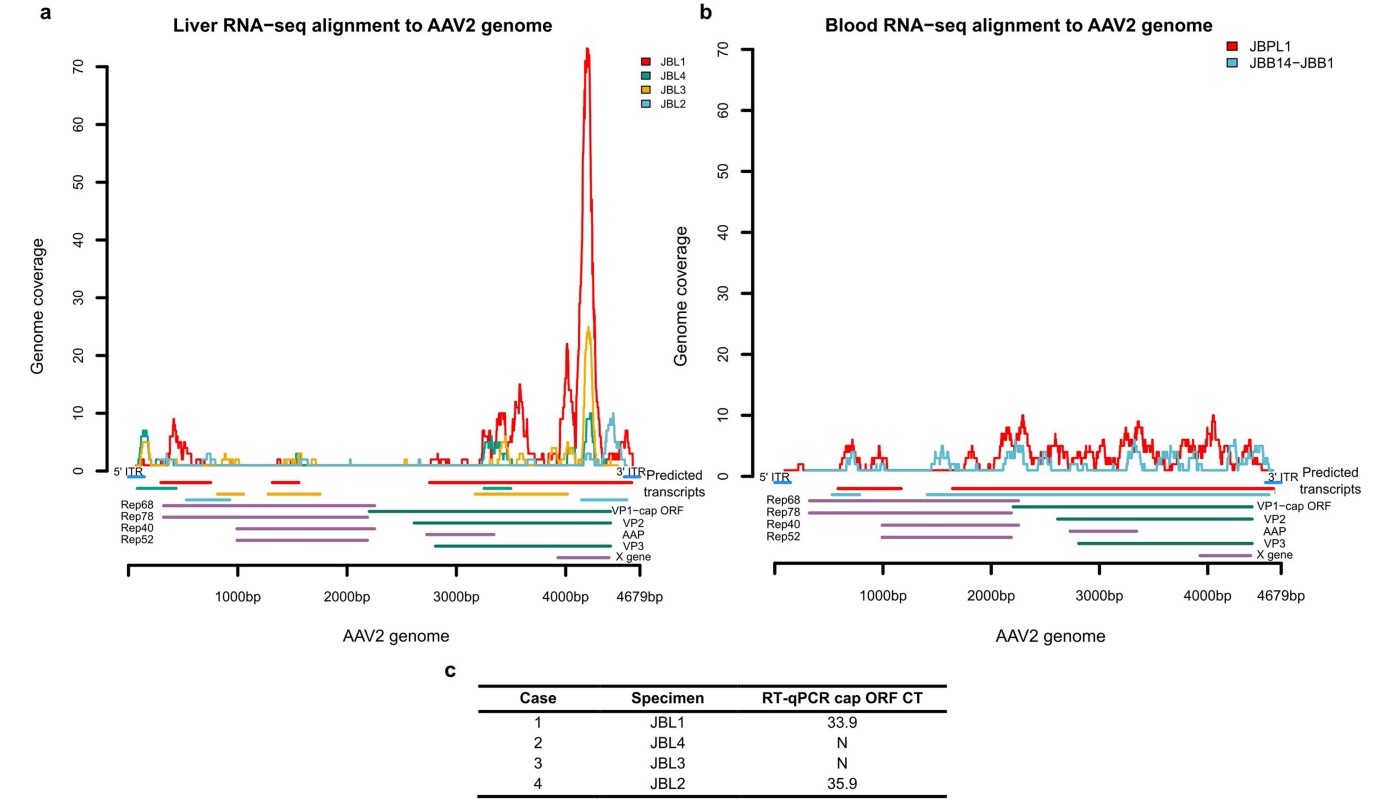

| Case | Specimen | RT-qPCR cap ORF CT |
|------|----------|--------------------|
| 1 | JBL1 | 33.9 |
| 2 | JBL4 | N |
| 3 | JBL3 | N |
| 4 | JBL2 | 35.9 |

**Extended Data Fig. 1 | Evidence of AAV2 replication from meta-transcriptomics and RT-PCR.** Mapping of AAV2 reads to the reference genome for **a** liver RNA-Seq from 4 cases, **b** blood RNA-Seq from 2 cases. The horizontal lines in the same colour as the coverage graph are the predicted transcripts for each case. The horizontal lines in purple and green are the AAV2 genes. **c**, RT-PCR results for liver cases. N: Negative PCR result.

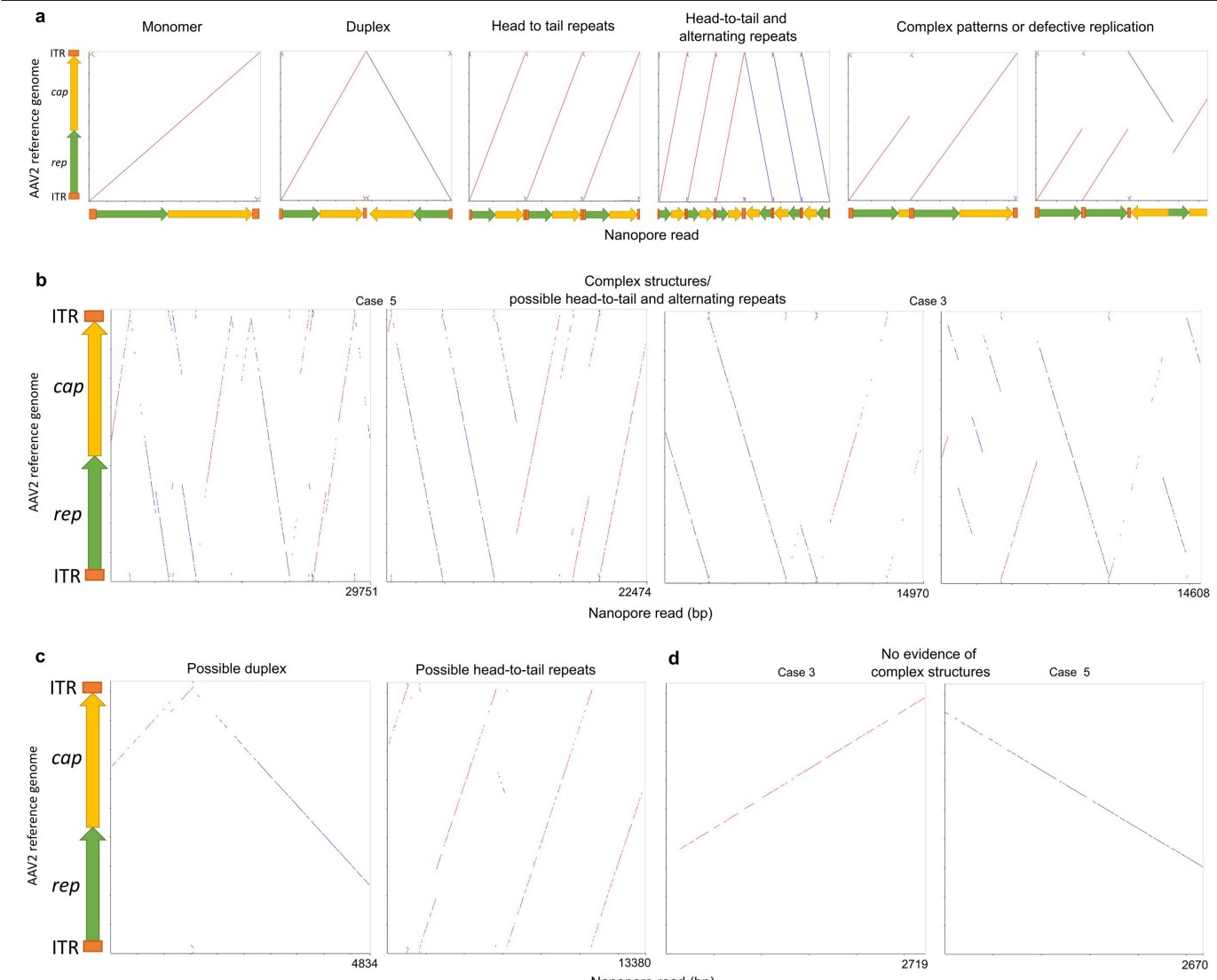

**Extended Data Fig. 2 | Examples of AAV2 complexes.** The y axis shows the coordinates of a full length AAV2 genome (rep gene in green and cap gene in yellow). X axis is the nanopore read with the length of the read indicated. Red dots indicate alignment to the forward strand and blue dots the reverse. **a**, indicative complexes based on literature[8] **b** and **c**. Examples of complex structures with both head to tail and alternating repeats, from a total of n = 25 and n = 75 such reads for cases 3 and 5 respectively. **b** shows the longest 2 reads for each case. **d**. Examples of truncated monomeric structures, from a total of n = 27 and n = 103 such reads for cases 3 and 5 respectively (Supplementary Table 3). The longest such read for each case is shown.

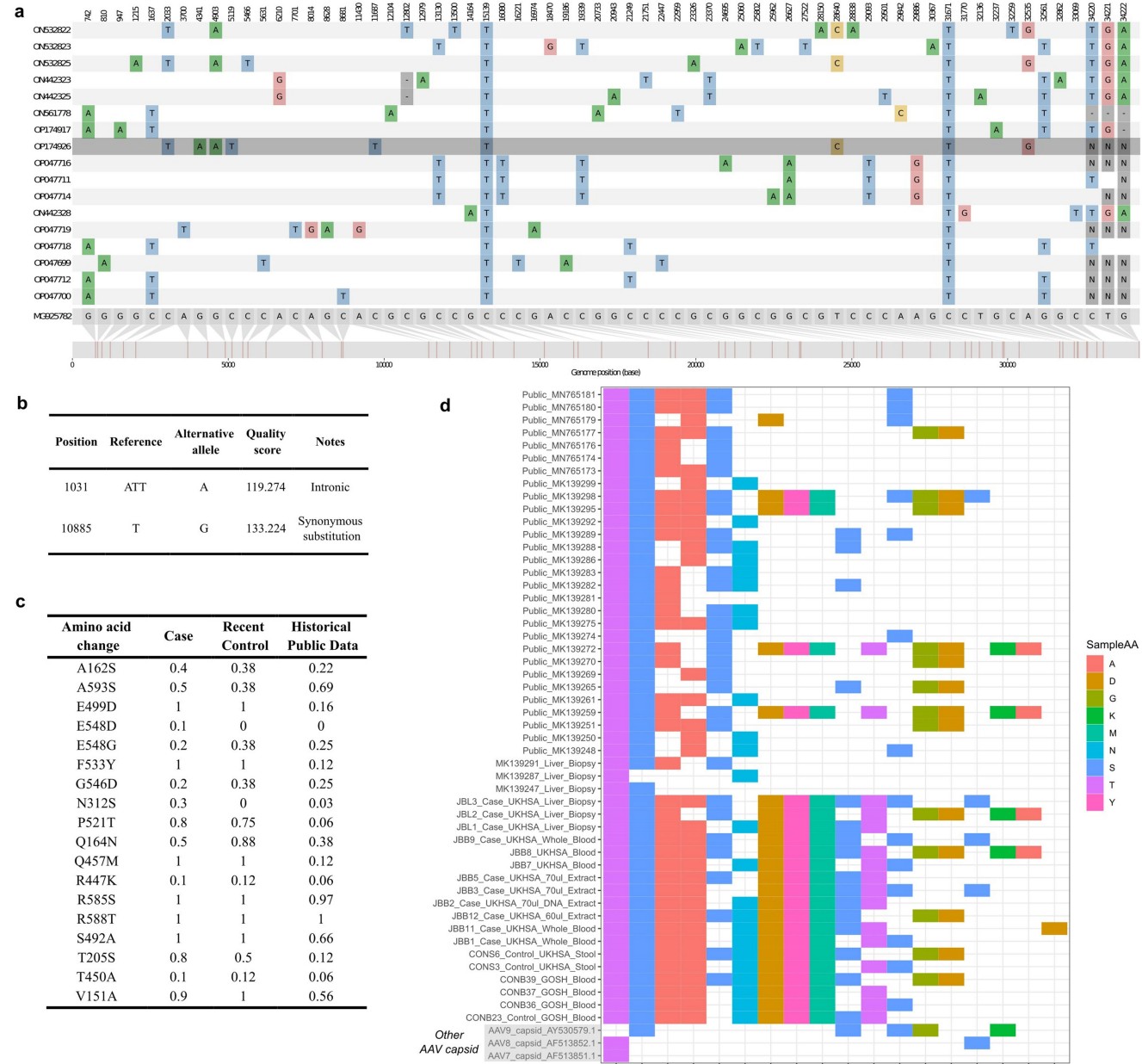

**b**

| Position | Reference | Alternative allele | Quality score | Notes |
|---|---|---|---|---|
| 1031 | ATT | A | 119.274 | Intronic |
| 10885 | T | G | 133.224 | Synonymous substitution |

**c**

| Amino acid change | Case | Recent Control | Historical Public Data |
|---|---|---|---|
| A162S | 0.4 | 0.38 | 0.22 |
| A593S | 0.5 | 0.38 | 0.69 |
| E499D | 1 | 1 | 0.16 |
| E548D | 0.1 | 0 | 0 |
| E548G | 0.2 | 0.38 | 0.25 |
| F533Y | 1 | 1 | 0.12 |
| G546D | 0.2 | 0.38 | 0.25 |
| N312S | 0.3 | 0 | 0.03 |
| P521T | 0.8 | 0.75 | 0.06 |
| Q164N | 0.5 | 0.88 | 0.38 |
| Q457M | 1 | 1 | 0.12 |
| R447K | 0.1 | 0.12 | 0.06 |
| R585S | 1 | 1 | 0.97 |
| R588T | 1 | 1 | 1 |
| S492A | 1 | 1 | 0.66 |
| T205S | 0.8 | 0.5 | 0.12 |
| T450A | 0.1 | 0.12 | 0.06 |
| V151A | 0.9 | 1 | 0.56 |

**Extended Data Fig. 3 | HAdV and AAV2 sequence analysis. a**, HAdV SNP plot: Visualisation of the multiple alignment of HAdV-F41 genomic sequences from the same clade as the single sequence from a case (highlighted in grey) (Fig. 3a). Includes both contemporary controls and publicly available HAdV-F41 genomes from GenBank. Consensus-level mutations differing from the reference sequence (bottom) are highlighted across the genome. Genomic position of the mutation is shown at the top of the plot. **b**, Variants between stool complete HAdV genome from case JBB27 and combined blood partial genomes from other cases. **c**, Frequency table of capsid residues in cases and historical controls. There is no difference between the capsid sequences of

cases and contemporaneously circulating controls. However, there are changes compared with historical controls in all contemporary sequences. None of the recently acquired capsid changes are shared with known hepatotrophic strains in AAV7, 8 and 9. **d**, Amino acid differences between AAV2 capsid sequences from cases, contemporaneously circulating controls and historical publicly available sequences compared with the AVV2 reference sequence NC_001401.2. Also shown are the capsid sequences from known AAV7, 8 and 9 hepatotropic capsids compared to the reference sequence NC_001401.2.

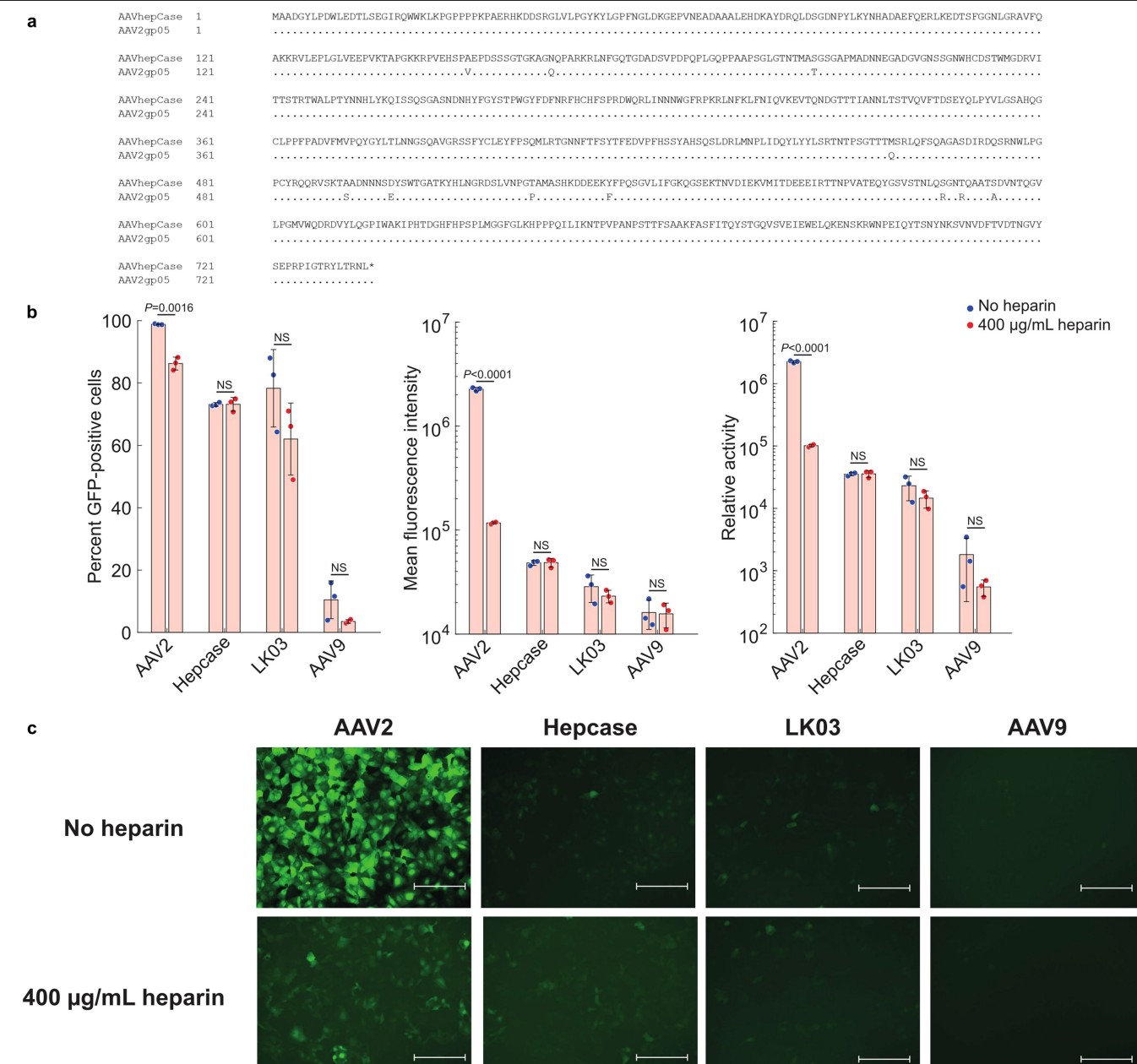

**Extended Data Fig. 4 | AAV2 capsid analysis. a**, Amino acid sequence of novel AAV capsid variant. The consensus sequence of the VP1 sequence used for investigation of capsid transduction characteristics (AAVHepcase) is shown with alignment to canonical AAV2 VP1 (AAV2gp05). The alignment shows AAV2 amino acids that are different to the AAVHepcase sequence, with dots indicating matched amino acids sequence. **b**, In vitro analysis of AAV capsid transduction characteristics. Huh-7 hepatocytes were treated at MOI 100,000 with rAAV vectors containing capsid sequences derived from canonical AAV2, a consensus sequence derived from patient sequencing samples (Hepcase), LK03, or AAV9 (n = 3 each treatment). Transduction efficiency was determined by flow cytometry, based on the percentage of EGFP-positive cells, the EGFP fluorescence intensity in positive cells, and the 'relative activity' of EGFP expression (calculated by multiplying %GFP-positive cells by MFI/10070).

Transductions were performed in the presence or absence of 400 μg/mL heparin to investigate the role of HSPG interaction. rAAV2 was significantly affected by heparin competition, whereas other capsids, including that derived from AAV Hepcase, were not. Heparin competition significantly affected rAAV2 transduction in terms of percentage of GFP-positive cells (P = 0.0016), MFI (P = 0.000008), and relative activity (P = 0.000008), whereas other capsids, including that derived from AAV Hepcase, were not affected by heparin. All data were analysed by 2-sided t-test with Bonferroni post-hoc analysis. Error bars indicate standard deviation from the mean value. **c**, Images of Huh-7 cells treated with rAAV vectors *in vitro*. Images of transduced Huh-7 cells. Each cell population was treated with MOI 100,000 of the relevant viral vector, in the presence or absence of 400 μg/mL heparin and analysed by EGFP fluorescence 72-hours post-transduction. Scale bars = 300 μm.

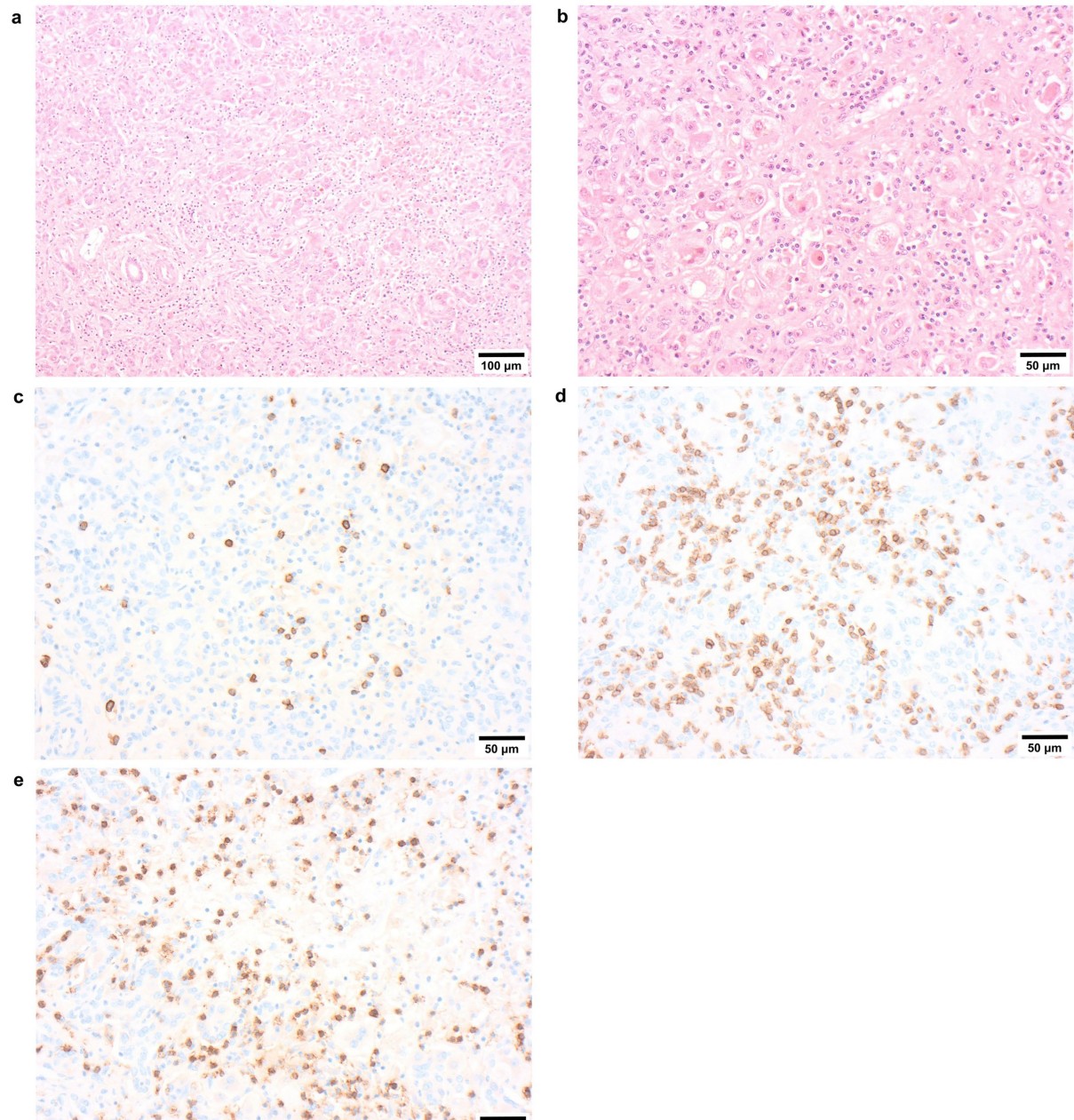

**Extended Data Fig. 5 | Representative histology of case livers. a & b**, H&E sections x100 and x200 showing a pattern of acute hepatitis with parenchymal disarray, there is a normal, uninflamed, portal tract lower left image **a**. Spotty inflammation and apoptotic bodies are shown in **b** along with perivenular hepatocyte loss/necrosis. Immunohistochemistry shows fewer mature B lymphocytes (CD20 panel **c**) than T lymphocytes (CD3, panel **d**, pan T cell marker) most of which are cytotoxic CD8 lymphocytes (panel **e**). In conclusion the livers of these children have a distinctive pattern of damage which does not indicate a specific aetiology, it does not exclude but does not offer positive support for either autoimmune hepatitis or a direct cytopathic effect of virus on hepatocytes. Each image shows a representative result from histology carried out on a minimum of five cases.

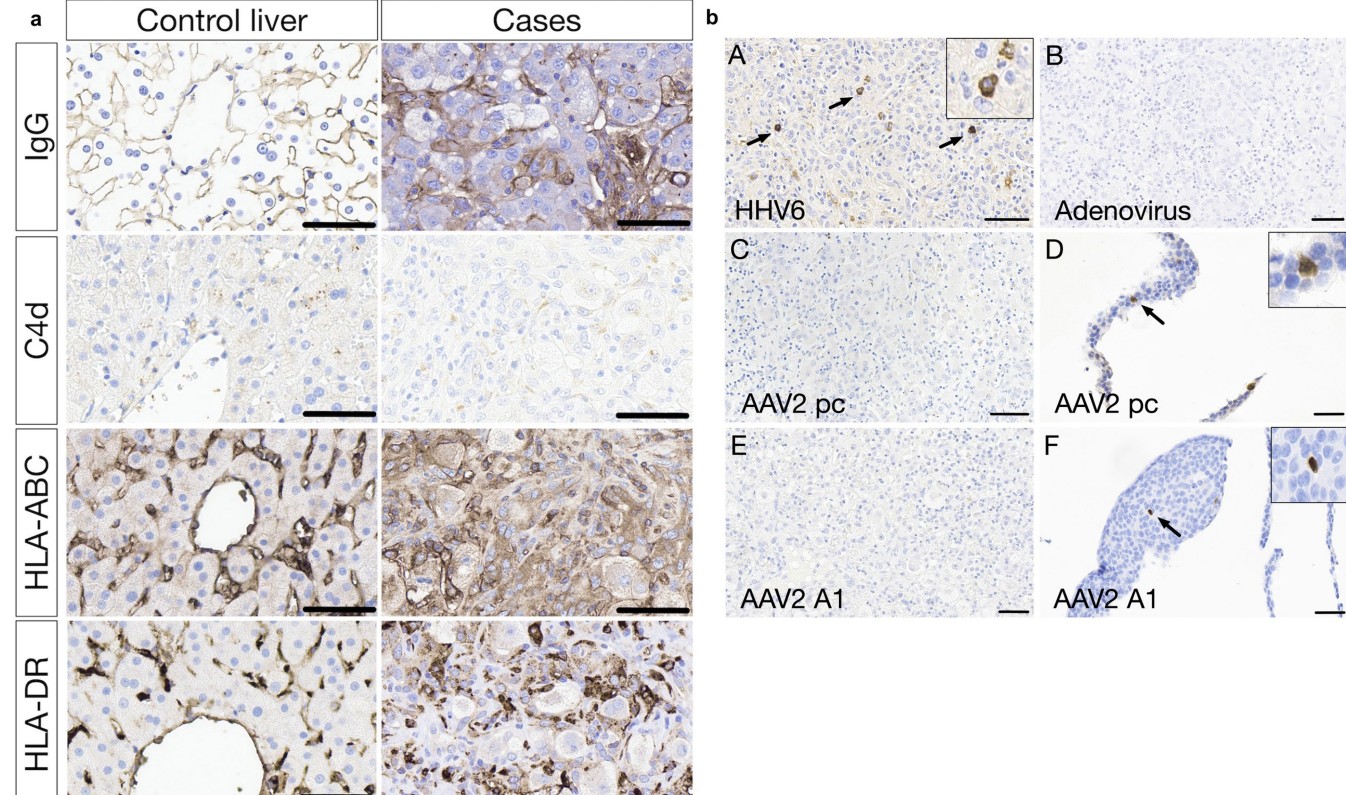

**Extended Data Fig. 6 | Immunohistochemistry results for cases of unexplained hepatitis and control tissues. a**, Inflammatory markers (IgG, C4d, HLA-ABC, HLA-DR) in acute hepatitis cases and control liver. IgG, HLA-ABC and HLA-DR show a canalicular pattern in the control liver. This pattern is disrupted in the acute hepatitis cases due to the architectural collapse. In addition, there is increased staining associated with inflammatory cell/macrophage infiltrates. C4d shows very weak staining in the acute hepatitis cases associated with macrophages but with without endothelial staining. All stains were undertaken on 5 affected cases and 13 control cases.

**b**, Representative images of the immunohistochemistry (IHC). Acute hepatitis liver explant cases stained for HHV6, arrow shows staining of **A** representative cells, **B** adenovirus, AAV2 (**C** polyclonal antibody, **E** monoclonal antibody, clone A1). Paraffin embedded AAV2 transfected cell lines stained as positive controls for AAV2 (**D** polyclonal antibody, **F** monoclonal antibody, clone A1). All scale bars are 60 micrometres. HHV6, AAV2 (polyclonal) stains were undertaken on 15 affected cases and 13 controls. AAV2 (A1) stains were undertaken on 5 affected cases and 13 control cases. Staining for adenovirus was undertaken on 5 affected cases.

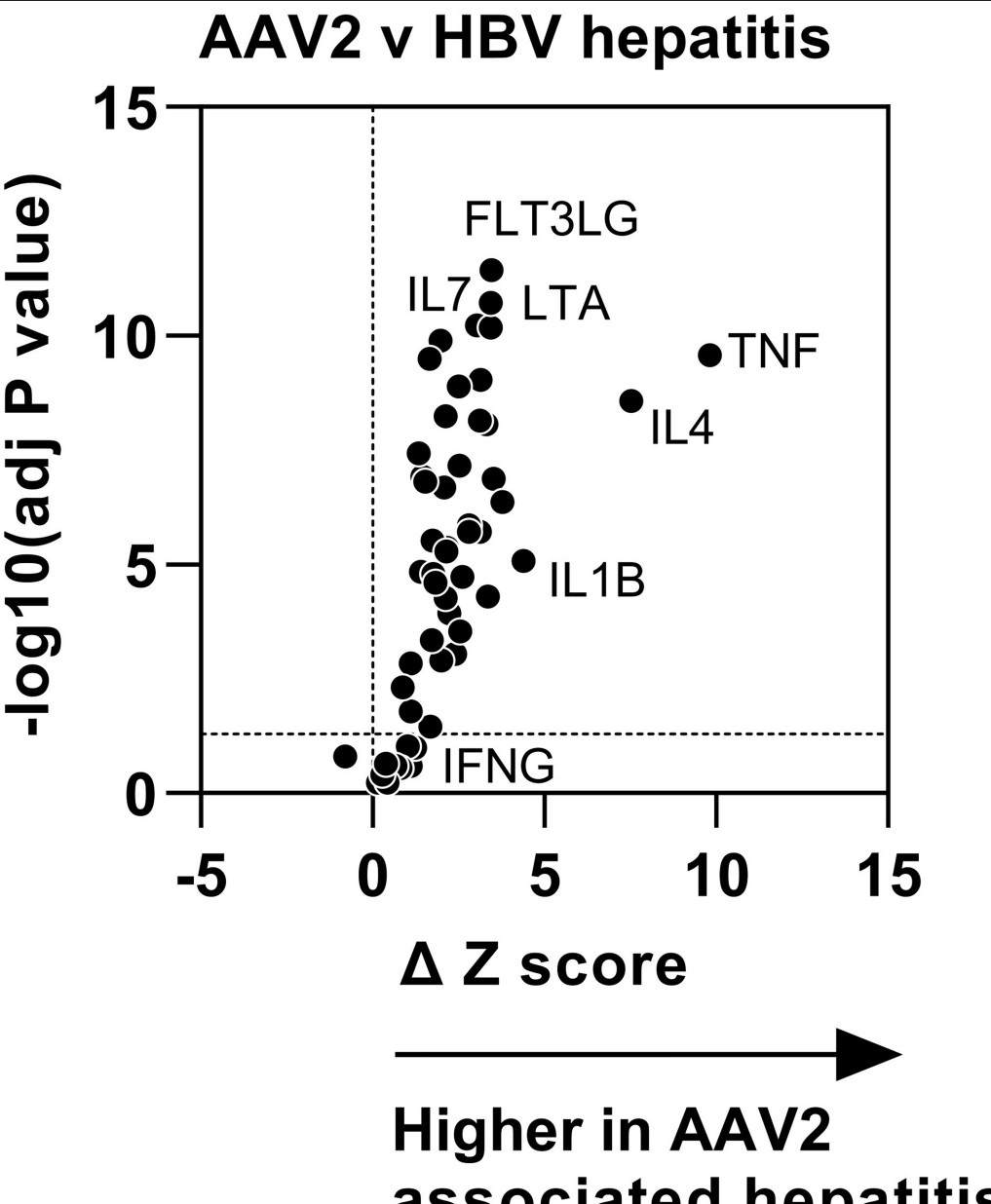

**Extended Data Fig. 7 | Cytokine inducible transcriptional modules.** Volcano plot of cytokine inducible transcriptional modules (n = 52) comparing their Z score expression in AAV2-associated hepatitis (n = 4) and HBV-associated hepatitis (n = 17) requiring transplantation using two-tailed unpaired t tests with Holm Sidak multiple testing correction for adjusted p values (n refers to number of patients). Each point represents a specific module listed in full in Supplementary Table 13. Labels for selected modules are shown.

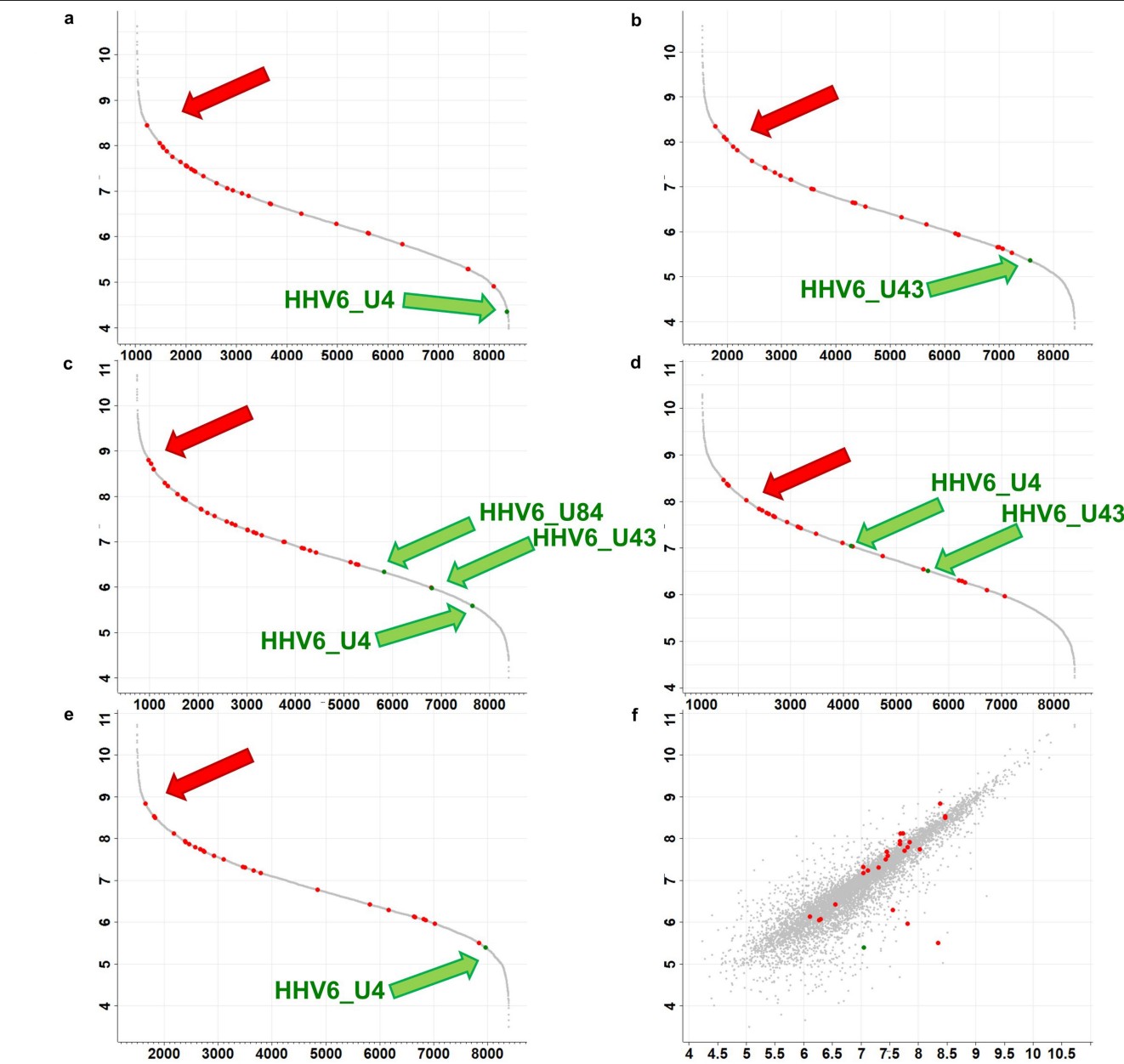

**Extended Data Fig. 8 | HLA and HHV-6B proteins in case livers. a & b** Ranking of the quantified proteins using the log10 of iBAQ values for **a** JBL1, **b** JBL2, **c** JBL3, **d** JBL4, **e** JBL5. **f**, Scatter plot of quantified proteins in sample JBL4 versus JBL5. HLA proteins are highlighted in red. Red arrows denote HLA-DRB1 proteins. HHV6 proteins are highlighted in green and marked with green arrows.

**Extended Data Table 1 | PCR and whole genome sequencing for samples from cases where metagenomic sequencing was not performed**

| Case | Sample | PCR CT values | | | Viral WGS Coverage | | |
|------|--------|------|------|--------|------------|----------|-----------|
| | | AAV2 | HAdV | HHV-6B | AAV2 (10X) | HAdV (1X) | HAdV (30X) |
| **Blood** | | | | | | | |
| 6 | JBB9 | 20 | 36 | 37 | 94 | 35.52 | - |
| 7 | JBB11 | 21 | 36 | 37 | 94 | 29.35 | - |
| 8 | JBB13 | 22 | P/N | -/N | 94 | - | - |
| 11 | JBB2 | 20 | 31 | 37 | 94 | 7.25 | 0.22 |
| 12 | JBB12 | 21 | 37 | N/N | 94 | - | - |
| 13 | JBB7 | 21 | 31 | - | 95 | - | - |
| 14 | JBB8 | 20 | 30 | - | 95 | - | - |
| 15 | JBB3 | 21 | 29 | - | 94 | 68.47 | 0.32 |
| 15 | JBB4 | 22 | 30 | - | 94 | 76.42 | 0.39 |
| 16 | JBB5 | 23 | 33 | - | 94 | 17.51 | 0.31 |
| 18 | JBB19 | - | 34 | N/- | - | 15.7 | - |
| 19 | JBB20 | - | 36 | - | - | 4.1 | - |
| 19 | JBB23 | - | 37 | P | - | 1.8 | - |
| 20 | JBB21 | - | 36 | - | - | 13.2 | - |
| 25 | JBB31 | - | P/- | - | - | 96.09 | 0.28 |
| 27 | JBB24 | - | 34 | - | - | 20.8 | - |
| **Respiratory** | | | | | | | |
| 2 | JBN1 | 25 | P/N | -/N | 88 | - | - |
| 17 | JBB18 | 30 | 39 | 45 | 85 | - | - |
| 21 | JBB26 | - | 36 | - | - | 21.6 | - |
| 23 | JBB28 | - | P/- | - | - | 100 | 99.88 |
| **Stool** | | | | | | | |
| 17 | JBB17 | 30 | -/N | -/N | 79 | - | - |
| 22 | JBB27 | - | P/- | - | - | 99.99 | 99.13 |
| 23 | JBB30 | - | P/- | - | - | 100 | 99.51 |
| 24 | JBB29 | - | P/- | - | - | 33.54 | 0.12 |
| 26 | JBB32 | - | P/- | - | - | 99.05/91.29 | 0.5/0.79 |
| **Liver (FFPE)** | | | | | | | |
| 28 (tr) | JBL6 | 25 | -/N | 32 | - | - | - |
| 29 (tr) | JBL7 | 24 | -/N | 30 | - | - | - |
| 29 (tr) | JBL8 | 25 | 40 | 30 | - | - | - |
| 30 | JBL9 | 36 | -/N | -/N | - | - | - |
| 31 (tr) | JBL10 | 24 | -/N | 30 | - | - | - |
| 32 (tr) | JBL11 | 25 | -/N | -/N | - | - | - |
| 33 (tr) | JBL12 | 24 | 41 | 31 | - | - | - |
| 34 (tr) | JBL13 | 23 | 44 | 37 | - | - | - |
| 35 | JBL14 | 34 | -/N | -/N | - | - | - |
| 36 (tr) | JBL15 | 25 | 41 | 31 | - | - | - |
| **Serum** | | | | | | | |
| 32 (tr) | JBB34 | 28 | P/N | N/N | - | - | - |
| 34 (tr) | JBB36 | 28 | P/N | P/N | - | - | - |
| 35 | JBB35 | 29 | P/N | P/N | - | - | - |
| 36 (tr) | JBB37 | 27 | 42 | -/N | - | - | - |
| 37 | JBB38 | 27 | 39 | P/N | - | - | - |
| 38 | JBB39 | 32 | P/N | -/N | - | - | - |

-: Not tested due to insufficient residual material.

N: negative PCR result.

P: Positive PCR result in referring laboratory.

Where two results are shown, the first refers to the referring laboratory and the second to GOSH. Where there was a discrepancy, the positive result is shown.

F: Failed.

Where there is more than one sample for a single patient, Ct values represent the mean across the samples that were tested.

*Metagenomics reads: the result of combining the datasets from two blood samples from the same case.

De novo assembly of unclassified metagenomics reads was unremarkable.

**Extended Data Table 2 | Controls and comparators**

### a

| Control Group | PERFORM | DIAMONDS | Total |
|---|---|---|---|
| Healthy control | 13 | 0 | 13 |
| Adenovirus, normal ALT | 10 | 7 | 17 |
| Adenovirus, normal ALT (blood) | 8 | 0 | 8 |
| Adenovirus, raised ALT | 4 | 1 | 5 |
| Critical Illness, raised ALT | 6 | 5 | 11 |
| Non-adenovirus, raised ALT | 4 | 1 | 5 |
| Other hepatitis | 5 | 1 | 6 |
| **Total** | **50** | **15** | **65** |

### b

| Control group | ALT (U/L) | Sample type | Number of controls |
|---|---|---|---|
| Adenovirus, raised ALT | >500 | Blood | 14 |
| CMV, raised ALT | >500 | Blood | 3 |
| Liver biopsy | 198–3528 | Tissue | 4 |

### c

| Age (years) | GOSH | DIAMONDS | PERFORM |
|---|---|---|---|
| 0-1 | 12 | 9 | 15 |
| 2-3 | 3 | 1 | 9 |
| 4-5 | 2 | 3 | 17 |
| 6-7 | 0 | 2 | 9 |
| **Total** | **17** | **15** | **50** |

**a** Summary of DIAMONDS and PERFORM immunocompetent controls. **b** immunocompromised comparators. **c** age distribution of blood comparator and control patients from GOSH, DIAMONDS and PERFORM.