## [Peer Review File · Nature]

Manuscript Title: Genomic investigations of acute hepatitis of unknown aetiology in children

Reviewer Comments & Author Rebuttals

Reviewer Reports on the Initial Version:

Referee #1 (Remarks to the Author):

Morfopoulou et al report the results of a retrospective observational study on 28 cases of unexplained hepatitis (nonA-E, transaminases >500IU, <10yrs) and 136 controls.

Out of the 28 cases of hepatitis, five cases underwent liver transplantation.

An increase in cases of non-A-E hepatitis in children under the age of 10 has recently been detected in several countries. With approximately 5-6% of cases requiring transplantation approximately 2% mortality this is a serious condition, yet the etiology of this form of pediatric hepatitis remains currently unknown. Human adenovirus infection (hAdV) and SARS-CoV-2 infection has been discussed as possible causes, yet the evidence has not been convincing. Identification of causes of pediatric hepatitis is urgently needed, in order to provide better diagnostic and therapeutic options for patients.

Here, the authors present evidence that human adeno-associated virus 2 (AAV2) might be associated with unexplained hepatitis. The key finding of the report is a highly significant difference in positivity rates for AAV2 within the pediatric hepatitis cohort (10/11 (91%) in blood, and 5/5 cases in liver (100%)) compared to immunocompromised controls (6/100 (6%)) and immunocompetent controls (11/32 (34%)).

Overall, these are interesting and potentially important results that might contribute to reveal the etiology of unexplained pediatric hepatitis. The main limitation of the study is its retrospective design and the heterogeneity of the case- and control cohorts, with substantial risk of bias and confounding. The inclusion and exclusion criteria for the control cohort are not clear and critical clinical information is missing.

Specific comments:

Overall, the manuscript is hard to read. The abstract is very difficult to follow, specifically with regards to the numbers. The authors state that they investigated 28 pediatric hepatitis cases and 136 controls. Out of the 28 cases, 5 cases underwent liver transplantation. Later, the abstract states that AAV2 was detected in 10/11 blood samples (which 11 samples? in Figure 1B it looks as if there were n=17 blood samples, whereas Figure 1C shows varying numbers of blood samples for the cases, e.g. n=19 for AAV2) and hAdV and HHV-6B was detected in 15/17 and 6/9 (which again is not congruent with Figure 1C). Then they state that AAV2 was found in 6/100 controls, yet before the number of control samples was given as n=136 and Figure 1C also refers to n=132 blood samples and n=4 liver samples in the control cohort. This is a source of confusion and should be clarified. The

results section is also somewhat difficult to follow, for example by separate description of cases and controls in different paragraphs. Again the numbers in the different cohorts and analysis do not always match. It is not clear to me why the number of samples for PCR analysis differs for the different viruses (AAV2 n=19, hAdV n=24, HHV6 n=34, Fig. 1), i.e. why weren't all samples analyzed for all three viruses, or at least for AAV2?

The main issue I have with the study is that the selection criteria for samples in the control group is unclear. The authors need to provide details on their criteria for including or excluding control samples in their analysis. To this end, a prospective design with parallel recruitment of cases and controls following prespecified criteria would have been preferable. The authors need to discuss this limitation and clearly lay out their rationale for the retrospective design and their inclusion and exclusion criteria. For instance, what was the rationale for using the DIAMONDS cohort as controls? Why was the control cohort not matched, for example based on key demographic and clinical criteria.

A table detailing the demographic and clinical characteristics of all cohorts analysed in the study is missing and needs to be added (typically Table 1).

The authors state that clinical information for some of the samples both in the control and in the case group was missing. It was stated that one case met the definition of <10 years of age non-A-E hepatitis, but that information on other investigations was not available. Missing clinical information should be retrieved or samples from patients with missing important clinical data should be excluded from the analysis.

The authors argue that the absence of viral AAV2 proteins in the liver sections from transplanted patients points to an indirect pathology rather than a direct lytic infection and liver damage. This is an interesting hypothesis and would be in line with the somewhat lower liver tropism of AAV2 compared to other AAVs. What is the authors hypothesis for the high levels of AAV2 RNA in the liver, in the absence of protein and virions in the liver (i.e. no evidence of productive infection)?

Minor comments

Reference list is missing from the manuscript file

The cohort of immunocompromised children is referred to as "GOSH" throughout the MS and the Figures. Although the samples derive from GOSH hospital, it would be preferable to label them as "immunocompromised" for clarity.

Referee #2 (Remarks to the Author):

This study reports the association of viral sequences with cases of otherwise non-explained severe hepatitis in children in the UK during the past ~1 year. Since the first reports of such cases a number of others have been reported, so that >200 cases have been reported in the UK and >1000 cases worldwide. The manuscript is quite hard to untangle, and there are large amounts of information in

the tables and supplementary data. The main focus is on finding pathogen or infectious-disease related sequences, and the most consistently found sequence in the hepatitis cases was for AAV2, which were found in the same proportion in the control samples. As AAV2 is not known to replicate on its own, they also sought out possible helper viruses - human adenoviruses or herpes viruses, and showed that some of those were present at vary levels or frequencies.

No evidence of any virus infections or virus infected cells by immunohistochemistry were found, so in the end their conclusion is that there is a suggestion of a role of AAV2 in the liver disease, but the mechanism is not clear and it is possible that there are other (or multiple other) causes for the disease seen. This study was quite difficult to review, and while it seems to be well done as far as it goes, the conclusion that searching for more data on the involvement of these specific viruses, or the identification of other possible causes - either infectious or non-infectious, needs to be considered in future studies...I agree that is a reasonable conclusion, but would think it is quite likely that AAV2 is not involved.

Specific review:

- 1) The premise of the study and the definition of the outbreak is difficult to follow - it is stated that this is an outbreak of infections, with a number of cases in the UK, as well as a number elsewhere in the world. It is not clear what the evidence is of a connection between the different cases (in the UK and elsewhere in Europe, but particularly elsewhere in the world); the number is reported as unusual - but is there an ascertainment issue with the publicity in this time period? Have the cases continued? Has the possibility of non-infectious causes of hepatitis or liver disease been excluded?
- 2) Was infection with SARS CoV-2 ruled out as an association, since the cases were occurring during the large omicron wave of infections?
- 3) Along the same lines, what is the case definition or evidence that the cases being reported are due to infections? How is it known that they might be due to to the same or similar agent?
- 4) Parvoviral DNA often persists in the tissues of animals long after infection (maybe for life), and for the AAVs integration may occur, so the finding that there is some DNA present is not a surprise, but what would be expected. The finding that it is primarily associated with the children with hepatitis is the strongest (only) evidence of an association, as the authors mention, since the relationship to hepatic disease seems mysterious.
- 5) Were there any DNAs from other AAVs besides AAV2 found in any of the samples?
- 6) A parvovirus has been associated with hepatitis in horses (known as Theiler's disease); the disease mechanism is likewise not defined so-far.

Referee #3 (Remarks to the Author):

The study by Morfopoulou et al. investigates the sudden rise in acute severe hepatitis of unknown etiology. In a seminal finding, the authors detect high levels of adeno-associated virus 2 (AAV2) in cases both in liver biopsies and blood samples but find only low levels of AAV2 in controls. Importantly, the authors then demonstrate that it is not lytic infection of hepatocytes that is occurring using proteomics, immunohistochemistry and EM. Unfortunately, the relevancy of the association with AAV2 and acute hepatitis is not investigated further. While the increase is interesting, we do not learn substantially more about the acute severe hepatitis cases of unknown

origin and as the authors conclude, it is not a lytic infection of hepatocytes. It is thus pure speculation that AAV2, together with HAdV41 or HHV6 could contribute to liver pathology.

1. General comment: The current wave of acute severe hepatitis of unknown etiology is a pressing issue in the (pediatric) hepatology community due to high rate of liver transplantation (5%) in these kids. The majority of case studies thus far found an association with adenovirus, particularly the F type HAdV41. This however is puzzling since adenovirus induced hepatitis is rare and usually restricted to severely immunocompromised individuals. Furthermore, on histology adenovirus hepatitis is hallmarked by nuclear viral inclusion particles, something that has not been detected in the current outbreak. Furthermore, the current cases occur in otherwise healthy children and it was thus far difficult to reconcile adenovirus infection with fulminant hepatitis leading to liver failure. As it currently stands there are four major unresolved questions: i.) is there a novel adenovirus strain that might cause acute liver failure, ii.) is another pathogen involved not yet detected, iii.) what is the role of SARS-Cov2, which has been described to cause hepatitis and persistence in organ systems such as the intestine and iv.) do these patients show an autoimmune-like (AIH-like) liver failure or is it truly an infectious hepatitis. The study now adds valuable information, identifying AAV2 as a potential new infectious agent, however due to the superficial workup of this finding, they do not provide meaningful further evidence towards solving the mystery.

Specific criticisms:

2. To understand the liver pathology of these kids, liver samples are paramount. However, the control group here is immunocompromised kids, which seems deeply flawed. While the difficulty obtaining proper control samples in this real life setting is absolutely appreciated, it still precludes a proper analysis of the liver samples. While the argument of the authors, basically suggesting immunocompromised kids would have more AAV2, is valid it still makes any comparison extremely difficult to interpret. Also, the age difference (3 vs. 10 years) is extreme. Lastly, what was the cause for hepatitis in the controls?

3. Clinical characteristics are minimal. The authors should add a detailed table with clinical parameters (mean ALTs, ASTs, IgG, histology, fibrosis, autoantibody titers, medication, BMI)

4. In Suppl. Fig. 3 the authors conclude that there is substantial viral replication, however, histology, EM and proteomics fail to detect any viral particles. How do the authors reconcile these findings? Is the replication occurring outside of the liver? What happens in a model system such as cell culture when AAV replicates? Would one expect to only detect RNA then?

5. The phylogeny data in Fig. 3 are important and seem underreported in the manuscript. One of the most important findings of this study is that all detected viruses seem known and no new SNPs were detected. That also further indicates that these cases are not linked to lytic hepatitis.

6. While the authors are applauded for including Covid seroprevalence, to rule out Covid as an immune-triggering factor, more workup is necessary (see above comments). A Covid reservoir in the intestine could be present, alternatively in cholangiocytes (that express high levels of ACE2). Is there evidence for TCR specificity in liver to spike protein?

More general comments

7. While AAV2 is known to have liver tropism, it is not commonly known to cause hepatitis. It is unclear why the current outbreak is linked to the presence of AAV2. Simple experiments such as infection of primary human hepatocytes or coinfections with adenovirus and AAV2 are lacking which would help to identify if liver damage would really be due to AAV2. Such experiments could show whether AAV2 infects hepatocytes and impacts hepatocyte viability / cellular metabolism (as opposed to trigger an immune response causing an “autoimmune-like” hepatitis).

8. Almost all cases were actively or previously infected with SARS-CoV-2. Furthermore, both the US and UK have the highest reported cases of unknown acute hepatitis and both countries had high numbers of all previous SARS-CoV-2 variants and have a high rate of seroprevalence compared to other places such as continental Europe. This at least hints at a potential role for SARS-CoV-2 in these cases. While discussed, both studies take little efforts to rule out Covid-19 as a potential explanation. Scientific questions that need to be addressed are SARS-CoV-2 persistence in stool, T-cell receptor skewing in the liver and systemic or hepatic IFN- γ upregulation. These analyses could provide evidence of a SARS-CoV-2 superantigen mechanism in an adenovirus-41F- or AAV2 sensitised host. While I appreciate that Covid must not be the explanation per se, of all the potential pathogens, Covid is the new kid on the block and both studies do not provide sufficient evidence by a mile to rule out a contribution of Covid in this epidemic of cases.

9. The evidence provided that this is indeed a viral hepatitis is limited. The association of high levels of AAV2 in affected individuals both in liver and blood is striking. However, the study did not find evidence of virus proteins, signs of infected hepatocytes on histology, or even provide principal evidence of AAV2 directed hepatotoxicity from model systems. This strongly suggests an immune-mediated pathology as discussed. To investigate this further, a proper immunologic workup, including TCR sequencing, immunofluorescence, immune phenotyping etc. should be done. This is of highest clinical relevance as the result for clinical management will be either immunosuppression or directed novel antivirals.

10. After exclusion of known etiologies, autoimmune hepatitis is often the cause of acute liver failure in both adults and children. AIH can be triggered by drugs or infections (molecular mimicry) and it has been speculated that these acute severe hepatitis cases are due to AIH-like inflammation triggered by e.g. adenovirus. Both studies undertake little efforts to rule out or investigate virus-triggered AIH. It would be paramount to provide histology of the liver samples, scoring by a liver pathologist, autoantibody titers, quantitative immunoglobulins etc. I assume that the authors have those datasets and they should be provided. In fact, the genetic association with HLA-DRB1*04:01 is strong for AIH and is even part of the international AIH study groups (IAIHG) diagnostic scoring, further suggesting these kids could have an AIH-like hepatitis.

11. One potential clue to understand if AAV2-induced hepatitis is occurring vs. an autoimmune hepatitis (potentially AAV2-triggered) could result from analyzing complement factors. Complement does not play an important role in AIH (Biewenga et al. 2020, PMID 32234403), however,

complement activation and deposition is an essential part of the immune response to AAVs as seen in rodents (Zaiss et al. 2008, PMID 32234403) and in the clinical trials that used AAVs as vectors that reported severe side effects (ClinicalTrials.gov identifiers: NCT03362502 and NCT03368742, reviewed by Muhuri et al. 2021, PMID 33393506). Could the authors stain complement in liver tissue and measure complement factors in blood?

Minor comment:

12. Is there additional data on why the cases developed severe hepatitis vs. controls? More C-sections? Breastfeeding? Environmental factors that could explain higher incidence such as daycare exposure, more or less quarantine?

Referee #4 (Remarks to the Author):

Summary, originality and significance: The study by Morfopoulou et al., presents an investigation of 28 cases of pediatric hepatitis of unknown etiology that are compared against 136 controls with a wide range of presentations. A variety of sample types (liver, blood, stool, respiratory swabs) are studied. Using both genomic and molecular diagnostic approaches the authors find AAV2 to be present in almost all cases of unexplained hepatitis, a frequency significantly higher than that observed in controls. They also find evidence of HAdV and HHV6 infection in cases, both of which can enable AAV2 replication. Although AAV2 showed evidence of replication in liver samples, no evidence was found of direct lytic infection of this tissue.

This work is original, presenting the largest study to my knowledge to date of cases from this outbreak reporting AAV2 association with non-A-E-hepatitis. The authors use a variety of unbiased and targeted techniques to investigate the etiology of these unexplained and severe cases of hepatitis. However, the significance of these important results is somewhat compromised by small sample sizes and the lack of consistency in comparisons throughout the study (i.e. almost every assay was run on a different sub-group of cases).

Data & methodology: Overall, the methods used are valid to address the questions and incorporate the necessary controls (positive and negative) for interpretation. However, in several places insufficient information is given to allow for independent evaluation of the data.

The authors present several different types of sequencing information, including unbiased metagenomic (DNA) and metatranscriptomic (RNA) sequencing, target enrichment sequencing for HAdV and HHV6 and amplicon-based sequencing for AAV2. This is all good, but a few things were unclear to me:

- I found the fragmented presentation of the metagenomic/metatranscriptomic sequencing results hard to follow. I believe it would be better to bring together the sections on de novo assembly with the description of AAV, HAdV and HHV6 and to more clearly link this with the section on replication
- Is there a clinical reason why 1 patient of the 5 non-transplant patients had 2 blood draws analyzed while others had 1? If not, it may be more clear to present just 1 blood sample (or merged

results) for this case.

- Can the authors elaborate on why a target enrichment approach was used for HHV-6 and HAdV while amplicon sequencing with virus-specific primers was used for AAV2? And why were metagenomic libraries not used for enrichment, which would have potentially increased the number of samples that could have been analyzed this way?
- Can the authors clarify why different aligners are used for analysis of unbiased vs target enriched sequencing data?
- Also, please check method details for primer synthesis for AAV2 as there appears to be a discrepancy in the description regarding ordering from Merck vs Agilent.
- Did the authors attempt reference-based or reference-guided assembly from the metagenomic / metatranscriptomic data? Given the high sequencing depths (12 samples on a NextSeq) I would expect more success in viral genome assembly from this information. It is noted under “bait design” that one sequence was assembled for AAV2 but it was not clear what approach was used for this.

After genome assembly from the WGS, the authors present a phylogenetic analysis (Figure 3). For this analysis please clarify the following;

- I found the presentation of the HAdV genome data hard to follow. The authors indicate in the text that only 1 case sample (from stool) assembled a complete genome. Consistent with this, the tree shows JBB27 alongside 2 control samples. However, in table 2B two other samples (JBB28 and JBB30 – both the same individual) also seem to have complete HAdV genome assemblies at 30X coverage. Why are these not presented in further analyses?
- For AAV2 genome sequences 14 genomes from blood are reported. However, based on tables 2A and 2B “case” column it would suggest that 3 of these are from the same individual (cases 6,7,8 are in both tables). Is this correct or are numberings not continuous across the tables? Relatedly, in Table 2A and 2B are case IDs consecutive? If these are indeed the same individuals please specify and comment on if genomes are identical.
- In addition to coverage depth, it would be helpful to report genome assembly length in table 2 and/or the genome length cut-off that was used for inclusion in the phylogenetic trees
- How were contextual sequences selected for the phylogenetic trees?
- How is the tree rooted?
- Please present the trees all in the same style and annotated with meaningful data (e.g. date and location rather than GenBank accession ID). It is very hard to follow the interpretations in the text based on the way the data is presented in these trees

The confirmatory PCR data is very challenging to interpret owing to the fragmented nature of the results and assays run. Though I understand these are precious clinical material, is it possible to perform the PCR testing on a coherent set of samples for all assays? In some cases the reasons for these shifting denominators is not explained – for example, it is stated that material was available from 12 non-transplant cases but then results for 14 cases are reported for HHV-6. I am sure I missed something, but it was quite confusing.

There were several instances where I had the impression that information was missing from the results and/or methods. Please ensure that all results are detailed in the relevant section and that all methods used to produce these results are explained in the methods or if generated elsewhere this is cited. For example;

- HLA-typing information was very minimal in the results. The phrase “Class II alleles were highly abundant” is not sufficiently descriptive nor specific, and this result becomes a key part of the final conclusion.
- I could not see where SARS-CoV-2 results were presented in the results or methods
- There was no description in the methods of the assay used to assess AAV2 replication by RT-qPCR, again this is a key result referred to several times in the conclusion

Use of statistics, conclusions and suggested improvements:

Overall, the choice of statistical methods seems appropriate, however I have a few comments on the comparisons presented between cases and controls:

- Regarding the quantitative comparisons based on differences in CT value, were all qPCRs run together for case and control cohorts? If so, this should be clearly stated in methods, otherwise I don't think it is robust to compare these values since CT is a relative measure. Was an external control or standard curve not available to instead calculate copy number?
- The sample size of the case-control comparison for liver biopsies feels too small to be meaningful, especially given the heterogeneity in clinical presentation among control samples. Similarly, for the results from whole blood samples presented in the subsequent section (and shown in Figure 2), I suggest removing presentation of groups with very few or a single individual (e.g. MIS-C or the individual not meeting case definition) unless the authors can more clearly define why these are meaningful comparison points to the case results.
- Additionally, the presentation of the majority of these results (whole blood samples section, paragraph 2) are extremely hard to interpret as so many different comparisons are presented without any context for interpretation. I suggest separating out; i) detection of each virus in control cohorts; ii) statistical comparison of differences between cases & controls in detection of AAV2; iii) statistical comparison of differences in AAV2 levels.

I have reservations that a single HAdV-F41 genome is sufficient evidence to support the conclusion that there is no difference in outbreak genomes and believe this statement should be more caveated. However, perhaps there are more genomes that can be included (see above comment)

I may have misunderstood, but the results of sequencing and PCR appear to suggest the presence of HAdV, HHV6 and AAV2 in many samples, with both HAdV and HHV6 at low levels. This seems surprising to me, and different from my understanding of other findings where HAdV is more common and both viruses are not generally seen together. Although the authors focus on HAdV and AAV2, as HHV6 was not found at higher frequencies in cases than controls, can the authors speculate on the reasons for so many HHV6 positive cases here in both case and control groups? Is this what would be expected in HAdV cases? Also, I was not sure why the counts in Figure 2C for cases did not look the same as the 6/9 cases for HHV6 in blood reported in the summary.

The statement regarding AAV2 co-occurrence with HAdV or HHV-6 in controls compared to cases is unclear. Specifically, “almost all the low level AAV2 detected in controls was in association with HAdV or HHV-6 infection, albeit at lower levels than in the cases” seems to imply the association between AAV2 and HAdV/HHV-6 is lower in controls than cases, but I don't believe that is what is being shown in Figures 2F and 2G.

I was not able to find the presentation of the SARS-CoV-2 PCR and serology findings in the results, but have some reservations about the interpretation of this data in the conclusion. It would be helpful to present the rates of SARS-CoV-2 infection at the time these cases presented to the hospital as context. However, more generally I feel that this reflects one of the challenges of small sample sizes and this should be acknowledged. Moreover, I do not believe that the absence of SARS-CoV-2 in blood or liver is convincing evidence of no contribution of SARS-CoV-2 since detection in both of these sample types is expected to be rare in the majority of cases. Given the sparse data that is presented for SARS-CoV-2, I believe the authors should instead state that no well-supported conclusion can be made from their data and instead refer to other studies.

Why do the authors suggest that 12 days is likely insufficient time for titers to decline? Is there a citation or other source that can be pointed to for this? For SARS-CoV-2 (most readers' likely frame of reference) 12 days is well into the time when most have cleared the virus. Even in very severe cases, viral load declines relatively quickly.

The section on replication of AAV2 is a nice addition to the manuscript but I believe needs to be built out further to be robust. In addition to describing the methods for cap ORF RT-qPCR including details on why this can be equated with replication, are the authors able to reconstruct transcripts rather than just look at read mapping from RNA sequencing? I was also unsure how to read Supplementary Figure 3 as the cap ORF was not marked. Ideally sequencing could be compared to a non-replication control or ratios of RNA to DNA compared in a more quantitative manner. It would also be helpful in the discussion to specifically clarify the interpretation of observing on the one hand evidence of viral replication in liver samples (PCR / sequencing) but a lack of evidence for direct lytic infection (proteomics, IHC).

Are the authors able to replicate the association of HLA DRB1*0401 allele referenced in the Scottish study by performing HLA typing in controls as well as cases?

General comments on presentation and clarity:

The manuscript appropriately cites and credits previous work.

The summary is clear, just 2 minor comments; 1) that the denominator is not 23 for any of the results presented for the 23 non-transplant cases [AAV2 (10/11), HAdV (15/17) and HHV6B (6/9)] is confusing as the first result you see; 2) is the statement "children infected in the recent HAdV-F41 outbreak" fully supported by data in the study?

The introduction is generally well written and presents all necessary information and motivation for the work. However, there is redundancy between the final paragraph of the introduction and the first paragraph of the section "whole blood samples". Overall, I feel the presentation of control cohorts is clearer in the introduction but is out of place there and suggest creating a separate section where all cohorts are clearly presented, described, and assigned shorthand names that are then used consistently throughout the manuscript to aid comprehension. To help readability, more attention should be given to the reasons each control groups is a meaningful comparison to the cases, rather than simply listing each comparison.

Tables 3 and 4 are cited before Table 2, please reorder.

There are several places where vague terminology is used; e.g. “some positive staining” in immunohistochemistry. Please be more specific when giving any results.

Would the authors consider restructuring the results around questions or purpose, rather than by technical method? In addition to my earlier comment on the fragmentation of sequencing results I believe this would also help put in context areas where results were negative (e.g. proteomics and IHC). For example, the authors might consider a section on characterization of AAV2 in cases where they bring together evidence for replication, immunohistochemistry, microscopy and proteomics. If this were well structured and clearly motivated I believe this would provide a more cohesive picture of infection in these samples.

Please write out terms in full unless abbreviation is clearly stated (e.g. “library preps” should be library preparations).

In Figure 1A; what is the source of information for the blue line? Please include in the legend more information on what this is showing. In Figures 1B & C; it would be helpful somewhere (perhaps in supplement) to show how these different sample types and assays run overlap for each patient (e.g. for which patients were multiple samples received and/or were multiple tests run and were these results consistent). I know this is shown in the tables but I believe a visual representation would be easier.

Finally, Adenovirus is abbreviated to HAdV in text but Adv in figure 1, it would be easier if only one abbreviation were used consistently.

I believe some of the larger Supplementary tables would be better presented as extended data (excel) files

The discussion is well-written but perhaps too long and could be shortened by removing repetition of information that is presented either in the introduction (first paragraph) or elsewhere in the discussion section.

The Reporting Summary is missing details on human research participants and life sciences study design. Please also add details to code availability statement for any custom code

Author Rebuttals to Initial Comments:

Referee #1

Morfopoulou et al report the results of a retrospective observational study on 28 cases of

unexplained hepatitis (non A-E, transaminases >500IU, <10yrs) and 136 controls.
Out of the 28 cases of hepatitis, five cases underwent liver transplantation.

An increase in cases of non-A-E hepatitis in children under the age of 10 has recently been detected in several countries. With approximately 5-6% of cases requiring transplantation approximately 2% mortality this is a serious condition, yet the etiology of this form of pediatric hepatitis remains currently unknown. Human adenovirus infection (hAdV) and SARS-CoV-2 infection has been discussed as possible causes, yet the evidence has not been convincing. Identification of causes of pediatric hepatitis is urgently needed, in order to provide better diagnostic and therapeutic options for patients.

Here, the authors present evidence that human adeno-associated virus 2 (AAV2) might be associated with unexplained hepatitis. The key finding of the report is a highly significant difference in positivity rates for AAV2 within the pediatric hepatitis cohort (10/11 (91%) in blood, and 5/5 cases in liver (100%)) compared to immunocompromised controls (6/100 (6%)) and immunocompetent controls (11/32 (34%)).

Overall, these are interesting and potentially important results that might contribute to reveal the etiology of unexplained pediatric hepatitis. The main limitation of the study is its retrospective design and the heterogeneity of the case- and control cohorts, with substantial risk of bias and confounding. The inclusion and exclusion criteria for the control cohort are not clear and critical clinical information is missing.

We thank this reviewer for their supportive assertions as to the severity and importance of this outbreak of unexplained paediatric hepatitis. We agree that a retrospective analysis is not ideal but in a fast-moving outbreak, it was the only option. To try to reduce the heterogeneity of case and control cohorts, we have

- a. Restricted immunocompetent controls to children in the same age range as cases, P 5, line 260.
- b. Tried to explain better the rationale for examining samples from non-aged matched or immunocompromised children, P 5-6, lines 276, line 285.

Specific comments:

1. Overall, the manuscript is hard to read. The abstract is very difficult to follow, specifically with regards to the numbers. The authors state that they investigated 28 pediatric hepatitis cases and 136 controls. Out of the 28 cases, 5 cases underwent liver transplantation. Later, the abstract states that AAV2 was detected in 10/11 blood samples (which 11 samples? in Figure 1B it looks as if there were n=17 blood samples, whereas Figure 1C shows varying numbers of blood samples for the cases, e.g. n=19 for AAV2 – and hAdV and HHV-6B was detected in 15/17 and 6/9 (which again is not congruent with Figure 1C). Then they state that AAV2 was found in 6/100 controls, yet before the number of control samples was given as n=136 and Figure 1C also refers to n=132 blood samples and n=4 liver samples in the control cohort. This is a source of confusion and should be clarified. The results section is also somewhat difficult to follow, for example by separate

description of cases and controls in different paragraphs. Again the numbers in the different cohorts and analysis do not always match. It is not clear to me why the number of samples for PCR analysis differs for the different viruses (AAV2 n=19, hAdV n=24, HHV6 n=34, Fig. 1), i.e. why weren't all samples analyzed for all three viruses, or at least for AAV2.

We apologise for the complexity of the manuscript. These figures were correct, but we accept that they require better clarification. The examples quoted above have now changed due to additional samples from further cases and better matching of controls, but we have checked that all figures are consistent across the text and figures. We have rewritten the paper to try to clarify better the numbers of cases and controls tested and for which viruses. We have also emphasised that we were largely limited in carrying out more comprehensive analyses by lack of material (Introduction line 142, Results, confirmatory real time PCR - lines 221-222; Figure 1B Legend).

2. The main issue I have with the study is that the selection criteria for samples in the control group is unclear. The authors need to provide details on their criteria for including or excluding control samples in their analysis. To this end, a prospective design with parallel recruitment of cases and controls following prespecified criteria would have been preferable. The authors need to discuss this limitation and clearly lay out their rationale for the retrospective design and their inclusion and exclusion criteria. For instance, what was the rationale for using the DIAMONDS cohort as controls? Why was the control cohort not matched, for example based on key demographic and clinical criteria.

We have added a section on the retrospective design of the study and the impact this had on control/comparator selection in the discussion of the limitations (P 14 , line 632).

We have clarified the selection criteria for controls as follows:

Immunocompetent controls

We selected immunocompetent controls (PERFORM/DIAMONDS) recruited during the COVID-19 pandemic and before. This control group was chosen because it represented a comprehensive study of large numbers of children in the relevant age group who presented with acute febrile illness for whom whole blood samples were available. We matched control groups against cases defined by four criteria 1) age <8 years, 2) adenovirus infection, 3) hepatitis, 4) healthy controls. We have clarified these criteria in the controls section of the PCR results. (Page 8, line 260)

Supplementary Table 5 shows the data on the causes of hepatitis for immunocompetent DIAMONDS/PERFORM controls.

Immunocompromised comparators

We selected these to counter speculation that AAV is latent in the liver of most children and that its detection in the unexplained paediatric hepatitis represented reactivation caused by adenovirus, hepatitis or both. By choosing immunocompromised children with hepatitis with or without adenovirus, we sought to maximise the chances of detecting reactivating AAV2. We have now explained this in the controls section of the PCR results. While we did detect AAV2 in some of these children, it was in the minority and at much lower levels than in cases of unexplained paediatric hepatitis. We have tried to make clearer the rationale for including these samples (Page 6, line 285).

Supplementary Table 6 shows the data on the causes of hepatitis for the immunocompromised comparators.

Additional controls

As an additional control group, we contacted UKHSA to seek blood samples from children who may have had HAdVF41 as part of the 2022 outbreak but who did not develop Unexplained paediatric hepatitis. UKHSA identified only one such control, a child initially classified as a case but then found to have a pre-existing cause for their hepatitis. This patient is included in the analysis (Page 6, Other subjects, line 301).

3. A table detailing the demographic and clinical characteristics of all cohorts analysed in the study is missing and needs to be added (typically Table 1).

The authors state that clinical information for some of the samples both in the control and in the case group was missing. It was stated that one case met the definition of <10 years of age non-A-E hepatitis, but that information on other investigations was not available. Missing clinical information should be retrieved or samples from patients with missing important clinical data should be excluded from the analysis.

We have contacted UKHSA and they do not have additional data on the cases that they have already sent us. Unfortunately, UKHSA were also not able to help in the time available with obtaining ethical consent for us to contact the clinical teams to consent patients and obtain these further data. We are hopeful that this will now happen but unfortunately the data are not available within the timeframe of this response.

We have obtained more clinical details for the five previously reported cases who required liver transplants and data on an additional 11 cases from the paediatric liver transplant Unit at King's College Hospital. These data have been added to Supplementary Table 1.

4. The authors argue that the absence of viral AAV2 proteins in the liver sections from transplanted patients points to an indirect pathology rather than a direct lytic infection and liver damage. This is an interesting hypothesis and would be in line with the somewhat lower liver tropism of AAV2 compared to other AAVs. What is the authors hypothesis for the high levels of AAV2 RNA in the liver, in the absence of protein and virions in the liver (i.e. no evidence of productive infection)?

To further understand why there is so much AAV2 DNA and RNA in the liver, we have undertaken a number of additional analyses. These include:

1. Construction of an AAV vector containing the consensus AAV2 sequence from contemporaneously circulating viruses. This vector has been used to evaluate the efficiency of hepatocyte transduction and the impact of heparin on hepatocyte infection. The data show
 - a. That the AAV2 capsid able to transduce hepatocytes as well as highly hepatotropic clone LK03¹

- b. That current AAV2, unlike the canonical reference strain, does not require heparin sulphate proteoglycans (HSPG) for entry ie like the hepatotropic strains LK03 and AAV9. Loss of the requirement for HSPG for hepatocyte transduction is associated wider tropism *in vivo* including for hepatocytes.

Taken together, these data show that that the circulating AAV2 is as hepatotropic as so called hepatotropic strains. More work is still needed to look at more recent changes in the capsid. It still remains to be determined whether changes in the viral AAP and X proteins both of which are involved in AAV replication also alter hepatotropism and replication. However, the data so far provide some evidence that the circulating AAV2s are capable of robust replication in liver.

2. Nanopore sequencing of the material from the five frozen liver biopsies.

The data from nanopore sequencing provides evidence for high levels of disrupted and some concatemeric AAV2 genomes in in the liver. Since AAV2 is only able to encapsidate single genomes of the correct length, we hypothesise that the concatemers and disrupted AAV2 genomes represent non-productive viral genome replication that did not result in viable proteins, was not packaged into capsids and may therefore have accumulated in the liver. (Page 6 line 194 and supplementary Figure S2). We have not yet specified where the DNA is located in the liver and calculated the copy number. However, we note that in children who died of liver failure following infusion of AAV SMN vectors, the highest AAV DNA levels at Post Mortem were in the liver https://www.ema.europa.eu/en/documents/product-information/zolgensma-epar-product-information_en.pdf

3. More detailed *in silico* analyses of the proteomic and transcriptomic data, using publicly available datasets (albeit on adults) as comparators.

These analyses now point to an immune-mediated hepatitis, and are explained in the proteomics and transcriptomics sections in the results on pages 10, lines 435 & 452, Figure 4, and in the discussion Page 13, line 593. In the absence of viral proteins and particles we propose that the pathogenesis of the unexplained hepatitis may involve the large amounts of unencapsidated AAV genetic material as a target or otherwise a hepatic protein with homology to one of the three candidate viruses.

4. From discussions with the AAV gene therapy community, the most frequent complication of AAV gene therapy is elevated transaminases in the patients, for 4-6 weeks, managed by steroids. While this has been particularly noted for more hepatotropic AAVs, it has also been observed for AAV2². Preliminary evidence is that the hepatitis is likely to be immune mediated but the mechanism is not clear as discussed in the paper. This together with evidence that fulminant hepatitis following the AAV-SMN vector infusion was associated with high levels of AAV DNA in the liver provide a rationale for thinking that the target might be AAV DNA. We discuss these data on page 13 line 575.

Referee #2

This study reports the association of viral sequences with cases of otherwise non-explained severe hepatitis in children in the UK during the past ~1 year. Since the first reports of such cases a number of others have been reported, so that >200 cases have been reported in the UK and >1000 cases worldwide. The manuscript is quite hard to untangle, and there are large amounts of information in the tables and supplementary data. The main focus is on finding pathogen or infectious-disease related sequences, and the most consistently found sequence in the hepatitis cases was for AAV2, which were found in the same proportion in the control samples. As AAV2 is not known to replicate on its own, they also sought out possible helper viruses - human adenoviruses or herpes viruses, and showed that some of those were present at vary levels or frequencies.

No evidence of any virus infections or virus infected cells by immunohistochemistry were found, so in the end their conclusion is that there is a suggestion of a role of AAV2 in the liver disease, but the mechanism is not clear and it is possible that there are other (or multiple other) causes for the disease seen. This study was quite difficult to review, and while it seems to be well done as far as it goes, the conclusion that searching for more data on the involvement of these specific viruses, or the identification of other possible causes - either infectious or non-infectious, needs to be considered in future studies...I agree that is a reasonable conclusion, but would think it is quite likely that AAV2 is not involved.

Specific review:

1) The premise of the study and the definition of the outbreak is difficult to follow - it is stated that this is an outbreak of infections, with a number of cases in the UK, as well as a number elsewhere in the world. It is not clear what the evidence is of a connection between the different cases (in the UK and elsewhere in Europe, but particularly elsewhere in the world); the number is reported as unusual - but is there an ascertainment issue with the publicity in this time period? Have the cases continued? Has the possibility of non-infectious causes of hepatitis or liver disease been excluded?

In the introduction, we have now clarified which numbers refer to UK, Europe and the world.

There has been a decline in cases since April but new cases were still being reported as of July – we have included this in the introduction.

All cases met the cases definition of previously healthy children aged under 10 years with no evidence of hepatitis A-E. For 16 cases referred from the liver units detailed data were available and none of these had any evidence for other causes of their hepatitis (Supplementary Table 1). Moreover, the four cases who did not receive liver transplants all recovered without sequelae (Dino Hadjic personal communication). Although undoubtedly some children originally classified as cases turned out to have other causes for their hepatitis, these were few compared to the majority with no other cause and were re-classified where discovered as in our one control cases with HAdV-F41 viraemia (page 6, line 301). Unfortunately, the data from cases referred to us from UKHSA are limited and we were unable to obtain more. However, we, UKHSA and the three paediatric liver transplant units in the UK do not think that this outbreak represents a question of ascertainment. The sheer numbers of children admitted to UK hospitals, intensive care units and requiring liver transplants over such a short space of time was completely unprecedented. The reports from the USA and other countries also support this as being an outbreak of something not previously

recognised. The fact that cases have now declined also supports the outbreak as being something unusual.

2) Was infection with SARS CoV-2 ruled out as an association, since the cases were occurring during the large omicron wave of infections?

From our data alone, we cannot rule out SARS-CoV-2 as a contributory factor to the findings. The incidence of SARS-CoV-2 antibodies was 66% (8/12) in the patients from the liver units and 88% (7/8) (Supplementary Table 1), in the non-transplant patients from UKHSA-England where information was available, while it was 30% in the DIAMOND controls, (latest control recruited October 2021) (Supplementary Table 5) . That being said, UKHSA have conducted a large case-control study, described in Technical Briefing 4, in which they have tested cases and controls for SARS-CoV-2. Results from this as well as from published papers in the NEJM³ and elsewhere and data from HSA Scotland, published in the Scottish study described in the paper from our collaborators⁴, did not find a difference between the detection of SARS-CoV-2 RNA or antibodies in cases from matched controls.

3) Along the same lines, what is the case definition or evidence that the cases being reported are due to infections? How is it known that they might be due to the same or similar agent?

The case definition is stated in the introduction on page 1. The evidence that cases are due to infection is circumstantial:

- The majority of cases for whom we have data gave a history of preceding gastro enteritis and this was corroborated by studies done in the USA and UK, as well as UKHSA data ^{3,5,6}
- HAdV has been found in blood and livers from the majority of cases tested and is not normally found in blood and livers of uninfected individuals.
- AAV2 has been found in blood and liver of the majority of cases tested here (27/28) but only infrequently in the controls (6/65).
- A UKHSA report has failed to identify an obvious toxic cause for the unexplained paediatric hepatitis (Hepatitis incident. Exposures from food, drink and water sources: an updated interim report, unpublished)

4) Parvoviral DNA often persists in the tissues of animals long after infection (maybe for life), and for the AAVs integration may occur, so the finding that there is some DNA present is not a surprise, but what would be expected. The finding that it is primarily associated with the children with hepatitis is the strongest (only) evidence of an association, as the authors mention, since the relationship to hepatic disease seems mysterious.

One of the reasons that we tested liver tissue and blood from immunocompromised children with hepatitis is to see whether AAV2 reactivation occurred in this group at most risk of viral reactivation. Our data do not find evidence for frequent AAV2 presence nor high levels AAV2 in the livers of immunocompromised children with hepatitis, suggesting that its detection is not a bystander effect of hepatitis. Nor was AAV2 DNA found in control FFPE tumour margin liver from a child with a hepatoblastoma. We did not find any AAV2 proteins in the publicly available normal liver proteomic data analysed as controls. Neither did we find any evidence of AAV2 integration in the liver of cases tested and indeed integration of wild-type virus is thought to be rare.

While AAV2 is found at high levels in almost all the blood and serum from cases and in paired serum from cases with AAV2 in their livers, it is infrequently found in healthy children. AAV2 was detected in blood from children with adenovirus and immunocompromised children with adenovirus hepatitis, but at much lower levels than in cases. In the light of these findings and other data we now wonder if AAV2 might be implicated as a co-factor in severe cases of HAdV hepatitis occurring outside of the outbreak, or in so called seronegative hepatitis, the histological appearance of which is said to resemble these cases. Work is ongoing to elucidate whether there is an association.

While wildtype AAV has not hitherto been associated with hepatitis, AAV gene therapy is well known to cause hepatitis which is thought to be immune mediated. This experience from the gene therapy community accords well with the proteomic and transcriptomic data in our cases which clearly shows an immune mediated process. Taken together, we now think there is now enough evidence to warrant further investigation of AAV2 as a potential trigger for this unusual cluster of unexplained paediatric hepatitis cases.

5) Were there any DNAs from other AAVs besides AAV2 found in any of the samples?

No. Even though there was a small number of reads initially classified as AAV3 or AAV1 in some samples, these reads were also originating from AAV2 – just from genomic areas that were similar between the RefSeq sequence for AAV2 and the Refseq for AAV3. Confirmatory mapping resolved these as AAV2.

6) A parvovirus has been associated with hepatitis in horses (known as Theiler's disease); the disease mechanism is likewise not defined so-far.

Thank you for this. We have now referenced Theiler's disease in the discussion (line 590).

Referee #3

The study by Morfopoulou et al. investigates the sudden rise in acute severe hepatitis of unknown etiology. In a seminal finding, the authors detect high levels of adeno-associated virus 2 (AAV2) in cases both in liver biopsies and blood samples but find only low levels of AAV2 in controls. Importantly, the authors then demonstrate that it is not lytic infection of hepatocytes that is occurring using proteomics, immunohistochemistry and EM. Unfortunately, the relevancy of the association with AAV2 and acute hepatitis is not investigated further. While the increase is interesting, we do not learn substantially more about the acute severe hepatitis cases of unknown origin and as the authors conclude, it is not a lytic infection of hepatocytes. It is thus pure speculation that AAV2, together with HAdV41 or HHV6 could contribute to liver pathology.

1. General comment: The current wave of acute severe hepatitis of unknown etiology is a pressing issue in the (pediatric) hepatology community due to high rate of liver transplantation (5%) in these kids. The majority of case studies thus far found an association with adenovirus, particularly the F type HAdV41. This however is puzzling since adenovirus induced hepatitis is rare and usually restricted to severely immunocompromised individuals. Furthermore, on histology adenovirus hepatitis is hallmarked by nuclear viral inclusion particles, something that has not been detected in the current outbreak. Furthermore, the current cases occur in otherwise healthy children and it was thus far difficult to reconcile adenovirus infection with fulminant hepatitis leading to liver failure. As it currently stands there are four major unresolved questions: i.) is there a novel adenovirus strain that might cause acute liver failure, ii.) is another pathogen involved not yet detected, iii.) what is the role of SARS-Cov2, which has been described to cause hepatitis and persistence in organ systems such as the intestine and iv.) do these patients show an autoimmune-like (AIH-like) liver failure or is it truly an infectious hepatitis.

The study now adds valuable information, identifying AAV2 as a potential new infectious agent, however due to the superficial workup of this finding, they do not provide meaningful further evidence towards solving the mystery.

Specific criticisms:

2. To understand the liver of pathology of these kids, liver samples are paramount. However, the control group here is immunocompromised kids, which seems deeply flawed. While the difficulty obtaining proper control samples in this real life setting is absolutely appreciated, it still precludes a proper analysis of the liver samples. While the argument of the authors, basically suggesting immunocompromised kids would have more AAV2, is valid it still makes any comparison extremely difficult to interpret. Also, the age difference (3 vs. 10 years) is extreme. Lastly, what was the cause for hepatitis in the controls?

Regrettably we have not been able to source frozen tissue from normal livers in children for definitive comparison with the metagenomic and proteomic data from our cases. Comparison of our results with results on FFPE tissue from controls would rightly be criticised, as the latter are much more likely to be falsely negative or low titre. We agree that the liver tissue from older immunocompromised children is not ideal and have accordingly renamed this as comparator rather than control material. However, the results do make the point that AAV2 detection cannot simply be attributed to bystander reactivation in cases of hepatitis. We have detailed the clinical

information available for the cases of hepatitis in the comparator liver biopsies (line 276). This was limited by our ethical approval to just the data available on the referral request.

3. Clinical characteristics are minimal. The authors should add a detailed table with clinical parameters (mean ALTs, ASTs, IgG, histology, fibrosis, autoantibody titers, medication, BMI)

We apologise for the lack of data on most of the samples received from UKHSA. All samples fulfil the case definition and where additional information was available, cases were reclassified such as occurred in the control detailed on page 6. Unfortunately, UKHSA are not able to help us with any more data on cases. However, we do have more data on the existing five patients who received liver transplants and an additional 11 cases received from a second liver transplant unit. In all these cases, there was no evidence for other causes of hepatitis. Details on the above are included where it is available (Supplementary Table 1). Informally, we are told that the 39 of 44 cases from Birmingham described in the NEJM paper³ who did not receive liver transplants and 4 cases from Kings Liver unit who also did not receive liver transplants, have recovered without sequelae suggesting that they did not have another underlying cause for their hepatitis.

4. In Suppl. Fig. 3 the others conclude that there is substantial viral replication, however, histology, EM and proteomics fail to detect any viral particles. How do the authors reconcile these findings? Is the replication occurring outside of the liver? What happens in a model system such as cell culture when AAV replicates? Would one expect to only detect RNA then?

Please see our answer to reviewer 1 point 4.

5. The phylogeny data in Fig. 3 are important and seem underreported in the manuscript. One of the most important finding of this study is that all detected viruses seem known and no new SNPs were detected. That also further indicates that these cases are not linked to lytic hepatitis.

We have expanded this section in the results (P6-8) and discussion.

6. While the authors are applauded for including Covid seroprevalence, to rule out Covid as a immune-triggering factor, more workup is necessary (see above comments). A Covid reservoir in the intestine could be present, alternatively in cholangiocytes (that express high levels of ACE2). Is there evidence for TCR specificity in liver to spike protein?

No SARS-CoV-2 was detected in metagenomics of liver which would have included bile ducts.

We have been able to analyse data from the transcriptomics and proteomics and these point to an immune mediated pathogenesis involving multiple pathways including CD8+ T cells. Since TCR sequencing on tissue resident cells would not be able to tell us more about the antigen, we have not prioritised this investigation, although it may be something we can do at a later stage.

More general comments

7. While AAV2 is known to have liver tropism, it is not commonly known to cause hepatitis. It is unclear why the current outbreak is linked to the presence of AAV2. Simple experiments such as infection of primary human hepatocytes or coinfections with adenovirus and AAV2 are lacking which would help to identify if liver damage would really be due to AAV2. Such experiments could

show whether AAV2 infects hepatocytes and impacts hepatocyte viability / cellular metabolism (as opposed to trigger an immune response causing an “autoimmune-like” hepatitis).

To try to address whether the AAV2 variant seen in current strains may be contributing to this picture, we have worked with collaborators to infect primary human hepatocytes with AAV2 vectors, expressing the contemporary AAV2 capsids. We report the results of these preliminary experiments on page 8, line 371. Further work is now beginning with a longer timescale to undertake further experiments to better understand the mechanism of unexplained hepatitis. In the meantime, with the proteomic and transcriptomic pointers towards an immune mediated mechanism, the data from AAV gene therapy that immune-mediated AAV related hepatitis is well known and the evidence that large amounts of disrupted AAV DNA is present in the liver of cases and the absence of any HAdV or AAV2 proteins or viral particles, the possibility remains that AAV2 DNA is the target. This hypothesis would fit with data from the AAV gene therapy literature showing that fatal cases of hepatitis following AAV gene therapy were associated with large amounts of vector DNA accumulating in the liver. An alternative is a viral protein that triggered immunity against a host molecular mimic. We are collaborating with colleagues working in AAV gene therapy to explore whether it can shed more light on a potential mechanism.

8. Almost all cases were actively or previously infected with SARS-CoV-2. Furthermore, both the US and UK have the highest reported cases of unknown acute hepatitis and both countries had high numbers of all previous SARS-CoV-2 variants and have a high rate of seroprevalence compared to other places such as continental Europe. This at least hints at a potential role for SARS-CoV-2 in these cases. While discussed, both studies take little efforts to rule out Covid-19 as a potential explanation. Scientific questions that need to be addressed are SARS-CoV-2 persistence in stool, T-cell receptor skewing in the liver and systemic or hepatic IFN- γ upregulation. These analyses could provide evidence of a SARS-CoV-2 superantigen mechanism in an adenovirus-41F- or AAV2 sensitised host. While I appreciate that Covid must not be the explanation per se, of all the potential pathogens, Covid is the new kid on the block and both studies do not provide sufficient evidence by a mile to rule out a contribution of Covid in this epidemic of cases.

See response to reviewer 2 point 2. SARS-CoV-2 has not been widely associated with hepatitis and there has been no suggestion the HLA-DRB1*04:01 is associated with severe SARS-CoV-2 disease. However, we do agree that COVID still needs to be ruled out and this will have to be the focus of future investigations and case control studies.

We also wonder whether this outbreak might be an indirect effect of COVID. The children who were most severely affected were aged <5 years. This cohort may have missed out on naturally acquiring immunity to AAV during their early years when social mixing was reduced. We do know that respiratory and GI adenovirus infections were much reduced during the pandemic and our data suggest that AAV co-infection is common with these. Could it be that encountering AAV2 and HAdV41 together with no prior exposure and at an older than usual age predisposed them to an abnormally severe infection with high viral loads which in those genetically predisposed led to an immune mediated hepatitis? This seems to use to fit the data best but we agree needs a lot more work to definitively confirm.

9. The evidence provided that this is indeed a viral hepatitis is limited. The association of high levels of AAV2 in affected individuals both in liver and blood is striking. However, the study did

not find evidence of virus proteins, signs of infected hepatocytes on histology, or even provide principal evidence of AAV2 directed hepatotoxicity from model systems. This strongly suggests an immune-mediated pathology as discussed. To investigate this further, a proper immunologic workup, including TCR sequencing, immunofluorescence, immune phenotyping etc. should be done. This is of highest clinical relevance as the result for clinical management will be either immunosuppression or directed novel antivirals.

The data from the transcriptomic and proteomic analyses strongly supports an immune mediated pathogenesis. Also the histological appearances and the data from colleagues in Scotland demonstrating an HLA predisposition. We agree that more work needs to be done and are actively setting up collaborations to pursue some of the suggestions listed. In the meantime, the finding of exceptionally high TNF levels in the livers of these patients, substantially higher than for fulminant hepatitis B infection does, we hope, present a potential immediate therapeutic option for ongoing cases.

10. After exclusion of known etiologies, autoimmune hepatitis is often the cause of acute liver failure in both adults and children. AIH can be triggered by drugs or infections (molecular mimicry) and it has been speculated that these acute severe hepatitis cases are due to AIH-like inflammation triggered by e.g. adenovirus. Both studies undertake little efforts to rule out or investigate virus-triggered AIH. It would be paramount to provide histology of the liver samples, scoring by a liver pathologist, autoantibody titers, quantitative immunoglobulins etc. I assume that the authors have those datasets and they should be provided. In fact, the genetic association with HLA-DRB1*04:01 is strong for AIH and is even part of the international AIH study groups (IAIHG) diagnostic scoring, further suggesting these kids could have an AIH-like hepatitis.

The liver histology has been reviewed by six histopathologists including three from specialist liver transplant units. They all agree that the appearance is not typical of AIH and we have detailed their conclusions on Page 9, line 398. They do note an influx of lymphocytes including CD8+ T cells, CD20 + B cells and CD 79+ B lineage cells. These histological findings fit well with the data from transcriptomics and proteomics which show showed upregulation of immunoglobulins upregulation of B-cell, T-cell and neutrophil lineages, increased variable region peptides and extremely high pro inflammatory cytokines. There is also evidence for upregulation of type 1 interferon. Taken together, the data certainly do support an immune mediated pathogenesis involving multiple pathways and the finding of an association with HLA DRB1*04:01 by others is indeed supportive of this.

11. One potential clue to understand if AAV2-induced hepatitis is occurring vs. an autoimmune hepatitis (potentially AA2-triggered) could result from analyzing complement factors. Complement does not play an important role in AIH (Biewenga et al. 2020, PMID 32234403), however, complement activation and deposition is an essential part of the immune response to AAVs as seen in rodents (Zaiss et al. 2008, PMID 32234403) and in the clinical trials that used AAVs as vectors that reported severe side effects (ClinicalTrials.gov identifiers: NCT03362502 and NCT03368742, reviewed by Muhuri et al. 2021, PMID 33393506). Could the authors stain complement in liver tissue and measure complement factors in blood?

We have now carried out Cd4 staining for complement (Supplementary Figure 9i), which was non-

specific (see page 9). Having discussed this with specialist paediatric liver histopathologists, they feel that complement staining in liver often nonspecific and as such it is rarely used in the UK. We do find evidence for upregulation of complement proteins in the proteomic analysis (Page 10) as well as antibody and agree that this pathway may be important.

Minor comment:

12. Is there additional data on why the cases developed severe hepatitis vs. controls? More C-sections? Breastfeeding? Environmental factors that could explain higher incidence such as daycare exposure, more or less quarantine?

Unfortunately, very little such data are available. As discussed above, we are in the process of obtaining additional ethical approval to gain more data on the cases but this will not be completed in time for this response.

Referee #4

Summary, originality and significance: The study by Morfopoulou et al., presents an investigation of 28 cases of pediatric hepatitis of unknown etiology that are compared against 136 controls with a wide range of presentations. A variety of sample types (liver, blood, stool, respiratory swabs) are studied. Using both genomic and molecular diagnostic approaches the authors find AAV2 to be present in almost all cases of unexplained hepatitis, a frequency significantly higher than that observed in controls. They also find evidence of HAdV and HHV6 infection in cases, both of which can enable AAV2 replication. Although AAV2 showed evidence of replication in liver samples, no evidence was found of direct lytic infection of this tissue.

This work is original, presenting the largest study to my knowledge to date of cases from this outbreak reporting AAV2 association with non-A-E-hepatitis. The authors use a variety of unbiased and targeted techniques to investigate the etiology of these unexplained and severe cases of hepatitis. However, the significance of these important results is somewhat compromised by small sample sizes and the lack of consistency in comparisons throughout the study (i.e. almost every assay was run on a different sub-group of cases).

Data & methodology: Overall, the methods used are valid to address the questions and incorporate the necessary controls (positive and negative) for interpretation. However, in several places insufficient information is given to allow for independent evaluation of the data.

1. The authors present several different types of sequencing information, including unbiased metagenomic (DNA) and metatranscriptomic (RNA) sequencing, target enrichment sequencing for HAdV and HHV6 and amplicon-based sequencing for AAV2. This is all good, but a few things were unclear to me:

- **I found the fragmented presentation of the metagenomic/metatranscriptomic sequencing results hard to follow. I believe it would be better to bring together the sections on de novo assembly with the description of AAV, HAdV and HHV6 and to more clearly link this with the section on replication**

The AAV2 replication and de novo assembly sections have been moved to directly after the metagenomic sequencing section in the results (Page 3), since they relate to analysis of the metagenomic data.

2. • Is there a clinical reason why 1 patient of the 5 non-transplant patients had 2 blood draws analyzed while others had 1? If not, it may be more clear to present just 1 blood sample (or merged results) for this case.

There was no clinical reason, so we have presented merged results for this case.

3 • Can the authors elaborate on why a target enrichment approach was used for HHV-6 and HAdV while amplicon sequencing with virus-specific primers was used for AAV2? And why were metagenomic libraries not used for enrichment, which would have potentially increased the number of samples that could have been analyzed this way?

We have clarified in the WGS section of the methods that a target enrichment approach was used for the HHV6 and HAdV due to the expected diversity (HAdV) and size (HHV-6B) of the genomes.

AAV2 is a small genome so is more amenable to PCR based methods, which in our hands tend to be more sensitive than targeted enrichment.

Where we received sufficient material (explanted livers) we were able to undertake metagenomics, however in general we were limited by the sample amount sent by UKHSA and were therefore only able to do this on the liver explants and a handful of blood samples.

4• Can the authors clarify why different aligners are used for analysis of unbiased vs target enriched sequencing data?

These were different established pipelines in our lab which historically have used different aligners.

5• Also, please check method details for primer synthesis for AAV2 as there appears to be a discrepancy in the description regarding ordering from Merck vs Agilent.

We have corrected this in the methods.

6• Did the authors attempt reference-based or reference-guided assembly from the metagenomic / metatranscriptomic data? Given the high sequencing depths (12 samples on a NextSeq) I would expect more success in viral genome assembly from this information. It is noted under “bait design” that one sequence was assembled for AAV2 but it was not clear what approach was used for this.

We attempted de novo assembly from metagenomics data for the 5 livers and we got full genomes for 3/5. These were identical or highly similar to the reference-based assembled genomes from AAV2 amplicon sequencing. As an example, the JBL2 mNGS de novo assembled genome was 100% identical to the OP161115.1 that was the AAV2 WGS-derived genome from the same case, JBL1 identical to the reference based assembled genome from AAV2-seq derived genome from the same case uploaded in Genbank as OP161114.1 etc

7. After genome assembly from the WGS, the authors present a phylogenetic analysis (Figure 3). For this analysis please clarify the following;

- I found the presentation of the HAdV genome data hard to follow. The authors indicate in the text that only 1 case sample (from stool) assembled a complete genome. Consistent with this, the tree shows JBB27 alongside 2 control samples. However, in table 2B two other samples (JBB28 and JBB30 – both the same individual) also seem to have complete HAdV genome assemblies at 30X coverage. Why are these not presented in further analyses?
- For AAV2 genome sequences 14 genomes from blood are reported. However, based on tables 2A and 2B “case” column it would suggest that 3 of these are from the same individual (cases 6,7,8 are in both tables). Is this correct or are numberings not continuous across the tables? Relatedly, in Table 2A and 2B are case IDs consecutive? If these are indeed the same individuals please specify and comment on if genomes are identical.
- In addition to coverage depth, it would be helpful to report genome assembly length in table 2 and/or the genome length cut-off that was used for inclusion in the phylogenetic trees
- How were contextual sequences selected for the phylogenetic trees?

We apologise for the confusion. We indeed had only one case from stool with a full HAdV-F41 genome, while the other 2 cases with full genomes were of different type, namely were HAdV-C. The phylogenetic tree is for HAdV-F41 sequences.

AAV2 phylogeny: we have included only one sample per case, please see supplementary Table 11.

We used 90% of the genome at 10x as cutoff for inclusion in the phylogenetic tree for AdV and AAV2, while for HHV6 80% at 5x.

We chose all complete genomes available on Genbank for the specific type (HAdV-F41 and AAV2) for inclusion in the phylogenies. For HHV6 we also included genomes from clinical samples previously sequenced in our lab.

- **How is the tree rooted?**

HAdV and HHV6 trees are mid-point rooted, while AAV2 tree is rooted to the Refseq AAV2 sequence (NC_001401.2).

- **Please present the trees all in the same style and annotated with meaningful data (e.g. date and location rather than GenBank accession ID). It is very hard to follow the interpretations in the text based on the way the data is presented in these trees**

We have now added date, location as well sample type and genbank ID for both our data and genbank data (where this information was available, otherwise we denoted with NA if missing).

8. The confirmatory PCR data is very challenging to interpret owing to the fragmented nature of the results and assays run. Though I understand these are precious clinical material, is it possible to perform the PCR testing on a coherent set of samples for all assays? In some cases the reasons for these shifting denominators is not explained – for example, it is stated that material was available from 12 non-transplant cases but then results for 14 cases are reported for HHV-6. I am sure I missed something, but it was quite confusing.

We agree that the data on which samples were tested for what is confusing and have therefore re-written the introduction and results to try to simplify. Essentially, we did not have enough material from all samples to test for all three viruses. The information on what was tested is summarised in figure 1B and Tables 1 and 2.

9. There were several instances where I had the impression that information was missing from the results and/or methods. Please ensure that all results are detailed in the relevant section and that all methods used to produce these results are explained in the methods or if generated elsewhere this is cited. For example;

- **HLA-typing information was very minimal in the results. The phrase “Class II alleles were highly abundant” is not sufficiently descriptive nor specific, and this result becomes a key part of the final conclusion.**

We apologise for this. We now have more extensive transcriptomic and proteomic data and will endeavour to make sure it is all presented. HLA typing on five transplanted cases was carried out at the liver unit and we just refer to their results. There are no data on the HLA typing for the remaining

liver cases and we did not have material to undertake this ourselves. This is all the information we have received on HLA typing methods: Next Generation Sequencing (Sequencing by synthesis (Illumina) using AllType kits (VHBio/OneLambda) – high resolution HLA typing method, that we have also included in the Methods.

10 I could not see where SARS-CoV-2 results were presented in the results or methods

We did not obtain results on SARS-CoV-2 directly, apart from not finding it in metagenomics. The SARS-Cov-2 data available for cases is presented in Supplementary Tables 1 and for controls in Supplementary Table 5. We have very little data on the presence or absence of SARS coV-2 RNA in cases other than for the liver transplant recipients, 3/12 of whom were SARS-CoV-2 positive on admission. Data on SARS-CoV-2 seropositive status from a larger cases-control study from the UKHSA case control study (Technical briefing 4) which shows no difference in seropositivity between cases and controls.

We have added a section in the discussion (line 662, page 15) emphasising that nothing can be concluded about an association with SARS-CoV-2 from our results alone but citing other studies that do address this.

11 There was no description in the methods of the assay used to assess AAV2 replication by RT-qPCR, again this is a key result referred to several times in the conclusion

Real-time PCR of AAV2 is described in the PCR section of the methods (page 22).

12 Use of statistics, conclusions and suggested improvements:

Overall, the choice of statistical methods seems appropriate, however I have a few comments on the comparisons presented between cases and controls:

- **Regarding the quantitative comparisons based on differences in CT value, were all qPCRs run together for case and control cohorts? If so, this should be clearly stated in methods, otherwise I don't think it is robust to compare these values since CT is a relative measure. Was an external control or standard curve not available to instead calculate copy number?**

The qPCR tests were not run at the same time but they were run using the same protocol. Unfortunately we did not have the material with which to construct a standard curve for calculation of copy numbers.

13 • The sample size of the case-control comparison for liver biopsies feels too small to be meaningful, especially given the heterogeneity in clinical presentation among control samples. Similarly, for the results from whole blood samples presented in the subsequent section (and shown in Figure 2), I suggest removing presentation of groups with very few or a single individual (e.g. MIS-C or the individual not meeting case definition) unless the authors can more clearly define why these are meaningful comparison points to the case results.

We have removed groups with fewer than three individuals from the analysis. This leaves 17 comparator whole bloods from immunosuppressed patients. Most of these were Adenovirus positive.

14 • Additionally, the presentation of the majority of these results (whole blood samples section, paragraph 2) are extremely hard to interpret as so many different comparisons are presented without any context for interpretation. I suggest separating out; i) detection of each virus in control cohorts; ii) statistical comparison of differences between cases & controls in detection of AAV2; iii) statistical comparison of differences in AAV2 levels.

We apologise for the confusion. We have rewritten the results to try to clarify the data.

15 I have reservations that a single HAdV-F41 genome is sufficient evidence to support the conclusion that there is no difference in outbreak genomes and believe this statement should be more caveated. However, perhaps there are more genomes that can be included (see above comment)

In addition to the one complete genome, we have used our kmer genotyping method to identify the reads mapping across the genome in partial sequences and further mapping strategies (supplementary figure 3, supplementary tables 9 and 10). Analyses of these show that they share the most SNPs with the HAdV-F41 from the case and the control who was viraemic as well as other genomes circulating in London.

16 I may have misunderstood, but the results of sequencing and PCR appear to suggest the presence of HAdV, HHV6 and AAV2 in many samples, with both HAdV and HHV6 at low levels. This seems surprising to me, and different from my understanding of other findings where HAdV is more common and both viruses are not generally seen together. Although the authors focus on HAdV and AAV2, as HHV6 was not found at higher frequencies in cases than controls, can the authors speculate on the reasons for so many HHV6 positive cases here in both case and control groups? Is this what would be expected in HAdV cases? Also, I was not sure why the counts in Figure 2C for cases did not look the same as the 6/9 cases for HHV6 in blood reported in the summary.

The natural history of HHV-6B is such that most seroconversion occurs in very young preschool children. We do not usually look for the virus in healthy children so the point prevalence of HHV6 viraemia is not known in this age group. HHV6 is known to be associated with liver disease and has been found to be upregulated in various liver conditions⁷. That being said, we detected HHV6B DNA in the liver of 11/12 cases who required transplantation but not in two cases who did not. We also identified that HHV-6B is likely to have contributed to the accumulation of disordered AAV2 DNA replication products in all five transplanted livers analysed. So in summary, while we cannot completely explain the high levels of HHV-6B in controls, we do think that it has contributed to pathogenesis in cases of unexplained paediatric hepatitis, particularly in those worst affected.

17 The statement regarding AAV2 co-occurrence with HAdV or HHV-6 in controls compared to cases is unclear. Specifically, “almost all the low level AAV2 detected in controls was in association with HAdV or HHV-6 infection, albeit at lower levels than in the cases” seems to imply the association between AAV2 and HAdV/HHV-6 is lower in controls than cases, but I don’t believe that is what is being shown in Figures 2F and 2G.

We apologise, we meant that most controls that were positive for AAV2 were also positive for HAdV and/or HHV6, and AAV2 levels were generally lower in controls than in cases. We have made both these points elsewhere, and we agree that this sentence may be confusing, so it has been removed.

18 I was not able to find the presentation of the SARS-CoV-2 PCR and serology findings in the results, but have some reservations about the interpretation of this data in the conclusion. It would be helpful to present the rates of SARS-CoV-2 infection at the time these cases presented to the hospital as context. However, more generally I feel that this reflects one of the challenges of small sample sizes and this should be acknowledged. Moreover, I do not believe that the absence of SARS-CoV-2 in blood or liver is convincing evidence of no contribution of SARS-CoV-2 since detection in both of these sample types is expected to be rare in the majority of cases. Given the sparse data that is presented for SARS-CoV-2, I believe the authors should instead state that no well-supported conclusion can be made from their data and instead refer to other studies.

We have added a section in the discussion (page 15) emphasizing that nothing can be concluded about an association with SARS-CoV-2 from our results alone but citing other studies that do address this. The serology data, when available, for the cases has been added in Supplementary Table 1 and discussed in the limitations to the study.

19 Why do the authors suggest that 12 days is likely insufficient time for titers to decline? Is there a citation or other source that can be pointed to for this? For SARS-CoV-2 (most readers' likely frame of reference) 12 days is well into the time when most have cleared the virus. Even in very severe cases, viral load declines relatively quickly.

We have added a few sentences (Page 12) to address this issue. We agree that the majority of immunocompetent children with acute adenovirus infection do clear virus within a few days and have changed the discussion to reflect this. Notwithstanding, viraemia persisting for 14 days or more is described in some immunocompetent children with severe HAdV infection and we have added that reference too. We would also have expected to detect HAdV protein persisting in the liver cells, even in the absence of viral particles, if HAdV infected cells were the target of the immune mediated pathology.

20 The section on replication of AAV2 is a nice addition to the manuscript but I believe needs to be built out further to be robust. In addition to describing the methods for cap ORF RT-qPCR including details on why this can be equated with replication, are the authors able to reconstruct transcripts rather than just look at read mapping from RNA sequencing? I was also unsure how to read Supplementary Figure 3 as the cap ORF was not marked. Ideally sequencing could be compared to a non-replication control or ratios of RNA to DNA compared in a more quantitative manner. It would also be helpful in the discussion to specifically clarify the interpretation of observing on the one hand evidence of viral replication in liver samples (PCR / sequencing) but a lack of evidence for direct lytic infection (proteomics, IHC).

We have reconstructed transcripts using STAR and StringTie, and displayed them on in Supplementary Figure 1, as well as marking the CAP ORF.

To try to better understand what is happening we have used nanopore sequencing of the AAV2 to better understand the structure of AAV2 DNA in the livers. This work is described on Page 3 and discussed in Page 12. We find that AAV2 DNA is largely present as complexes with evidence of HAdV and HHV-6B-aided replication large numbers of disrupted AAV2 DNA products. Since AAV2 only packages single genomes, this might explain the absence of viral proteins and particles. So our

hypothesis is that massive AAV2 replication in the liver, helped by both HAdV and HHV-6B is leading to the accumulation of large amounts of DNA products which cannot be packaged into viral particles.

23 Are the authors able to replicate the association of HLA DRB1*0401 allele referenced in the Scottish study by performing HLA typing in controls as well as cases?

Unfortunately, we have not been able to obtain the ethical permission to do this as the samples are held by UKHSA. We are now working with them to progress this but it will not be achieved in time for this manuscript.

General comments on presentation and clarity:

The manuscript appropriately cites and credits previous work.

24 The summary is clear, just 2 minor comments; 1) that the denominator is not 23 for any of the results presented for the 23 non-transplant cases [AAV2 (10/11), HAdV (15/17) and HHV6B (6/9)] is confusing as the first result you see; 2) is the statement “children infected in the recent HAdV-F41 outbreak” fully supported by data in the study?

We have removed the reference to 23 non-transplant cases. All the cases had tested positive for adenovirus either by us or by the referring centre, but we can't say definitively from our data that they were all infected in the HAdV-F41 outbreak, so we have removed this phrase.

25 The introduction is generally well written and presents all necessary information and motivation for the work. However, there is redundancy between the final paragraph of the introduction and the first paragraph of the section “whole blood samples”. Overall, I feel the presentation of control cohorts is clearer in the introduction but is out of place there and suggest creating a separate section where all cohorts are clearly presented, described, and assigned shorthand names that are then used consistently throughout the manuscript to aid comprehension. To help readability, more attention should be given to the reasons each control groups is a meaningful comparison to the cases, rather than simply listing each comparison.

The detailed description of the controls has been moved from the introduction to the results and the section in the results has been edited for clarity (Pages 5 & 6). Shorthand names have been assigned and reasons for including each group has been given.

26 Tables 3 and 4 are cited before Table 2, please reorder.

We have now checked that the ordering of all figures and tables is correct.

27 There are several places where vague terminology is used; e.g. “some positive staining” in immunohistochemistry. Please be more specific when giving any results.

We have rewritten the IHC results section to describe some additional results and have been more specific (Page 9).

28 Would the authors consider restructuring the results around questions or purpose, rather than by technical method? In addition to my earlier comment on the fragmentation of sequencing results I believe this would also help put in context areas where results were negative (e.g.

proteomics and IHC). For example, the authors might consider a section on characterization of AAV2 in cases where they bring together evidence for replication, immunohistochemistry, microscopy and proteomics. If this were well structured and clearly motivated I believe this would provide a more cohesive picture of infection in these samples.

We have edited the entire results section for clarity, and have taken the advice from this referee to structure results accordingly, which we hope provides a clearer explanation.

29 Please write out terms in full unless abbreviation is clearly stated (e.g. “library preps” should be library preparations).

We have corrected this.

30 In Figure 1A; what is the source of information for the blue line? Please include in the legend more information on what this is showing.

The source of the information is UKHSA and specifically: Source: secondary Generation Surveillance system data: laboratory reports to UKHSA of a positive adenovirus result conducted by a laboratory in England, and includes any sample type.

In Figures 1B & C; it would be helpful somewhere (perhaps in supplement) to show how these different sample types and assays run overlap for each patient (e.g. for which patients were multiple samples received and/or were multiple tests run and were these results consistent). I know this is shown in the tables but I believe a visual representation would be easier.

We tried to make a suitable visualisation prior to submission, but the overlap is very complicated and visualisation made it more confusing. Where multiple samples were received for the same sample type, the mean result is presented in Figure 2, but reported individually in Table 2 and Supplementary Table 6.

31 Finally, Adenovirus is abbreviated to HAdV in text but Adv in figure 1, it would be easier if only one abbreviation were used consistently.

HAdV is now used throughout.

32 I believe some of the larger Supplementary tables would be better presented as extended data (excel) files

Supplementary Figure 6 (PCR data), the largest table, will instead be presented as an extended data file, as well as the Clinical details for cases (Supplementary Table 1) and controls (Supplementary Table 5), along with proteomics/transcriptomics tables (Sup. Tables 13,14,15,16).

33 The discussion is well-written but perhaps too long and could be shortened by removing repetition of information that is presented either in the introduction (first paragraph) or elsewhere in the discussion section.

We have removed the sentences in the first paragraph that duplicate information in the introduction and have checked for repetition elsewhere.

34 The Reporting Summary is missing details on human research participants and life sciences study design. Please also add details to code availability statement for any custom code

We have now added details on the 5 cases we had ethics to study their human transcriptomic/proteomic data in the human research participants and clarified for the remaining cases that we had ethics permitting us to only perform diagnostic tests on their clinical specimens.

The life sciences study design is not relevant as this was a case-series, however we now clarify we have excluded controls to age-match the cases (<8 years old) or controls that were in disease categories that had less than 3 subjects, addressing also one of your comments (#14).

1. Lisowski, L. *et al.* Selection and evaluation of clinically relevant AAV variants in a xenograft liver model. *Nature* **506**, 382–386 (2014).
2. Nathwani, A. C. Gene therapy for hemophilia. *Hematology: the American Society of Hematology Education Program* **2019**, 1 (2019).
3. Kelgeri, C. *et al.* Clinical Spectrum of Children with Acute Hepatitis of Unknown Cause. <https://doi.org/10.1056/NEJMoa2206704> (2022) doi:10.1056/NEJMoa2206704.
4. Ho, A. *et al.* Adeno-associated virus 2 infection in children with non-A-E hepatitis. *medRxiv* 2022.07.19.22277425 (2022) doi:10.1101/2022.07.19.22277425.
5. Sanchez, L. H. G. *et al.* A Case Series of Children with Acute Hepatitis and Human Adenovirus Infection. <https://doi.org/10.1056/NEJMoa2206294> (2022) doi:10.1056/NEJMoa2206294.
6. Investigation into acute hepatitis of unknown aetiology in children in England: technical briefing 4. (2022).
7. Ozaki, Y. *et al.* Frequent detection of the human herpesvirus 6-specific genomes in the livers of children with various liver diseases. *J Clin Microbiol* **39**, 2173–2177 (2001).

Reviewer Reports on the First Revision:

Referee #1 (Remarks to the Author):

The manuscript by Morfopoulou et al. has been improved by addition of further samples and additional analyses. Overall, the cohort has been increased by 11 cases of unexplained hepatitis to a total of 39 cases and additional controls, and the conclusions of the study have been strengthened by new data. The abstract and the description of the results has been improved, which has enhanced the clarity.

Yet, a main concern regarding incomplete basic demographic and clinical information on the cohort has not been fully addressed. The authors explain that the information could not be obtained. Although the difficulty of obtaining well-characterized clinical samples is fully appreciated, data derived from clinical samples of cases, for which key clinical or demographic data (sex, age, history, comorbidities, medication) are missing, are difficult to interpret and the reliability of the initial diagnosis cannot be validated. This is a significant weakness of the study. The authors stated in their reply that additional ethics would have to be obtained to acquire the missing data and it should be discussed when the missing information could be available.

A clear and complete overview of the clinical characteristics of all cases and controls included in the study should be compiled into one table.

Overall, the quality of the figures needs to be improved. Fonts are variable and the labels in Fig. 3+4 are hardly legible.

The additional experimental data are convincing and strengthen the manuscript.

Referee #2 (Remarks to the Author):

My main concerns were about the indirect evidence and the confusing nature of the study presented. The revised manuscript addresses most of my main concerns, and some new data is included that adds additional depth to the analysis.

Despite this, there is still a lot of uncertainty about the causes of the liver disease seen in the children, but that is better reflected in the current version.

Some technical or remaining issues to address, mainly about parvovirus and AAV details:

1) Lines 172 - 180. The statements regarding "high levels or abundant" AAV2 DNA seems to be at odds with the number of sequences being reported 7-42/million, while 1.2-42/million is called "lower"...

2) Ct values and DNA amounts needs to be clarified (e.g. around line 235 or line 240+, but also elsewhere) - what is called "low" and what is "high" is reversed, so care should be taken to indicate what the levels of DNA are - as that is the key finding of interest.

3) Various places - the HSPG binding of the original prototypic AAV-2 strains is likely a tissue culture adaptation since that virus was isolated as a contamination of an adenovirus culture, and appears not to be a feature of viruses infecting humans - the text should reflect that.

4) Care should be taken about the DNA sequence arrangements being "proof" of active replication. Line 535 - the concatamers of DNA are not proof of active AAV-2 replication, but are often arising from the introduction of viral DNA that is rearranged within the cell (i.e. most AAV vectors do not have Rep present (e.g. ref 7 or 44)) yet the DNA ends up in complex configurations likely due to cell responses to naked linear DNAs in the nucleus which is subject to DNA damage or other responses.

5) Line 557 and nearby - the lack of capsids is not related to the DNA replication, as capsids would form whether or not ssDNA is present (the genome is packaged into a preformed and stable capsid during replication).

Referee #3 (Remarks to the Author):

I acknowledge that the authors have addressed my comments. There are still some gaps and unknowns, but the additional evidence support the hypothesis and conclusion from the initial findings.

Referee #4 (Remarks to the Author):

Morfopoulou et al have added several new analyzes and findings to the revised paper. In particular, the Nanopore sequencing and in vitro experiments to investigate hepatotropism are nice additions offering more molecular details of AAV2 infection in undiagnosed hepatitis. The authors have also improved the presentation of the results and the study is less challenging to follow than before but misses a clear thesis. The authors have performed a large number of different analyses and investigations which are each individually interesting but it is unclear what the main conclusion from all these different findings are. A shorter and more focused discussion is one thing that could help here.

At the end of the summary and in response to one of my questions (#16) the authors note they believe HHV-6B has contributed to pathogenesis, in particular in more severe cases. Is this conclusion (regarding severity) based on detection in explanted liver but not in blood? Apologies if I missed more data here but this alone seems insufficient and open to many biases (e.g. tropism, sensitivity and small sample sizes).

I find the long-read sequencing data intriguing, but am concerned that 6-12 reads is insufficient to confidently determine these replication phenomena robustly. Can the authors sequence these libraries more deeply or use adaptive sequencing to selectively enrich for AAV2 sequences?

It seems to me that for the lack of detection of HAdV and HHV6 RNA a lack of sensitivity to detect this cannot be ruled out, in particular given the lower viral loads of these viruses compared to AAV2 by PCR and DNaseq and the detection of HHV6 protein. This analysis should be more carefully caveated or deeper sequencing or targeted enrichment (as done for the DNA) may be warranted.

It is unfortunate that no viral proteins are detectable by IHC, microscopy or proteomics which would support direct infection. The authors suggest time since infection as one possible reason for this, but is there evidence that the assay is sufficiently sensitive? My understanding is that other studies have identified such evidence in similar samples with a similar time since infection.

While I appreciate the broader efforts to simplify presentation of the sample sets, I find the revised abstract to be a little misleading. Specifically, the phrase; “using genomic, transcriptomic, proteomic and immunohistochemical methods, we undertook extensive investigation of 39 cases, ...” overstates the sample sets analyzed. For example, I believe only 5 cases were studied for both proteomic and IHC investigations.

The figures are not of as high quality as would be expected for publication. In figure 1, text is too small in figure 1B and what is the Y axis for the red and green dots on 1A? Figure 3 is very challenging to read and lacks important information including confidence values for A and C (they seem to be there for B but are too small to see).

Author Rebuttals to First Revision:

Referee #1

The manuscript by Morfopoulou et al. has been improved by addition of further samples and additional analyses. Overall, the cohort has been increased by 11 cases of unexplained hepatitis to a total of 39 cases and additional controls, and the conclusions of the study have been strengthened by new data. The abstract and the description of the results has been improved, which has enhanced the clarity.

Yet, a main concern regarding incomplete basic demographic and clinical information on the cohort has not been fully addressed. The authors explain that the information could not be obtained. Although the difficulty of obtaining well-characterized clinical samples is fully appreciated, data derived from clinical samples of cases, for which key clinical or demographic data (sex, age, history, comorbidities, medication) are missing, are difficult to interpret and the reliability of the initial diagnosis cannot be validated. This is a significant weakness of the study. The authors stated in their reply that additional ethics would have to be obtained to acquire the missing data and it should be discussed when the missing information could be available. A clear and complete overview of the clinical characteristics of all cases and controls included in the study should be compiled into one table.

We have taken the following steps to try to address the lack of clinical data

1. We now have a complete data set on all 16 patients seen by the two liver units as described in lines 140-149 and have summarised the information obtained in the manuscript (lines 140-149). Full details are provided in Supplementary Table 1
2. Of the 22 Health Security Agencies (HSAs)-referred cases, the identity of 12 was made available by UKHSA to the ISARIC consortium. Through ISARIC, parental consent to release additional data was obtained for seven of the 12, while five refused. One of the seven turned out to be a duplicate of one of the transplanted cases in Birmingham for whom we already had data. So, the total number of cases investigated has dropped to 38. In total therefore, we now have more detailed data on 22/38 cases (16 from liver units + 6) (Supplementary Table 1).
3. Of the remaining 10/22 HSA patients, (12 referred for consent for research to ISARIC), all fulfil the case definition. We are in contact with UKHSA to try to expedite transfer of these patient details to the ISARIC consortium in order that they can be consented for further analysis. Currently we have no information as to when this will happen.
4. Whilst obtaining additional data we discovered that one of our cases (Case 10) was in fact 9 years old and not 7 years old as previously thought. All DIAMOND/PERFORM controls had previously been selected to be aged <8 years to match the cases. To deal with this, we examined the controls received from DIAMONDS/PERFORM (we had previously been asked to remove controls that were not the same age) who were aged 8 and 9 years old. We identified 5, now included in Supplementary Table 4 & 5. The details of these children are outlined in the table below. Because none were positive for AAV2 or HAdV, we did not remake the figures but have instead added a footnote to Table 1 that one case was aged 9 years.

ID	Cohort	Sample type	Control type	AAV2 PCR result
JBB15	Case	blood	Case	Negative
BIV-1301-1222-E1	PERFORM	blood	Non-adenovirus raised ALT	Negative
BIV-1101-1425-E01	PERFORM	blood	Non-adenovirus raised ALT	Negative
BIV-1101-1428-E01	PERFORM	blood	Healthy	Negative
BIV-1101-1432-E01	PERFORM	blood	Healthy	Negative
BIV-1801-1448-E1	PERFORM	blood	Other hepatitis	Negative

5. Throughout the manuscript when talking about results on subsets of samples, we have identified the case or control number (eg highlighted in line 197)

Referee #2 (Remarks to the Author):

My main concerns were about the indirect evidence and the confusing nature of the study presented. The revised manuscript addresses most of my main concerns, and some new data included that adds additional depth to the analysis.

Despite this, there is still a lot of uncertainty about the causes of the liver disease seen in the children, but that is better reflected in the current version.

Please see the response to reviewer 1.

Some technical or remaining issues to address, mainly about parvovirus and AAV details:

1) Lines 172 - 180. The statements regarding "high levels or abundant" AAV2 DNA seems to be at odds with the number of sequences being reported 7-42/million, while 1.2-42/million is called "lower"...

We have removed the word "lower" in relation to the AAV2 reads in the blood. Lines 155-157 (highlighted in the text).

We have left the word "lower" in relation to HHV-6B as the upper limit of reads is less than the lower limit of AAV2 reads (lines 157-159).

2) Ct values and DNA amounts needs to be clarified (e.g. around line 235 or line 240+, but also elsewhere) - what is called "low" and what is "high" is reversed, so care should be taken to indicate what the levels of DNA are - as that is the key finding of interest.

We have removed references to low or high-cycle thresholds and substituted high or low levels of DNA with CT values in brackets, highlighted throughout the manuscript.

3) Various places - the HSPG binding of the original prototypic AAV-2 strains is likely a tissue culture adaptation since that virus was isolated as a contamination of an adenovirus culture, and appears not to be a feature of viruses infecting humans - the text should reflect that.

We have shortened the results section (lines 291-299) and indicated that the current strains all differ from the canonical strain (highlighted lines 419-420).

4) Care should be taken about the DNA sequence arrangements being "proof" of active replication. Line 535 - the concatamers of DNA are not proof of active AAV-2 replication, but are often arising from the introduction of viral DNA that is rearranged within the cell (i.e. most AAV vectors do not have Rep present (e.g. ref 7 or 44)) yet the DNA ends up in complex configurations likely due to cell responses to naked linear DNAs in the nucleus which is subject to DNA damage or other responses.

We agree with the referee and have changed the wording to better reflect his/her concerns.

The highlighted results (lines 173-174) now read "compatible with AAV2 replication". The highlighted discussion (lines 383-384) now reads "may occur during AAV2 replication⁸, while abnormal AAV2 DNA complexes and rearrangements have been observed in the liver following AAV gene therapy^{7,44}."

5) Line 557 and nearby - the lack of capsids is not related to the DNA replication, as capsids would form whether or not ssDNA is present (the genome is packaged into a preformed and stable capsid during replication).

We have removed this sentence.

Referee #3 (Remarks to the Author):

I acknowledge that the authors have addressed my comments. There are still some gaps and unknowns, but the additional evidence support the hypothesis and conclusion from the initial findings.

We thank this referee for their confidence in the data.

Referee #4 (Remarks to the Author):

Morfopoulou et al have added several new analyzes and findings to the revised paper. In particular, the Nanopore sequencing and in vitro experiments to investigate hepatotropism are nice additions offering more molecular details of AAV2 infection in undiagnosed hepatitis. The authors have also improved the presentation of the results and the study is less challenging to follow than before but misses a clear thesis. The authors have performed a large number of different analyses and investigations which are each individually interesting but it is unclear what the main conclusion from all these different findings are. A shorter and more focused discussion is one thing that could help here.

We thank this referee for their helpful suggestions. We have tried to clarify better the results and shortened the discussion by half.

At the end of the summary and in response to one of my questions (#16) the authors note they believe HHV-6B has contributed to pathogenesis, in particular in more severe cases. Is this conclusion (regarding severity) based on detection in explanted liver but not in blood? Apologies if I missed more data here but this alone seems insufficient and open to many biases (e.g. tropism, sensitivity and small sample sizes).

We have now clarified this further in the discussion (highlighted in lines 385-391).

“The pattern of complexes typify both HAdV and herpesvirus (including HHV-6B)-mediated AAV2 DNA replication⁶. The presence of HHV-6B DNA in 11/12 explanted livers, but not in livers (0/2) of non-transplanted children, or control livers as well as the expression, in 5/5 cases tested, of HHV-6B proteins, including U43, a homologue of the HSV1 helicase primase UL52 which is known to aid AAV2 replication, highlight a possible role for HHV-6B as well as HAdV, in the pathogenesis of AAV2 hepatitis, particularly in severe cases.”

We do not mention detection of HHV-6B in blood or serum since positivity was not a reliable indicator of HHV-6B in the liver; three of four whole bloods (matched to frozen liver samples) and none of three serum samples (matched to FFPE liver samples) i.e. only 3/7 liver transplant patients positive for HHV-6B in the liver were positive blood/serum.

I find the long-read sequencing data intriguing, but am concerned that 6-12 reads is insufficient to confidently determine these replication phenomena robustly. Can the authors sequence these libraries more deeply or use adaptive sequencing to selectively enrich for AAV2 sequences?

The original sequencing generated 5.8-22 million reads of sequence, and we apologise for not including this information in the manuscript. We have now increased this to 83-122 million reads (Supplementary Table 3) for two of the explanted livers for which we have sufficient material. We were concerned that adaptive sampling would fail to recognise aberrant AAV2 reads, and is not yet optimised for use with a Promethion, which was needed to increase the depth of the sequencing, and so we did not use this technique. This depth of sequence yielded 51-178 AAV2 reads with N50 8.7-9.0 kb per sample. Analysis of these reads shows the same patterns as previously observed with concatenated genomes, and over 40% of the reads showing abnormal forms with evidence of rolling circle and rolling hairpin forms. The additional information is captured in the results (highlighted lines 176-181) and in Supplementary Table 3.

It seems to me that for the lack of detection of HAdV and HHV6 RNA a lack of sensitivity to detect this cannot be ruled out, in particular given the lower viral loads of these viruses compared to AAV2 by PCR and DNaseq and the detection of HHV6 protein. This analysis should be more carefully caveated or deeper sequencing or targeted enrichment (as done for the DNA) may be warranted.

We agree that the lack of evidence for HAdV and HHV-6B may reflect a sensitivity issue, particularly as RNA degradation is a problem even in livers frozen at -80°C. Our diagnostic RNA metagenomics assay has been shown to have the same sensitivity as PCR for detection of RNA in tissue (although up to 1-2 logs lower for detection of DNA) in over 100 tissue biopsies used to validate the test. We therefore did not proceed to PCR enrichment for mRNA for HAdV or HHV-6B. Our experience and that of others (<http://dx.doi.org/10.1136/gutjnl-2021-324280>) suggests that protein is more stable in tissue than RNA, so we were not surprised to find HHV-6B proteins in liver even when negative by RNAseq.

It is unfortunate that no viral proteins are detectable by IHC, microscopy or proteomics which would support direct infection. The authors suggest time since infection as one possible reason for this, but is there evidence that the assay is sufficiently sensitive? My understanding is that other studies have identified such evidence in similar samples with a similar time since infection.

Mass spectrometry-based proteomics is a highly sensitive technique, but the depth of proteome coverage achieved depends on the abundance of proteins present in the analysed sample. Highly abundant proteins can mask the lower abundant ones. A means of achieving more proteome depth is to fractionate the samples prior to mass spectrometry analysis. When we first analysed these samples, we did not perform any such fractionation, however, when we reanalysed them, we fractionated the samples which allowed us to identify a higher number of proteins including HHV-6B. Even this sensitive method did not detect HAdV or AAV2. The HAdV IHC is a commercial test and has been validated diagnostically for detection of HAdV proteins. Although we are not aware of data on its sensitivity, it is likely to be more sensitive than either the AAV2 or HHV-6B assays, neither of which are in routine diagnostic use. As mentioned above, proteins may remain stable and detectable in tissue for days after the initial viral replication.

We are not aware of any other studies of paediatric hepatitis of unknown aetiology that have detected viral proteins in liver biopsies or other samples.

While I appreciate the broader efforts to simplify presentation of the sample sets, I find the revised abstract to be a little misleading. Specifically, the phrase; “using genomic, transcriptomic, proteomic and immunohistochemical methods, we undertook extensive investigation of 39 cases, ...” overstates the sample sets analyzed. For example, I believe only 5 cases were studied for both proteomic and IHC investigations.

We have removed the word “extensive” in line 93. More detailed explanation of what was done to which specimens would lengthen the abstract too much. Instead, we have added the words “a combination of” to line 89.

The figures are not of as high quality as would be expected for publication. In figure 1, text is too small in figure 1B and what is the Y axis for the red and green dots on 1A? Figure 3 is very challenging to read and lacks important information including confidence values for A and C (they seem to be there for B but are too small to see).

We have altered the figures to address the issues outline above.

Editorial Requests

Statistics: Wherever statistics have been derived (e.g. error bars, box plots, statistical significance) the legend needs to provide and define the n number (i.e. the sample size used to derive statistics) as a precise value (not a range), using the wording “n=X biologically independent samples/animals/cells/independent experiments/n= X cells examined over Y independent experiments” etc. as applicable. Legends requiring revision:

Please note that this information is missing in the legends of figures 2d-g, 4a.

N values have been added to the axis labels of Figure 2 and to the legend of figure 4a

All error bars need to be defined in the legends (e.g. SD, SEM) together with a measure of centre (e.g. mean, median). For example, the legends should state something along the lines of “Data are presented as mean values +/- SEM” as appropriate. All box plots need to be defined in the legends in terms of minima, maxima, centre, bounds of box and whiskers and percentile

1. Please note that the error bars need to be defined in the legend of supplementary figure 6

Done

2. Please note that the box plots need to be defined in terms of whiskers and percentile in the legends of figures 2d-g

Done

The figure legends must indicate the statistical test used. Where appropriate, please indicate in the figure legends whether the statistical tests were one-sided or two-sided and whether adjustments were made for multiple comparisons. For null hypothesis testing, please indicate the test statistic (e.g. F, t, r) with confidence intervals, effect sizes, degrees of freedom and P values noted. Please provide the test results (e.g. P values) as exact values whenever possible and with confidence intervals noted.

1. Please indicate the statistical test used for data analysis and where appropriate, please specify whether it was one-sided or two-sided and whether adjustments were made for multiple comparisons, in the legends of figures 4b-c, supplementary figure 6, supplementary tables 14, 15, 16

Done

2. Please note that the information on whether the statistical test used was one-sided or two-sided, where appropriate, is missing in the legends of figure 2d, supplementary figure 10

Done

3. Please note that the exact p value should be provided, when possible, in the legends of figures 2d-f, supplementary figure 6, supplementary tables 14, 15, 16

Done

Exact p values were already included in supplementary tables 14-16 (now 12-14) , so please could you clarify this if you require any additional information?

Reproducibility: Please state in the legends how many times each experiment was repeated independently with similar results. This is needed for all experiments, but is particularly important wherever results from representative experiments (such as micrographs) are shown. If space in the legends is limiting, this information can be included in a section titled “Statistics and Reproducibility” in the methods section

Please note that this information is missing in the legends of supplementary figures 8a-e, 9 (i), 9 (ii) a-f

Done

Micrographs: Please ensure that all micrographs include a scale bar and this scale bar is defined on the panels or in the figure legends

Please note that scale bar is missing for supplementary figures 8a-e.

Done